# MagIC-Cryo-EM, structural determination on magnetic beads for scarce macromolecules in heterogeneous samples

**Yasuhiro Arimura[1,2]\*, Hide A Konishi[1], Hironori Funabiki[1]\***

[1]Laboratory of Chromosome and Cell Biology, The Rockefeller University, New York, United States; [2]Basic Sciences Division, Fred Hutchinson Cancer Center, Seattle, United States

## eLife Assessment

This study follows up on Arimura et al's powerful new method MagIC-Cryo-EM for imaging native complexes at high resolution. Using a clever design embedding protein spacers between the antibody and the nucleosomes purified, thereby minimizing interference from the beads, the authors concentrate linker histone variant H1.8 containing nucleosomes. From these samples, the authors obtain **convincing** atomic structures of the H1.8 bound chromatosome purified from interphase and metaphase cells, finding a NPM2 chaperone bound form exists as well. Caveats previously noted have been addressed nicely in the revision, strengthening the overall conclusions. This is an **important** new tool in the arsenal of single molecule biologists, permitting a deep dive into structure of native complexes, and will be of high interest to a broad swathe of scientists studying native macromolecules present at low concentrations in cells.

**\*For correspondence:**
yarimura@fredhutch.org (YA);
funabih@rockefeller.edu (HF)

**Abstract** Cryo-EM single-particle analyses typically require target macromolecule concentration at 0.05~5.0 mg/ml, which is often difficult to achieve. Here, we devise *M*agnetic *I*solation and *C*oncentration (MagIC)-cryo-EM, a technique enabling direct structural analysis of targets captured on magnetic beads, thereby reducing the targets' concentration requirement to <0.0005 mg/mL. Adapting MagIC-cryo-EM to a Chromatin Immunoprecipitation protocol, we characterized structural variations of the linker histone H1.8-associated nucleosomes that were isolated from interphase and metaphase chromosomes in *Xenopus* egg extract. Combining *D*uplicated *S*election *T*o *E*xclude *R*ubbish particles (DuSTER), a particle curation method that excludes low signal-to-noise ratio particles, we also resolved the 3D cryo-EM structures of nucleoplasmin NPM2 co-isolated with the linker histone H1.8 and revealed distinct open and closed structural variants. Our study demonstrates the utility of MagIC-cryo-EM for structural analysis of scarce macromolecules in heterogeneous samples and provides structural insights into the cell cycle-regulation of H1.8 association to nucleosomes.

## Introduction

Recent advances in cryogenic electron microscopy (cryo-EM) technology have enabled the structural characterization of biomolecules isolated from their native conditions (*Azinas and Carroni, 2023*). However, the necessity for high sample concentration restricts its applicability to abundant targets (*Natchiar et al., 2017*; *Arimura et al., 2021*; *Arimura and Funabiki, 2022*; *Leesch et al., 2023*). The vitrification step of cryo-EM is a major contributor to this bottleneck. In a conventional plunge

**eLife digest** Whether a protein can properly carry out its role in a cell heavily depends on its unique three-dimensional shape. To understand the biological mechanisms underlying protein function – and how they can go awry in disease – scientists therefore need to understand the structure of each of these molecules with a high degree of detail. Researchers use computer-based systems, such as AlphaFold, to predict protein structures based on the properties of their building blocks. However, these predictions may not be accurate, and still need to be verified experimentally.

Cryo-electron microscopy (cryo-EM) is a technique that allows researchers to directly visualize proteins. The approach first requires flash-freezing a sample at extremely low temperatures, and then sending a beam of electrons through the specimen from several angles. The resulting series of two-dimensional images are analysed using advanced algorithms to construct a final three-dimensional protein model.

Despite its power, cryo-EM also has limitations, including requiring very large quantities of protein in the initial sample. The algorithms currently available for Cryo-EM analyses also struggle to determine the structure of smaller molecules.

To overcome these limitations, Arimura et al. developed MagIC-cryo-EM, a new method for preparing samples that relies on magnetic nano-beads. Proteins of interest are directly attached onto these particles using a short linking molecule, and the entire complex is then concentrated onto the imaging surface using a magnet. This greatly reduces the amount of protein needed in the initial sample. A refined analysis method, called DuSTER, was designed in parallel to help capture the structure of smaller molecules.

Arimura et al. successfully tested both approaches on the histone protein H1.8, which normally binds to cellular DNA and helps 'package' it stably. The experiments showed, for the first time, that H1.8 could exist in two distinct three-dimensional shapes. They also revealed the structure of a much smaller molecule associated with H1.8, NPM2.

MagIC-cryo-EM and DuSTER will allow scientists to use cryo-EM to examine the structure of proteins which are smaller, challenging to isolate or present in minute quantities inside cells. The knowledge gained from these findings could help us better understand a wide range of biological processes.

vitrification method, 3 µL of aqueous samples greater than 1 mg/mL are typically required to acquire sufficient numbers of particle images on cryo-EM micrographs for 3D structure reconstruction (*Table 1 Bhella, 2019*). This is because most of the target complexes in the sample solution applied on a cryo-EM grid must be removed by a blotting paper to make a thin ice layer suitable for analysis. Several methods are currently available to lower sample volume/concentration needed (*Table 1*). Jet vitrification (*Ravelli et al., 2020*) and Spotiton (*Dandey et al., 2018*) require sub-nanoliters of the sample volume, but they still require high-concentration samples. Affinity grids, such as Ni-NTA lipid monolayer grids (*Kelly et al., 2008*), chemically functionalized grids (*Llaguno et al., 2014*), antibody-attached grids (*Yu et al., 2016*), and streptavidin monolayer grids (*Wang et al., 2008*), are amenable for lower concentration samples (~0.05 mg/mL), but concentrating natively isolated targets to such a level and reproducibly generating the affinity grids remains challenging.

Structural characterization of native chromatin-associated protein complexes is particularly challenging due to their heterogeneity and scarcity: more than 300 proteins directly bind to the histone core surface (*Skrajna et al., 2020*), while each of these proteins is targeted to only a fraction of nucleosomes in chromatin. For their structural analysis, it is a common practice to assemble nucleoprotein complexes using purified recombinant proteins and a specific short (10–1000 bp) linear DNA. However, this reconstitution approach has a limitation since the structure and function of chromatin proteins can be altered by several variances under native conditions, such as DNA sequence, DNA and protein modifications, and short- and long-scale DNA folding. Although isolation of the endogenous chromatin-associated complexes can be achieved through *c*hromatin *i*mmuno*p*recipitation (ChIP; *Hebbes et al., 1988*; *Solomon et al., 1988*; *Gilmour and Lis, 1986*) to determine the associated DNA sequences and proteins (*Zou et al., 2022*; *Wang et al., 2013*), the amount obtained by this method is too little to apply for conventional structural analysis.

**Table 1.** The required sample amount for analyzing chromatin samples.

| Methods | Purity | Concentration | Volume | Amount | Advantage | Disadvantage |
|---|---|---|---|---|---|---|
| Cryo-EM (conventional) | Purified | 0.5~5.0 mg/mL | 3~4 µL/grid | >10 µg/sample | simple | High concentration of sample is required |
| Cryo-EM (Jet vitrification) | Purified | 4 mg/mL | 0.001 µL/grid | >4 ng/sample | Very low volume of sample is required | High concentration of sample is required |
| Cryo-EM (Affinity grid) | Crude | 0.05 mg/mL | 3~4 µL/grid | >1 µg/sample | Sample can be isolated and concentrated on grid | The maximum sample volume is limited |
| ChIP-seq | Crude | - | - | 10~50 ng DNA | Sample can be isolated and concentrated by beads | |
| SDS-PAGE (CBB stain) | Crude | 0.005~0.100 mg/mL | 1~20 µL/lane | >30 ng/band | | |
| MagIC-cryo-EM | Crude | <0.0005 mg/mL | 1~2000 µL | >5 ng (2 ng DNA)/grid | Sample can be isolated and concentrated by beads | cryo-EM data collection points are selected manually. |
| SDS-PAGE (Silver stain) | Crude | 0.0001~0.001 mg/mL | 1~20 µL/lane | >1 ng/band | | |

To obtain high-resolution cryo-EM structures of chromatin-associated protein complexes while they are functioning on the native chromosomes, we previously analyzed structural variation of nucleosomes isolated from interphase and metaphase chromosomes formed in *Xenopus laevis* egg extracts (*Arimura et al., 2021*). We found that the averaged structures of the nucleosome core particle (NCP) in interphase and metaphase chromosomes are essentially identical to the NCP crystal structure assembled with histone proteins and DNA with strong nucleosome positioning sequences (*Luger et al., 1997*; *Chua et al., 2012*). We also observed that the major structural variation of the nucleosome structures between interphase and metaphase chromosomes was attributable to the binding status of the oocyte-specific linker histone H1.8. We were able to resolve the 3D structure of the H1.8-bound nucleosome isolated from metaphase chromosomes but not from interphase chromosomes (*Arimura et al., 2021*). The resolved structure indicated that H1.8 in metaphase is most stably bound to the nucleosome at the on-dyad position, in which H1 interacts with both the entry and exit linker DNAs (*Bednar et al., 2017*; *Zhou et al., 2015*; *Zhou et al., 2021*; *Dombrowski et al., 2022*). This stable H1 association to the nucleosome in metaphase likely reflects its role in controlling the size and the shape of mitotic chromosomes through limiting chromatin accessibility of condensins (*Choppakatla et al., 2021*), but it remains unclear why H1.8 binding to the nucleosome in interphase is less stable. Since the low abundance of H1.8-bound nucleosomes in interphase might have prevented us from determining their structure, we sought to solve this issue by enriching H1.8-bound nucleoprotein complexes through adapting ChIP-based methods.

Aiming to reduce sample requirements for single particle cryo-EM analyses to levels lower than those widely used for ChIP-seq (10–50 ng DNA, *Table 1*; *Zou et al., 2022*), here we developed *Mag*netic *I*solation and *C*oncentration (MagIC)-cryo-EM, which enables direct cryo-EM analysis of target molecules enriched on superparamagnetic nanobeads. By adapting the ChIP protocol to MagIC-cryo-EM, we successfully determine the ~4 Å resolution structures of H1.8-GFP-bound nucleosomes using highly heterogeneous dilute fractions isolated from metaphase and interphase chromosomes. In addition, by combining the particle curation method, *D*uplicated *S*election *T*o *E*xclude *R*ubbish particles (DuSTER), which effectively removes particles with a low signal-to-noise ratio (S/N), we revealed structural variations of the H1.8-bound chaperone NPM2 isolated from interphase chromosomes, providing structural insights into the cell cycle regulation of H1.8 stabilization on nucleosomes.

## Results

### Development and optimization of MagIC-cryo-EM using nucleosomes

Inspired by a report using 200–300 nm superparamagnetic beads directly loaded onto a cryo-EM grid to image viral particles (*Bonnafous et al., 2010*), we examined the feasibility of 50 nm streptavidin nanobeads for cryo-EM single-particle analysis using poly-nucleosome arrays as pilot targets

(*Figure 1A*). Nanobeads were easily identified on the grid as black dots in the intermediate-magnification montage map (*Figure 1B*), facilitating target identification for subsequent high-magnification data collection. In the high-magnification micrographs, poly-nucleosome fibers were observed around the nanobeads as expected (*Figure 1C*). Using nucleosome-like particles selected from 550 micrographs by the machine-learning-based software Topaz (*Bepler et al., 2019a*), we successfully determined the 3D structure of the nucleosome at sub-nanometer resolution (*Figure 1D*). This result, however, revealed a notable issue; an intense halo-like scattering covered an ~30 nm radius around the nanobeads (*Figure 1D*, blue areas), interfering with the signal from particles that were proximal to the beads.

To reduce the effect of the halo-like scattering surrounding the nanobeads, a protein spacer module was attached to the beads so that the target biomolecules are placed outside the reach of the halo (*Figure 2A and B*). After several rounds of optimization using the in vitro reconstituted H1.8-bound nucleosome as a model target, we chose a spacer module comprising an 11 nm triple helical bundle (3HB) protein (*Huang et al., 2014*) and four copies of a 60 nm single alpha helix (SAH) protein (*Sivaramakrishnan and Spudich, 2011*) for its effectiveness and reasonable production yield (*Figure 2*, *Figure 2—figure supplement 1*). The distal end of the spacer module was engineered to allow for exchangeable target-capturing modules by SPYcatcher-SPYtag conjugation (*Figure 2B*; *Zakeri et al., 2012*). We hereon refer to these magnetic nanoparticles coated with the spacer and target-capturing modules as MagIC-cryo-EM beads.

To assess the feasibility of the MagIC-cryo-EM beads for structural analysis of a low-concentration target in heterogeneous samples, we isolated H1.8-GFP-bound nucleosomes by anti-GFP nanobody coupled to the MagIC-cryo-EM beads from a mixture of H1.8-GFP nucleosomes (1.7 nM, or 0.00047 mg/mL) and a large excess of unbound mono-nucleosomes (53 nM, or 0.012 mg/mL; *Figure 2C and D*). This target concentration was approximately 100–1000 times lower than the concentration required for conventional cryo-EM methods, including affinity grid approaches (*Kelly et al., 2008*; *Llaguno et al., 2014*; *Yu et al., 2016*). The magnetic beads were captured on a cryo-EM grid by neodymium magnets for 5 min in a humidified chamber (*Figure 2E*). This magnetic capture step significantly increased the number of beads that were found in the sample holes of the grid (*Figure 2F–I*), thereby mitigating the sample loss caused by filter paper blotting to generate a thin ice layer .

High-magnification micrographs of MagIC-cryo-EM beads show that the spacer module successfully placed nucleosome-like particles outside the halo-like scattering surrounding the nanobeads (*Figure 2J*). The local enrichment of target molecules around MagIC-cryo-EM beads offers a substantial advantage in data collection efficiency over available cryo-EM methods (*Kelly et al., 2008*; *Llaguno et al., 2014*; *Yu et al., 2016*), in which target molecules are disseminated across the grids and are difficult to identify. In contrast, the magnetic beads are easily identified in the Medium-Magnification Montage (MMM) map (*Figure 2G*), enabling the selection of target-rich areas prior to high-magnification data collection. Indeed, approximately 100 H1.8-GFP nucleosome particle images per bead were efficiently collected even with a sample concentration as low as 0.00047 mg/mL of H1.8-GFP nucleosomes in the heterogeneous sample (*Figure 2J* right panel).

After removing junk particles using decoy classification (*Arimura et al., 2021*; *Gong et al., 2016*; *Nguyen et al., 2019*; *Lilic et al., 2020*; *Figure 2—figure supplement 2*), an H1.8 density-containing nucleosome class was isolated via ab initio reconstruction and heterogeneous refinement using cryoSPARC (*Punjani et al., 2017*). Among the nucleosome-containing particles, 55.7% of them were classified as a nucleosome with H1.8 at the on-dyad position (*Figure 2—figure supplement 2*), yielding a final 3D structure at 3.6 Å resolution (*Figure 2K*). This high fraction of H1.8-bound nucleosome particles indicated that the MagIC-cryo-EM beads efficiently isolated the target molecules. Notably, this method only required 5 ng of H1.8-GFP-bound nucleosomes (including 2 ng of DNA) per cryo-EM grid, which is comparable to or even lower than the requirements of widely used ChIP-seq (*Zou et al., 2022*).

## MagIC-cryo-EM application for ChIP to assess structural features of H1.8 in chromosomes

We next adapted MagIC cryo-EM to ChIP protocols to elucidate the cell-cycle-specific mechanism that controls H1.8 stability on interphase and metaphase nucleosomes. We previously reported the cryo-EM structure of *Xenopus* H1.8 bound to the metaphase nucleosome at the on-dyad position,

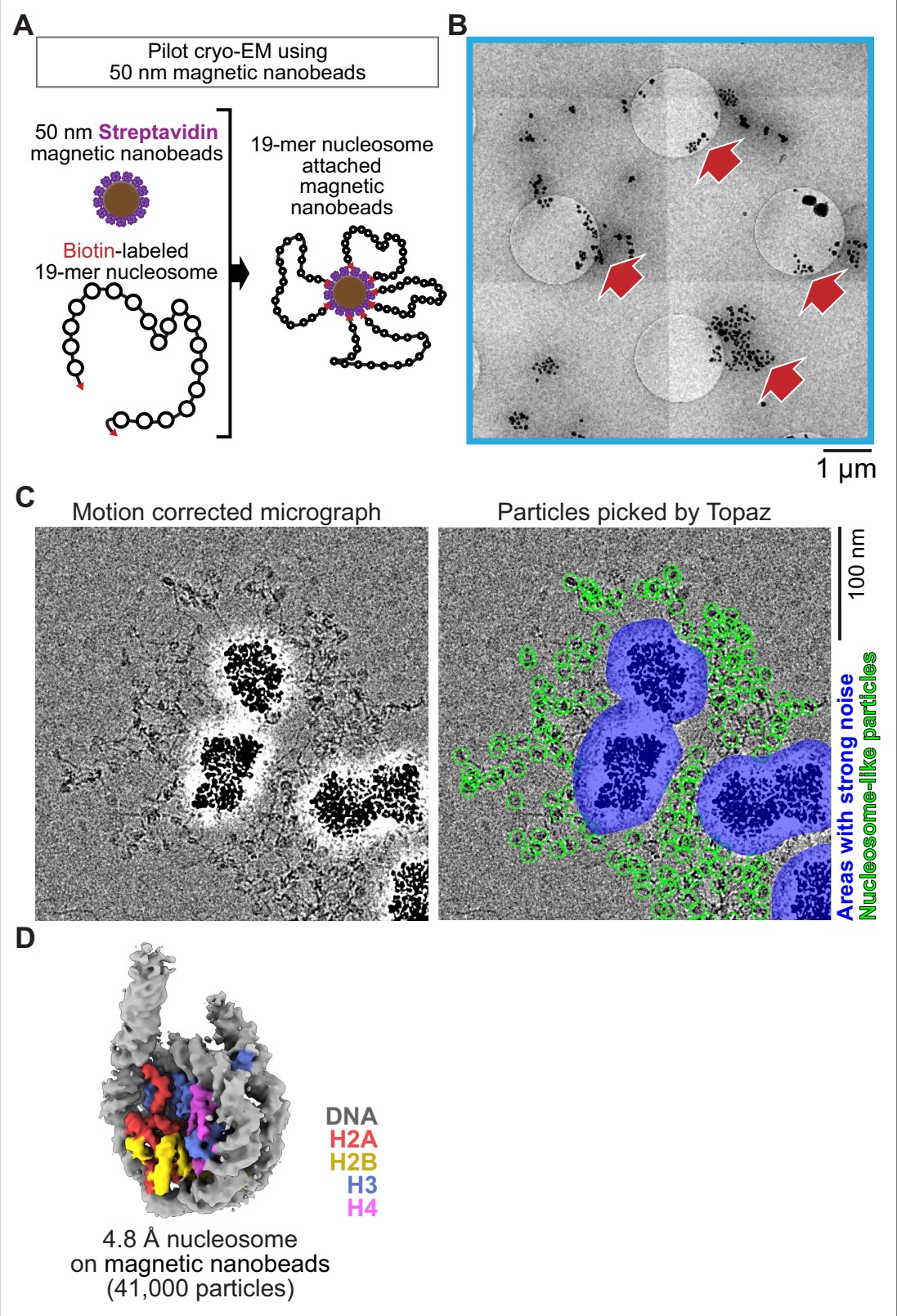

**Figure 1.** Single particle cryo-EM analysis of poly-nucleosomes attached to magnetic beads. (**A**) Schematic of a pilot cryo-EM experiment on magnetic beads. Biotin-labeled 19-mer nucleosome arrays attached to 50 nm streptavidin-coated magnetic nanobeads were loaded onto the cryo-EM grid. (**B**) Representative medium magnification micrographs. The magnetic beads are seen as black dots (red arrows). (**C**) Left; a representative highmagnification micrograph. The micrograph was motion-corrected and low-pass filtered to 5 Å resolution. Right; green circles indicate the nucleosome-like particles selected by Topaz, and the blue areas indicate the halo-like scattering. (**D**) The 3D structure of the nucleosome bound on magnetic beads.

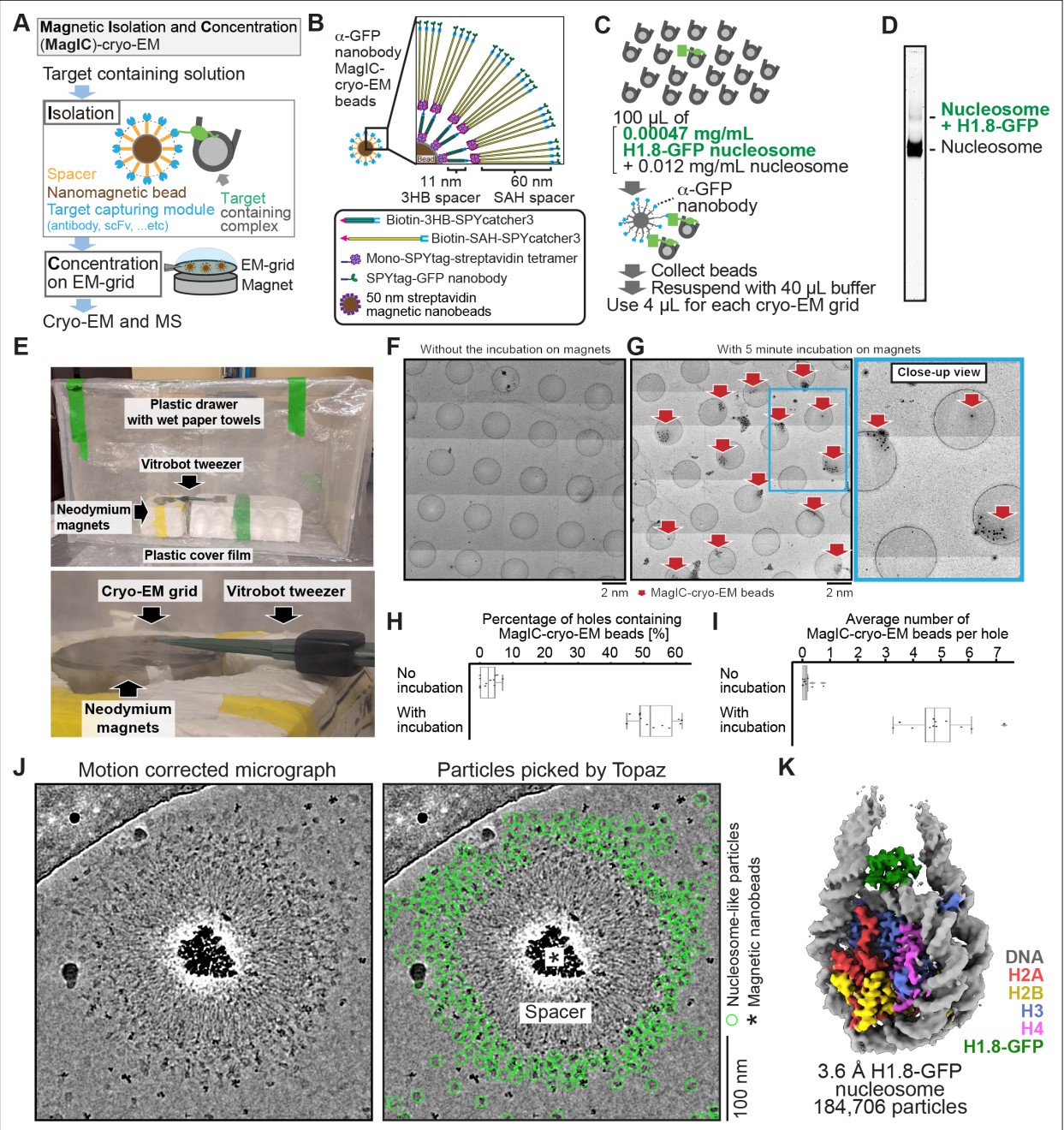

**Figure 2.** MagIC-Cryo-EM structural determination of low-quantity and low-purity targets. (**A**) Schematic depicting the principle steps of MagIC-cryo-EM. (**B**) Graphical representation of the MagIC-cryo-EM beads with 3HB and SAH spacers and GFP nanobody target capture module. (**C**) Schematic of MagIC-cryo-EM for in vitro reconstituted H1.8-GFP-bound nucleosomes isolated from an excess of H1.8-free nucleosomes. (**D**) Native PAGE analysis of H1.8-GFP-bound nucleosomes and unbound nucleosomes in the input. DNA staining by SYTO-60 is shown. (**E**) A handmade humidity chamber used for the 5 min incubation of the cryo-EM grids on the magnet. The humidity chamber was assembled using a plastic drawer. Wet tissues are attached to the side walls of the chamber, which is sealed with a plastic cover to maintain high humidity. Two pieces of neodymium magnets are stacked. A graphene grid is held by a non-magnetic vitrobot tweezer and placed on the magnets. 4 µL of sample is applied on the grid and incubated for 5 min. (**F**) Micrograph montage of the grids without using magnetic concentration. The GFP-nanobody-MagIC-cryo-EM beads (4 µL of 12.5 pM beads) were applied on the graphene-coated Quantifoil R 1.2/1.3 grid and vitrified without incubation on a magnet. (**G**) Micrograph montage of the grids without using magnetic concentration. The GFP-nanobody-MagIC-cryo-EM beads (4 µL of 12.5 pM beads) were applied on the graphene-coated Quantifoil R 1.2/1.3 grid and vitrified with 5 min incubation on two pieces of 40x20 mm N52 neodymium disc magnets. (**H**) Quantitative analysis of the percentage of holes containing MagIC-cryo-EM beads. Each data point represents the percentage of holes containing MagIC-cryo-EM beads on each square mesh. (**I**) Quantitative analysis of the average number of MagIC-cryo-EM beads per hole. Each data point represents the average number of MagIC-cryo-

*Figure 2 continued on next page*

*Figure 2 continued*

EM beads per hole on each square mesh. The edges of the boxes and the midline indicates the 25th, 50th, and 75th percentiles. Whiskers indicate the maximum and lowest values in the dataset, excluding outliers. For the quantification, 11 square meshes with 470 holes without magnetic concentration and 11 square meshes with 508 holes with 5 min incubation on magnets were used. (**J**) Representative motion corrected micrographs of in vitro reconstituted H1.8-GFP nucleosomes captured by MagIC-cryo-EM beads. The micrographs were low-pass filtered to 10 Å resolution. Green circles indicate the nucleosome-like particles picked by Topaz. (**K**) 3D structure of the in vitro reconstituted H1.8-GFP-bound nucleosome determined through MagIC-cryo-EM. The pipeline for structural analysis is shown in *Figure 2—figure supplement 2*.

The online version of this article includes the following source data and figure supplement(s) for figure 2:

**Figure supplement 1.** Optimization of the MagIC-cryo-EM beads.

**Figure supplement 1—source data 1.** The full image of gel shown in *Figure 2—figure supplement 1B*.

**Figure supplement 1—source data 2.** The raw image of gel shown in *Figure 2—figure supplement 1*.

**Figure supplement 2.** MagIC-cryo-EM single particle analysis of in vitro reconstituted H1.8-GFP-bound nucleosome.

whereas no H1.8-containing structures were reconstructed from interphase chromosomes (*Arimura et al., 2021*; *Figure 3A*, left). Despite the high accumulation of H1.8 in the nucleus (*Maresca et al., 2005*; *Figure 3B*), the amount of nucleosome-associated H1.8 in interphase is reduced to approximately 30% of that in metaphase (*Arimura et al., 2021*). Given the high mobility of the linker histone H1 on chromatin (*Choppakatla et al., 2021*; *Willcockson et al., 2021*; *Yusufova et al., 2021*), we hypothesized that H1.8 on nucleosome is destabilized by an interphase-specific mechanism. By enriching H1.8-bound nucleosomes from interphase and metaphase chromosomes using MagIC-cryo-EM, we intended to examine if H1.8 in interphase preferentially associates with nucleosomes at more unstable binding positions, such as at off-dyad positions (*Zhou et al., 2013*; *Song et al., 2014*; *Figure 3A*, positioning model), or if there is an interphase-specific mechanism (by chaperones, for example) that dissociates H1.8 from nucleosomes (*Figure 3A*, chaperone model), although the amount H1.8-bound NAP1, the known histone H1.8 chaperone *Miller and Heald, 2015*, did not differ between metaphase and interphase egg extracts (*Figure 3—figure supplement 1*).

To distinguish between these models, we applied MagIC-cryo-EM to enrich H1.8-bound nucleosomes from chromosomes assembled in interphase and metaphase *Xenopus* egg extracts. Sperm nuclei were incubated in egg extracts supplemented with H1.8-GFP to obtain replicated interphase chromosomes and metaphase chromosomes, which were crosslinked and fragmented to generate soluble nucleoprotein complexes (*Figure 3B*). We confirmed that H1.8-GFP is functional as it rescued the chromosome elongation phenotype caused by H1.8 immunodepletion (*Choppakatla et al., 2021*; *Maresca et al., 2005*; *Figure 3—figure supplement 1*). Sucrose density gradient centrifugation was conducted to separate different H1.8-containing complexes, including mono-nucleosome fractions and oligo-nucleosome fractions, as previously described *Arimura et al., 2021* (*Figure 3C*, *Figure 3—figure supplement 2*, and *Table 2*). As we had predicted that more H1.8 proteins would associate with nucleosomes in metaphase than in interphase *Arimura et al., 2021*, we increased the quantities of egg extract and sperm nuclei by 2.5-fold to prepare comparable amounts of H1.8-bound interphase nucleosomes as compared to metaphase (*Figure 3C*, fractions 4–11). To prevent the dissociation of H1.8 from nucleosomes during DNA fragmentation, the MNase concentration and the reaction time were optimized to generate DNA fragment lengths with 180–200 bp (*Figure 3—figure supplement 2B*), which is adequate for linker histone association *Zhou et al., 2015*. To ensure that most nucleosomes isolated through MagIC-cryo-EM were bound by H1.8, we selected the fractions enriched with H1.8-bound mono-nucleosomes (fraction 5 in *Figure 3C and D*), as oligo-nucleosomes (abundant in fractions 6–11) might include H1.8-free nucleosomes. These fractions contain highly heterogeneous protein mixtures (*Figure 3E*), in which H1.8-GFP is a minor constituent with an estimated concentration at 1–2 nM (corresponding to 0.00025–0.0005 mg/mL of H1.8-bound mononucleosomes; *Figure 3—figure supplement 2C*). Mass spectrometry (MS) analysis of these fractions also showed heterogeneity as they included several DNA-binding proteins, such as PCNA (*Table 2* and *Supplementary file 2*).

H1.8-GFP-bound mono-nucleosomes in fraction 5 (from metaphase and interphase chromosomes) were captured by GFP nanobody-MagIC-cryo-EM beads and applied to grids for cryo-EM analysis. MS analysis of the captured MagIC-cryo-EM beads confirmed selective enrichment of H1.8 over other nonhistone proteins found in fraction 5 (*Table 2*). To quantitatively assess the population of

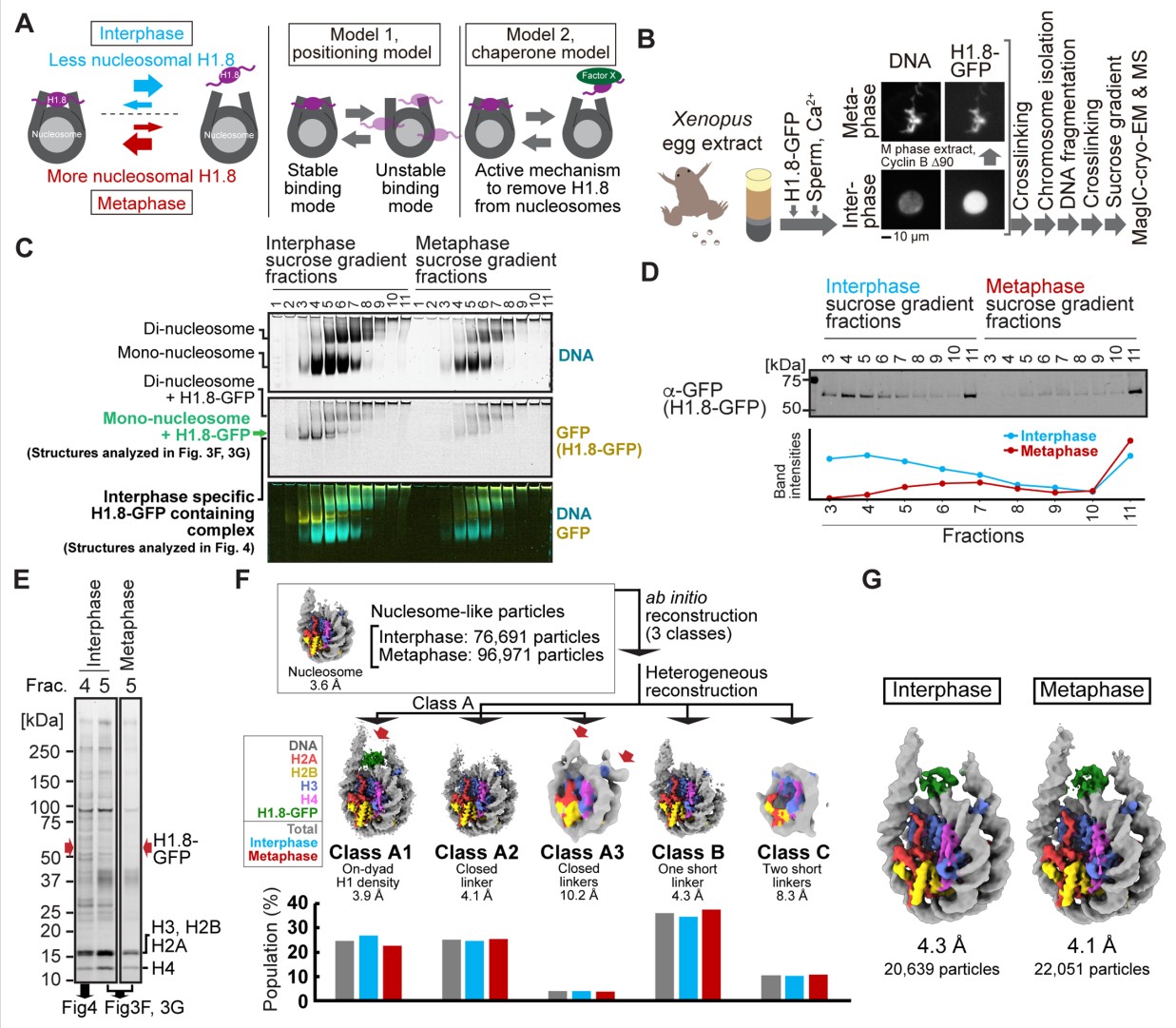

**Figure 3.** MagIC-Cryo-EM structural determination of H1.8-bound nucleosomes from interphase and metaphase chromosomes in *Xenopus* egg extract. (**A**) Models of potential cell cycle-dependent H1.8 dynamic binding mechanisms (**B**) Experimental flow of MagIC-cryo-EM analysis for GFP-H1.8 containing complexes isolated from chromosomes assembled in interphase and metaphase *Xenopus* egg extract. Fluorescence microscopy images indicate localization of GFP-H1.8 to interphase and metaphase chromosomes. DNA and GFP-H1.8 were detected either by staining with Hoechst 33342 or GFP fluorescence, respectively. (**C**) Native PAGE of fragmented interphase and metaphase chromosome sucrose gradient fractions. GFP-H1.8 and DNA were detected with either GFP fluorescence or SYTO-60 staining, respectively. (**D**) Western blot of GFP-H1.8 in interphase and metaphase chromosome sucrose gradient fractions. GFP-H1.8 was detected using anti-GFP antibodies. (**E**) SDS-PAGE of the sucrose gradient fractions 4 and 5 shown in (**C**), demonstrating heterogeneity of the samples. Proteins were stained by gel code blue. Red arrows indicate the H1.8-GFP bands. The full gel image is shown in *Figure 3—figure supplement 2A*. (**F**) In silico 3D classification of interphase and metaphase H1.8-bound nucleosomes isolated from chromosomes in *Xenopus* egg extract. To assess the structural variations and their population of H1.8-bound nucleosomes, ab initio reconstruction and heterogenous reconstruction were employed twice for the nucleosome-like particles isolated by the decoy classification. The initial round of ab initio reconstruction and heterogenous reconstruction classified the particles into three nucleosome-containing 3D models (**A, B, C**). Subsequent ab initio reconstruction and heterogenous reconstruction on the class A, which has weak H1.8 density, yielded three new nucleosome-containing structures, A1, A2, and A3. 3D maps represent the structural variants of GFP-H1.8-bound nucleosomes. Red arrows indicate extra densities that may represent H1.8. Green densities indicate on-dyad H1.8. The bar graphs indicate the population of the particles assigned to each 3D class in both interphase and metaphase particles (gray), interphase particles (blue), and metaphase particles (red). The pipeline for structural analysis is shown in *Figure 3—figure supplement 3A*. (**G**) Structures of H1.8-bound nucleosomes isolated from interphase and metaphase chromosomes.

The online version of this article includes the following source data and figure supplement(s) for figure 3:

**Source data 1.** Full images of gels and membranes shown in *Figure 3*.

**Source data 2.** Raw images of gels and membranes shown in *Figure 3*.

*Figure 3 continued on next page*

*Figure 3 continued*

**Figure supplement 1.** Functional assessment of H1.8-GFP in *Xenopus* egg extract.

**Figure supplement 1—source data 1.** Full images of gels and membranes shown in *Figure 3—figure supplement 1*.

**Figure supplement 1—source data 2.** Raw images of gels and membranes shown in *Figure 3—figure supplement 1*.

**Figure supplement 2.** Sucrose gradient fractions of the fragmented interphase and metaphase chromosomes formed in *Xenopus* egg extract.

**Figure supplement 2—source data 1.** Full images of gels and membranes shown in *Figure 3—figure supplement 2*.

**Figure supplement 2—source data 2.** Raw images of gels and membranes shown in *Figure 3—figure supplement 2*.

**Figure supplement 3.** Single-particle analysis pipeline for the MagIC-cryo-EM of the interphase and metaphase H1.8-GFP-bound nucleosomes formed in *Xenopus* egg extract.

the H1-bound structural modes of interphase and metaphase nucleosomes, we employed in silico mixing 3D classification (*Arimura et al., 2021*; *Hite and MacKinnon, 2017*). Micrographs of interphase and metaphase MagIC-cryo-EM were mixed and used for particle picking and decoy classification to isolate the nucleosome-containing classes (*Figure 3—figure supplement 3*). Subsequently, particles were classified into three nucleosome-containing 3D models (A, B, C), which were generated by ab initio reconstruction (*Figure 3F*, *Figure 3—figure supplement 3A*). Further 3D classification on the class A, which has weak H1.8 density, yielded three new nucleosome-containing structures, A1, A2, and A3 (*Figure 3F*, *Figure 3—figure supplement 3A*). Then, the populations of interphase and metaphase particles in each class were assessed (*Figure 3F*). Only class A1 had an apparent H1.8 density at the on-dyad position of the nucleosome, with 27% and 23% of the nucleosome particles assigned to this class coming from interphase and metaphase conditions, respectively. Although class A2 had linker DNA densities on both sides of the entry/exit sites of the nucleosome in a closed conformation, it did not have a clear H1.8 density. This suggested that the structures of H1.8 in the particles assigned to this class were not uniform, and that the H1.8 density was averaged out during the cryo-EM processing. Class A3, to which 3–4% of the nucleosome particles were assigned, had ambiguous extra densities outside of the on-dyad position (*Figure 3F*, red arrows), possibly representing H1.8 bound to non-dyad positions. Overall, the relative distributions of these five classes were largely similar between interphase and metaphase (*Figure 3F*), and the structures of H1.8-bound nucleosomes in interphase and metaphase were indistinguishable (*Figure 3G*). The structures of GFP-tagged H1.8-bound nucleosomes isolated from *Xenopus* egg extract chromosomes are essentially identical to the endogenous H1.8-bound nucleosome structure we previously determined *Arimura et al., 2021*. Therefore, although the usage of GFP-tagged H1.8 and MagIC-cryo-EM potentially affect the structure of the H1.8-bound nucleosome, we consider these influences to be minimal. Altogether, the results suggest that differential positional preferences of H1.8 on the nucleosome (*Figure 3A*, positioning model) are unlikely to drive the reduced H1.8 association to interphase nucleosomes.

## MagIC-cryo-EM and DuSTER reconstructed cryo-EM structure of interphase-specific H1.8-containing complex, NPM2

Although we could not discern structural differences of H1.8-bound mono-nucleosomes from metaphase and interphase samples, we noticed that substantial portions of H1.8 were enriched in sucrose fractions 3 and 4 isolated from interphase chromosomes but not from metaphase chromosomes (*Figure 3C*). As these interphase-specific H1.8 fractions were lighter than mono-nucleosome-containing fractions, we thought that they may contain regulatory proteins that preferentially dissociate H1.8 from nucleosomes in interphase, in line with the chaperone model (*Figure 3A*).

To characterize these interphase-specific fractions, we sought to determine their structural features using MagIC-cryo-EM. However, our initial attempt failed to reconstitute any reasonable 2D classes of the interphase-specific H1.8-containing complex (*Figure 4—figure supplement 1A*), even though Topaz successfully picked most of the 60~80 Å particles that are visible on motion-corrected micrographs and enriched around the MagIC-cryo-EM beads (*Figure 4—figure supplement 1A*). This was likely due to their small size; most of the particles did not have a high enough S/N to be properly classified during the 2D classifications as they were masked by background noise from the ice and/or spacer proteins (*Figure 4—figure supplement 1B*).

To solve this issue, we devised the particle curation method DuSTER that does not requires the successful 2D classifications (*Figure 4A*). The principle of DuSTER is based on our realization that

**Table 2.** The list of the chromatin proteins detected by mass spectrometry (MS) before and after enrichment on the MagIC-cryo-EM-beads.

The MS analysis was conducted to the sucrose gradient fractions 5, before (*Figure 3C*) and after (*Figure 3E*) enrichment with the GPF nanobody-MagIC-cryo-EM beads. Detectable MS signals for known chromatin proteins and the recombinant proteins used for assembling the MagIC-cryo-EM beads were manually selected and are listed here. See *Supplementary file 2* of the full MS data.

| Description | Full name or functions | Input interphase, fraction 5 | Input metaphase, fraction 5 | MagIC-cryo-EM, interphase, fraction 5 | MagIC-cryo-EM, metaphase, fraction 5 |
|---|---|---|---|---|---|
| XBmRNA11963\| h2ac17.L | histone H2A | $1.37 \times 10^{10}$ | $4.12 \times 10^{9}$ | $2.08 \times 10^{8}$ | $1.49 \times 10^{8}$ |
| XBmRNA31731\| LOC121402261 | histone H2B | $7.87 \times 10^{9}$ | $2.26 \times 10^{9}$ | $1.10 \times 10^{8}$ | $2.25 \times 10^{8}$ |
| XBmRNA31368\| LOC121402047 | histone H4 | $7.34 \times 10^{9}$ | $2.00 \times 10^{9}$ | $1.11 \times 10^{8}$ | $1.63 \times 10^{8}$ |
| XBmRNA36987\| h2aj.L | histone H2A.J | $6.74 \times 10^{9}$ | $3.64 \times 10^{9}$ | $2.44 \times 10^{8}$ | $1.29 \times 10^{8}$ |
| XBmRNA58735\| h2ax.2.L | histone H2A.X | $6.56 \times 10^{9}$ | $1.73 \times 10^{9}$ | $2.44 \times 10^{8}$ | $1.75 \times 10^{8}$ |
| XBmRNA1949\| h2az1.L | histone H2A.Z | $4.50 \times 10^{9}$ | $3.64 \times 10^{9}$ | $2.44 \times 10^{8}$ | $1.52 \times 10^{8}$ |
| XBmRNA75391\| h3-3b.L | histone H3.3 | $3.33 \times 10^{9}$ | $1.19 \times 10^{9}$ | $3.84 \times 10^{7}$ | $5.29 \times 10^{7}$ |
| XBmRNA25971\| npm2.L | Nucleoplasmin | $1.77 \times 10^{9}$ | n.d. | $1.89 \times 10^{8}$ | n.d. |
| XBmRNA28658\| npm2.S | Nucleoplasmin | $1.76 \times 10^{9}$ | n.d. | $1.31 \times 10^{8}$ | n.d. |
| XBmRNA25690\| pcna.L | PCNA | $1.29 \times 10^{9}$ | n.d. | n.d. | n.d. |
| XBmRNA28883\| pcna.S | PCNA | $1.29 \times 10^{9}$ | n.d. | n.d. | n.d. |
| XBmRNA41314\| h1-8.S | Linker histone H1.8 | $7.12 \times 10^{8}$ | $1.93 \times 10^{8}$ | $5.67 \times 10^{7}$ | $6.87 \times 10^{7}$ |
| XBmRNA9392\| supt16h.S | supt16h (histone chaperone FACT complex component) | $3.14 \times 10^{8}$ | n.d. | n.d. | n.d. |
| XBmRNA4198\| supt16h.L | supt16h (histone chaperone FACT complex component) | $3.04 \times 10^{8}$ | n.d. | n.d. | n.d. |
| XBmRNA62381\| ssrp1.S | ssrp1 (histone chaperone FACT complex component) | $2.96 \times 10^{8}$ | n.d. | n.d. | n.d. |
| sfGFP | sfGFP (Tagged with H1.8) | $2.84 \times 10^{8}$ | $8.15 \times 10^{7}$ | $6.02 \times 10^{7}$ | $4.58 \times 10^{7}$ |
| XBmRNA26658\| dnmt1.L | DNA methyltransferase 1 | $1.41 \times 10^{8}$ | n.d. | n.d. | n.d. |
| XBmRNA31618\| dnmt1.S | DNA methyltransferase 1 | $1.35 \times 10^{8}$ | n.d. | n.d. | n.d. |
| XBmRNA5952\| sub1.L | Activated RNA polymerase II transcriptional coactivator p15 | $8.60 \times 10^{7}$ | n.d. | n.d. | n.d. |
| XBmRNA24726\| pclaf.L | PCNA-associated factor | $8.38 \times 10^{7}$ | n.d. | n.d. | n.d. |
| XBmRNA4881\| ran.L | GTP-binding nuclear protein Ran | $7.38 \times 10^{7}$ | n.d. | n.d. | n.d. |
| XBmRNA42021\| msh2.L | DNA repair protein MutS | $7.08 \times 10^{7}$ | n.d. | n.d. | n.d. |
| XBmRNA52420\| mcm4.L | DNA replication licensing factor MCM4 | $6.61 \times 10^{7}$ | n.d. | n.d. | n.d. |
| XBmRNA51197\| mcm6.L | DNA replication licensing factor MCM6 | $6.54 \times 10^{7}$ | n.d. | n.d. | n.d. |
| XBmRNA60815\| rcc2.L | Regulator of chromosome condensation 2 | $6.29 \times 10^{7}$ | n.d. | n.d. | n.d. |
| XBmRNA29645\| pclaf.S | PCNA-associated factor | $5.26 \times 10^{7}$ | n.d. | n.d. | n.d. |
| XBmRNA12919\| rcc1.L | Regulator of chromosome condensation 1 | $5.24 \times 10^{7}$ | n.d. | n.d. | n.d. |
| XBmRNA27060\| mcm7.L | DNA replication licensing factor MCM7 | $5.08 \times 10^{7}$ | n.d. | n.d. | n.d. |

*Table 2 continued on next page*

*Table 2 continued*

| Description | Full name or functions | Input interphase, fraction 5 | Input metaphase, fraction 5 | MagIC-cryo-EM, interphase, fraction 5 | MagIC-cryo-EM, metaphase, fraction 5 |
|---|---|---|---|---|---|
| XBmRNA73621\| LOC108700788 | Importin subunit beta-like isoform X2 | $5.03 \times 10^7$ | n.d. | n.d. | n.d. |
| XBmRNA42017\| msh6.L | DNA repair protein MutS | $4.96 \times 10^7$ | n.d. | n.d. | n.d. |
| XBmRNA39469\| top1.2.S | DNA topoisomerase I | $4.92 \times 10^7$ | n.d. | n.d. | n.d. |
| XBmRNA82329\| kpna7.S | Importin subunit beta | $4.80 \times 10^7$ | n.d. | n.d. | n.d. |
| XBmRNA74418\| csnk2a1.L | Casein kinase II subunit alpha | $4.65 \times 10^7$ | n.d. | n.d. | n.d. |
| XBmRNA25938\| XB5867546.L | Nucleoplasmin isoform (lacking C-ter tail) | $4.31 \times 10^7$ | n.d. | n.d. | n.d. |
| XBmRNA7871\| uhrf1.S | E3 ubiquitin-protein ligase UHRF1 | $4.27 \times 10^7$ | n.d. | n.d. | n.d. |
| XBmRNA12245\| rpa1.L | Replication protein A | $4.10 \times 10^7$ | n.d. | n.d. | n.d. |
| XBmRNA41384\| LOC100192369 | Sperm-specific nuclear basic protein 1 | $4.08 \times 10^7$ | n.d. | n.d. | n.d. |
| XBmRNA52377\| mcm3.L | DNA replication licensing factor MCM3 | $4.04 \times 10^7$ | n.d. | n.d. | n.d. |
| XBmRNA36255\| mcm5.L | DNA replication licensing factor MCM5 | $3.86 \times 10^7$ | n.d. | n.d. | n.d. |
| XBmRNA36962\| mcm2.L | DNA replication licensing factor MCM2 | $3.73 \times 10^7$ | n.d. | n.d. | n.d. |
| XBmRNA60207\| pold1.L | DNA polymerase | $3.63 \times 10^7$ | n.d. | n.d. | n.d. |
| XBmRNA33152\| fen1.L | Flap endonuclease 1 A | $3.59 \times 10^7$ | n.d. | n.d. | n.d. |
| XBmRNA25831\| pold2.L | DNA polymerase delta subunit 2 | $3.37 \times 10^7$ | n.d. | n.d. | n.d. |
| XBmRNA38234\| ddb1.S | DNA damage-binding protein 1 | $2.98 \times 10^7$ | n.d. | n.d. | n.d. |
| XBmRNA78857\| dppa2.L | Developmental pluripotency associated 2 | $2.96 \times 10^7$ | n.d. | n.d. | n.d. |
| XBmRNA34892\| gins3.L | DNA replication complex GINS protein PSF3 | $2.93 \times 10^7$ | n.d. | n.d. | n.d. |
| XBmRNA83211\| hirip3.S | HIRA-interacting protein 3 | $2.69 \times 10^7$ | n.d. | n.d. | n.d. |
| XBmRNA35380\| nasp.L | histone chaperone NASP | $1.84 \times 10^7$ | n.d. | n.d. | n.d. |
| XBmRNA23497\| hmga2.L | High mobility group AT-hook 2 | $1.45 \times 10^7$ | n.d. | n.d. | n.d. |
| XBmRNA30566\| hmga2.S | High mobility group AT-hook 2 | $1.43 \times 10^7$ | n.d. | n.d. | n.d. |
| XBmRNA33324\| pola2.L | DNA polymerase alpha subunit B | $1.10 \times 10^7$ | n.d. | n.d. | n.d. |
| Streptavidin | MagIC-cryo-EM beads proteins | n.d. | n.d. | $1.22 \times 10^9$ | $2.19 \times 10^9$ |
| SPY-tagGFPnanobody | MagIC-cryo-EM beads proteins | n.d. | n.d. | $7.69 \times 10^8$ | $1.41 \times 10^9$ |
| Spycatcher3 | MagIC-cryo-EM beads proteins | n.d. | n.d. | $5.99 \times 10^8$ | $7.82 \times 10^8$ |
| 11 nm_3HB | MagIC-cryo-EM beads proteins | n.d. | n.d. | $4.32 \times 10^7$ | $5.90 \times 10^7$ |
| 60 nm_SAH | MagIC-cryo-EM beads proteins | n.d. | n.d. | $1.21 \times 10^7$ | $7.73 \times 10^6$ |

low S/N ratio particles were not reproducibly recentered during 2D classification (*Figure 4—figure supplement 2*). On the particles that were successfully recognized during 2D classification, picked points were shifted to the center of the particles (*Figure 4A*, black arrows). However, on the low S/N ratio particles that could not be recognized during 2D classification, picked points were shifted outside the center of the particles (*Figure 4A*, green arrows). To assess the reproducibility of the

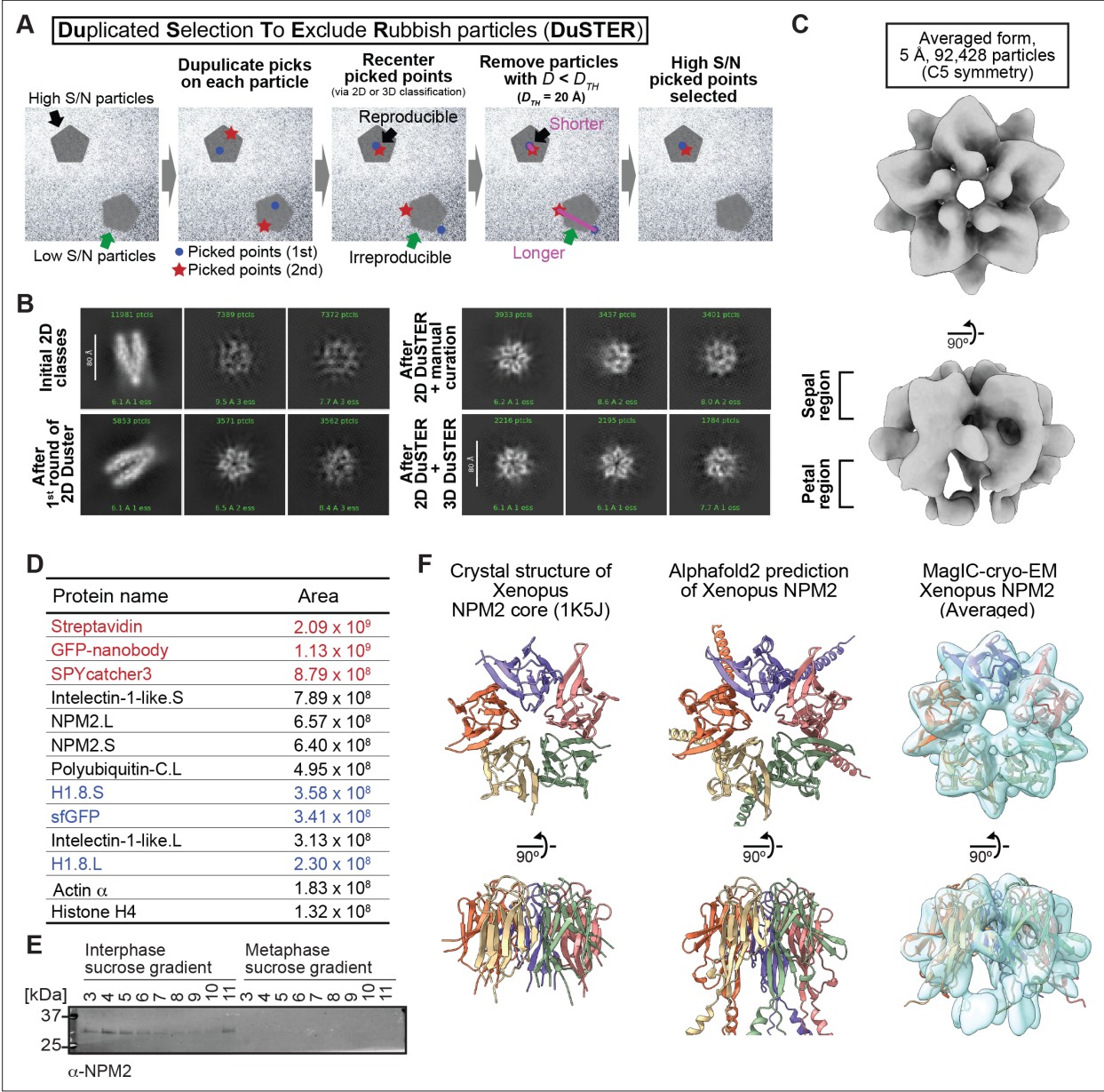

**Figure 4.** MagIC-cryo-EM and DuSTER reconstructed cryo-EM structures of interphase-specific H1.8-bound NPM2. (**A**) Schematic of DuSTER workflow. (**B**) 2D classes before and after particle curation with DuSTER. More 2D classes are shown in *Figure 4—figure supplement 5B–E*. (**C**) 3D cryo-EM structure of interphase-specific H1.8-containing complex. C5 symmetry was applied during structural reconstruction. The complete pipeline is shown in *Figure 4—figure supplement 3*, *Figure 4—figure supplement 5*, and *Figure 4—figure supplement 6*. (**D**) MS identification of proteins that cofractionated with H1.8 in sucrose gradient fraction 4 from interphase chromosomes shown in *Figure 3C*. Portions of MagIC-cryo-EM beads prepared for cryo-EM were subjected to MS. Proteins shown in red are the proteins that comprise the GPF nanobody-MagIC-cryo-EM beads. Proteins shown in blue represent signals from H1.8-GFP. (**E**) Western blot of NPM2 in the sucrose gradient fractions of interphase and metaphase chromosome fragments. (**F**) The structural comparison of the crystal structure of the pentameric NPM2 core (PDB ID: 1K5J), and AF2 predicted structure of the pentameric NPM2 core, and MagIC-cryo-EM structures of NPM2-H1.8. The MagIC-cryo-EM structures indicate NPM2 in the NPM2-H1.8 complex forms pentamer.

The online version of this article includes the following source data and figure supplement(s) for figure 4:

**Source data 1.** The full image of the membrane shown in *Figure 3E*.

**Source data 2.** The raw image of the membrane shown in *Figure 3E*.

**Figure supplement 1.** The fraction with the interphase-specific GFP-H1.8 containing complex had many particles with low S/N by cryo-EM.

**Figure supplement 2.** Reproducible particle centering after 2D classification as a criterion for particles with high S/N.

*Figure 4 continued on next page*

*Figure 4 continued*

**Figure supplement 3.** Pipeline for 2D DuSTER for reconstructing a 3D initial model of the interphase-specific H1.8-containing complex (NPM2-H1.8) that is used as a template of 3D DuSTER.

**Figure supplement 4.** The particles selected by DuSTER curation.

**Figure supplement 5.** Pipeline for 3D DuSTER.

**Figure supplement 6.** Pipeline for 3D structure determination of the interphase-specific H1.8-containing complex (NPM2-H1.8).

**Figure supplement 7.** Cryo-EM maps of NPM2 co-isolated with H1.8.

particle recentering during 2D classification, two independent particle pickings were conducted by Topaz so that each particle on the grid has up to two picked points (*Figure 4A*, second left panel). Some particles that only have one picked point will be removed in a later step. These picked points were independently subjected to 2D classification. After recentering the picked points by 2D classification, distances ($D$) between recentered points from the first picking process and other recentered points from the second picking process were measured. DuSTER keeps recentered points whose $D$ are shorter than a threshold distance ($D_{TH}$). By setting $D_{TH}$ = 20 Å, 2D classification results were dramatically improved in this sample; a five-petal flower-shaped 2D class was reconstructed (*Figure 4B*). This step also removes the particles that only have one picked point. Although approaches to utilize the reproducibility of 2D class assignments have been proposed (*Yang et al., 2012*), the advantage of DuSTER is that it can be applied to small particles that cannot even be properly classified in 2D classification.

Repetitive rounds of particle curation using the picked point locations recentered by 2D classification (referred to as 2D DuSTER) successfully reconstituted 2D classes of 60~80 Å complexes (*Figure 4B*, *Figure 4—figure supplement 3*). As expected, the particles rejected by DuSTER have a generally weak contrast (*Figure 4—figure supplement 4A*). Although higher contrast images can be generated by increasing the defocus (the distance between the target particles and the lens focus), the selected particles were evenly distributed in all defocus ranges between 1.5 and .5 μm (*Figure 4—figure supplement 4B*), demonstrating that DuSTER did not merely select any random high-contrast particles. By selecting these 2D classes, an initial 3D model was built (*Figure 4—figure supplement 3* and *Figure 4—figure supplement 5*). Using this 3D model, particle curation was revised with 3D DuSTER. In the 3D DuSTER, three 3D maps were used as the initial models for the cryoSPARC heterogenous refinement to centering the particles accurately ($D_{TH}$ = 15 Å; *Figure 4—figure supplement 5A*). 3D DuSTER enabled the reconstruction of 3D structure of the interphase-specific H1.8-containing complex, a pentameric macromolecule with a diameter of approximately 60 Å (*Figure 4C*, *Figure 4—figure supplement 7*).

To determine the identity of this complex, MagIC-cryo-EM beads used for isolating the complex were analyzed by MS (*Figure 4D*). Among the proteins detected by MS, NPM2 aligned well with the MagIC-cryo-EM result. Western blotting confirmed that NPM2 was preferentially enriched in interphase chromatin fractions compared to metaphase (*Figure 4E*), while NPM2 interacts with H1.8 in chromosome-free egg extracts both in interphase and metaphase (*Figure 3—figure supplement 1*). The native PAGE of the chromatin fractions indicated that NPM2 forms various complexes, including NPM2-H1.8, on the interphase chromatin fractions (*Figure 3—figure supplement 2D*). In addition, the crystal structure and AlphaFold2 (AF2)-predicted models of *Xenopus* NPM2 matched the MagIC-cryo-EM structure of the interphase-specific H1.8-bound complex (*Figure 4F*) *Dutta et al., 2001*.

## Structural variations of NPM2 bound to H1.8

In *Xenopus* eggs, NPM2 replaces sperm protamines with core histones upon fertilization, thereby promoting nucleosome assembly on sperm DNA *Laskey et al., 1978*; *Philpott et al., 1991*; *Ohsumi and Katagiri, 1991*. NPM2 can also extract out somatic linker histones from chromatin *Dimitrov and Wolffe, 1996*; *Ramos et al., 2005*; *Bañuelos et al., 2007*. X-ray crystallography suggested that recombinant *Xenopus* NPM2 forms a pentamer and a decamer (a dimer of pentamers; *Dutta et al., 2001*). The acidic tracts in the C-terminal tail of NPM2 binds H2A-H2B, histone octamers, and the linker histone H5 *Platonova et al., 2011*; *Ramos et al., 2010*; *Taneva et al., 2009*, while polyglutamylation and hyperphosphorylation of NPM2 promote its substrate sequestration ; *Lorton et al., 2023*. In addition, NPM1 (nucleophosmin), a paralog of NPM2, interacts with H1 *Bañuelos et al.,*

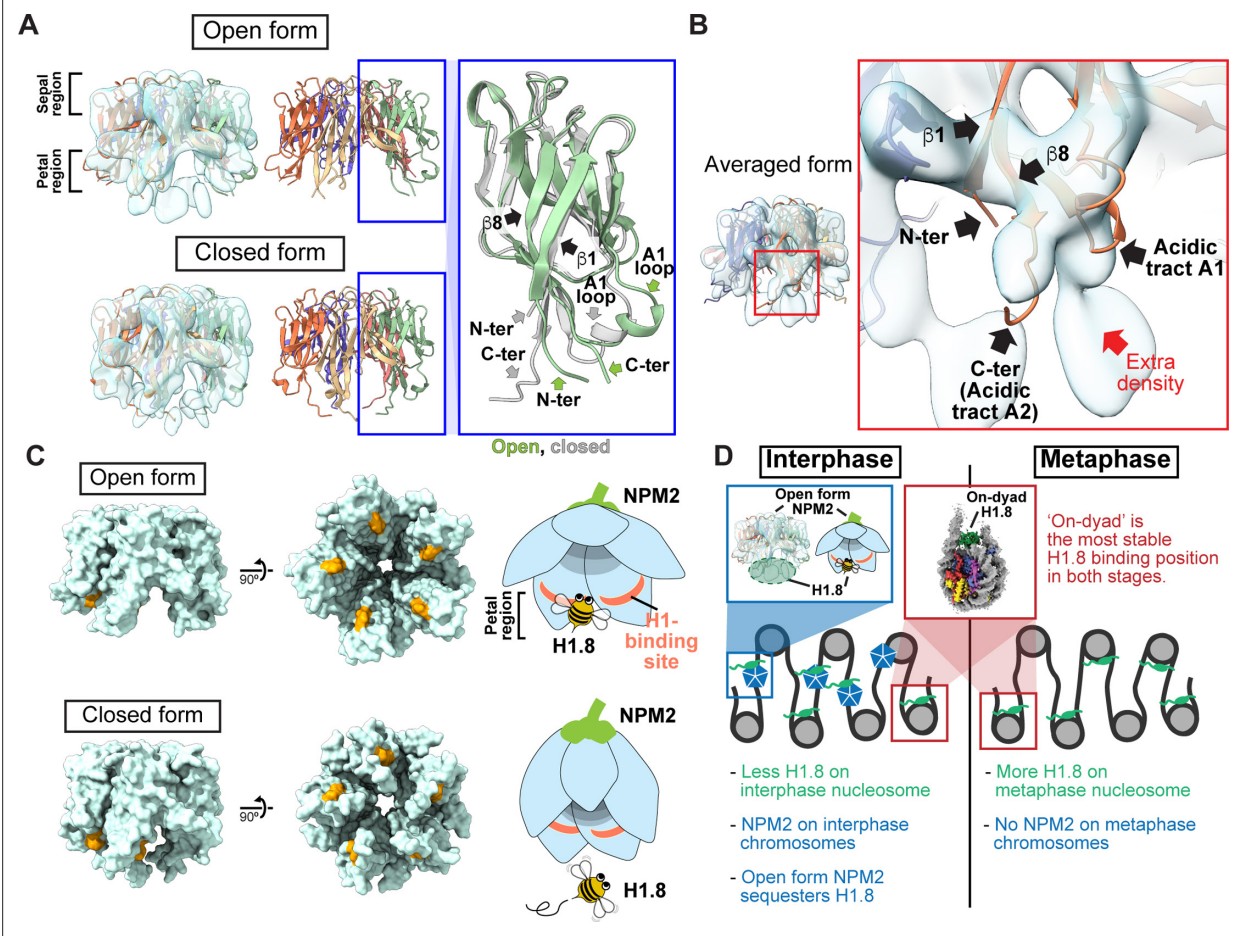

**Figure 5.** Structural variations of NPM2 bound to H1.8. (**A**) Structural differences between the opened and closed forms of NPM2. Left panels show cryo-EM maps of the opened and closed forms of NPM2 with H1.8. Middle panels show the atomic models. The right panel shows the zoomed-in view of the open form (green) and closed form (gray) of the NPM2 protomer. In the closed form, β8 runs straight from the sepal side to the petal side. In the open form, the C-terminal portion of β8 is bent outward to the rim. (**B**) Putative H1.8 density (red arrow) in the averaged NPM2-H1.8 structure. (**C**) The NPM2 surface that contacts the putative H1.8 density (corresponding to aa 42–44) is shown in orange. The surface structures were generated from atomic models. The H1.8-binding sites are accessible in the open form while they are internalized in the closed form. Note that C-terminal acidic tracts A2 and A3 (**Figure 5—figure supplement 1A**) are not visible in the cryo-EM structure but are likely to contribute to H1.8 binding as well in both open and closed forms. (**D**) Model of the mechanism that regulates the amount of the H1.8 in interphase and metaphase nucleosome.

The online version of this article includes the following figure supplement(s) for figure 5:

**Figure supplement 1.** Cryo-EM maps and atomic models of NPM2 co-isolated with H1.8.

**Figure supplement 2.** Asymmetric structures of NPM2 co-isolated with H1.8 without applying C5 symmetry.

*2007*; *Gadad et al., 2011*. However, no subnanometer-resolution structure of NPM2 or NPM1 with post-translational modifications or with substrates is currently available.

By further analyzing our cryo-EM structure representing the H1.8-bound state of NPM2, we identified two structural variants, classified as open and closed forms (**Figure 5**, **Figure 4—figure supplement 6**, and **Figure 4—figure supplement 7J–K**). Due to its structural similarity to a flower, we call the highly acidic putative substrate-binding surface the petal side, whereas the other more charge neutral surface the sepal side (**Figure 5**, **Figure 5—figure supplement 1**). The major structural differences between the two forms are found at C-terminal and N-terminal segments of NPM2 core and at the A1 loop (**Figures 5 and 6**, **Figure 5—figure supplement 1**). In the closed form, β8 runs straight from the sepal to the petal sides of each pentamer and has an extended C-terminal segment that protrudes past the petal side of the pentamer. In the open form, however, the C-terminal portion of β8 is bent outward to the rim (**Figure 5A**). Along with this β8 bending, C-terminal segment, N-terminal segment, and A1 loop are also positioned outward in the open form. The configuration of β1, β8, and

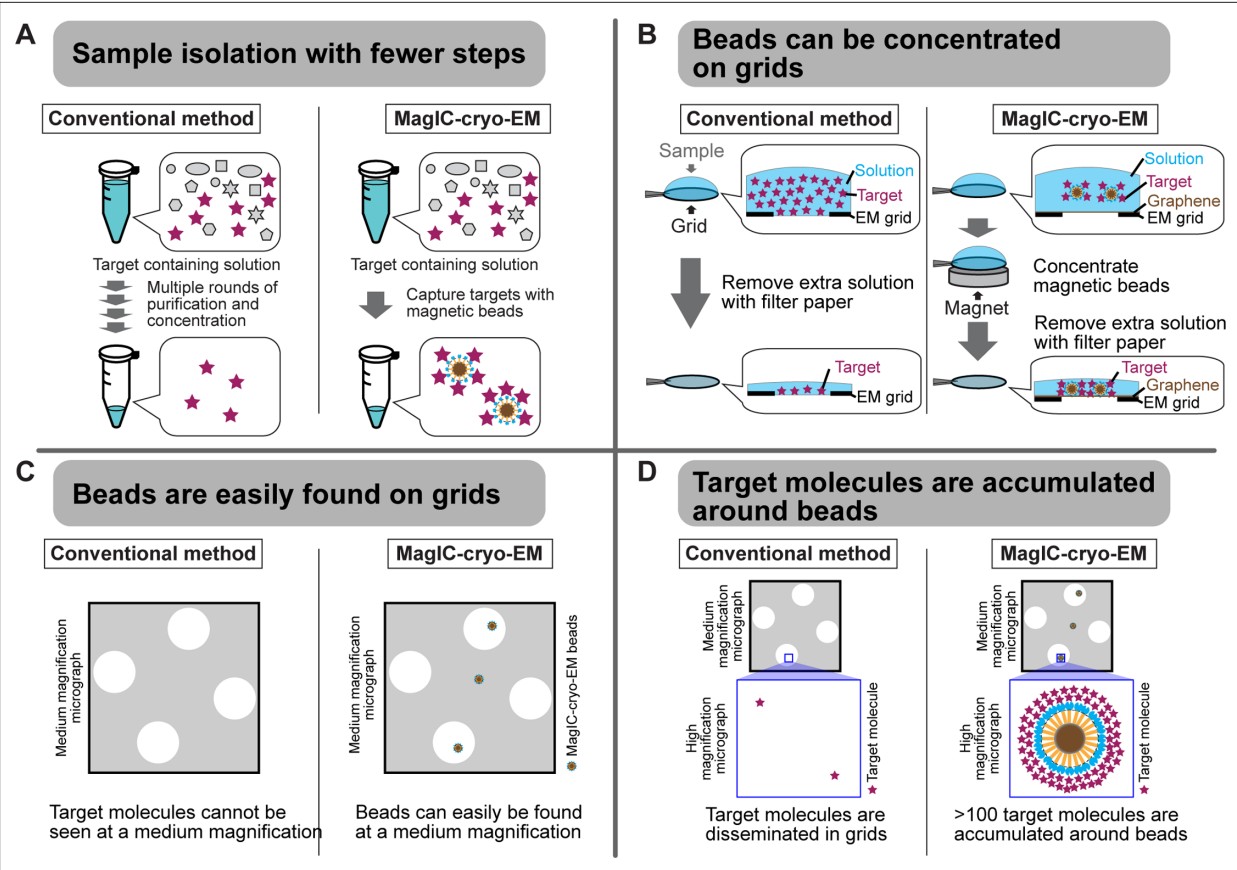

**Figure 6.** Advantages of MagIC-cryo-EM over conventional cryo-EM methods. (**A**) The on-bead-cryo-EM approach reduces preparation steps (e.g. target isolation, enrichment, and buffer exchange), which can lead to sample loss. (**B**) Sample loss during the grid-freezing process is reduced by magnet-based enrichment of the targets on cryo-EM grids. (**C**) The magnetic beads are easily identified in medium-magnification montage maps, enabling the selection of areas where targets exist prior to high-magnification data collection. (**D**) Targets are highly concentrated around the beads, ensuring that each micrograph contains more than 100 usable particles for 3D structure determination.

A1 loop in the crystal structure of *Xenopus* NPM2 *Dutta et al., 2001*, the AF2-predicted structure of *Xenopus* NPM2 *Mirdita et al., 2022*; *Jumper et al., 2021*; *Steinegger and Söding, 2017*, and the cryo-EM structure of the bacterially expressed human NPM1 *Saluri et al., 2023*, which were all determined in the absence of their target proteins, is similar to the closed form (*Figure 5—figure supplement 1B–D*). Notably, extra cryo-EM densities, which may represent H1.8, are clearly observed in the open form but much less in the closed form near the acidic surface regions proximal to the N terminus of β1 and the C terminus of β8 (*Figure 5A and B*). Supporting this idea, the acidic tract A1 (aa 36–40) and A2 (aa 120–140), which are both implicated in the recognition of basic substrates such as core histones *Dutta et al., 2001*; *Platonova et al., 2011*, respectively interact with and are adjacent to the putative H1.8 density (*Figure 5B*). In addition, the NPM2 surface that is in direct contact with the putative H1.8 density is accessible in the open form while it is internalized in the closed form (*Figure 5C*). This structural change of NPM2 may support more rigid binding of H1.8 to the open NPM2, whereas H1.8 binding to the closed form is less stable and likely occurs through interactions with the C-terminal A2 and A3 tracts, which are not visible in our cryo-EM structures.

In the aforementioned NPM2-H1.8 structures, for which we applied C5 symmetry during the 3D structure reconstruction, only a partial H1.8 density could be seen (*Figure 5B*). One possibility is that the H1.8 structure in NPM2-H1.8 does not follow C5 symmetry. As the size of the NPM2-H1.8 complex estimated from sucrose gradient elution volume is consistent with pentameric NPM2 binding to a single H1.8 (*Figure 3C* and *Table 3*), applying C5 symmetry during structural reconstruction likely blurred the density of the monomeric H1.8 that binds to the NPM2 pentamer. The structural determination of NPM2-H1.8 without applying C5 symmetry lowered the overall resolution but visualized

**Table 3.** Expected mass of the NPM2-H1.8-GFP complex.
Sucrose gradient elution volume indicates that the NPM2-H1.8-GFP complex is smaller than mononucleosome (around 230 kDa). Only the NMP2 pentamer complexed H1.8-GFP monomer (166 kDa) reasonably explains the sucrose gradient result.

| Name | Molecular wight (Da) |
|---|---|
| NPM2 monomer | 21,917 |
| NPM2 pentamer | 109,587 |
| NPM2 decamer | 219,175 |
| H1.8-GFP | 56,704 |
| NPM2 pentamer +H1.8 GFP monomer | 166,291 |
| NPM2 pentamer +H1.8 GFP pentamer | 393,107 |
| NPM2 decamer +H1.8 GFP monomer | 275,878 |
| NPM2 decamer +H1.8 GFP pentamer | 502,694 |
| Nucleosome (193 bp DNA) | 228,111 |

multiple structural variants of the NPM2 protomer with different degrees of openness co-existing within an NPM2-H1.8 complex (*Figure 5—figure supplement 2*), raising a possibility that opening of a portion of the NPM2 pentamer may affect modes of H1.8 binding. Although more detailed structural analyses of the NPM2-substrate complex are the subject of future studies, MagIC-cryo-EM and DuSTER revealed structural changes of NPM2 that was co-isolated H1.8 on interphase chromosomes.

## Discussion

MagIC-cryo-EM offers sub-nanometer resolution structural determination using a heterogeneous sample that contains the target molecule at 1~2 nM, which is approximately 100–1000 times lower than the concentration required for conventional cryo-EM methods, including affinity grid approach (*Kelly et al., 2008*; *Llaguno et al., 2014*; *Yu et al., 2016*). This significant improvement was achieved through the four unique benefits of MagIC-cryo-EM (*Figure 6*). First, the on-bead-cryo-EM approach minimizes preparation steps, which can lead to sample loss, such as target isolation, enrichment, and buffer exchange (*Figure 6A*). Second, sample loss during the grid-freezing process is reduced by magnet-based enrichment of the targets on cryo-EM grids (*Figures 2E–I and 6B*). Third, magnetic beads are easily identifiable on the grid (*Figures 2G and 6C*). Fourth, the target molecules are accumulated around magnetic beads, ensuring that each micrograph contains more than 100 usable particles independent of input sample concentration (*Figures 2J and 6D*). Adapting the ChIP-based method to MagIC cryo-EM, we successfully isolated and reconstructed the H1.8-bound nucleosome and the H1.8-bound NPM2 structures from interphase chromosomes, which have never been accomplished before.

To reconstitute the structure of H1.8-bound NPM2, we needed to devise the particle curation method DuSTER, which greatly helped the structural reconstitution of small particles with low S/N (*Figure 4*). By combining MagIC-cryo-EM and DuSTER, we were able to determine the sub-nanometer structure and structural variations of the NPM2-H1.8-GFP complex, in which the mass of the ordered region is only 60 kDa. Notably, particle curation by DuSTER does not require human supervision or machine learning, except for determining the distance threshold between repeatedly picked particles. This feature may allow for automating particle curation via DuSTER in the future.

MagIC-cryo-EM and DuSTER approaches hold the potential for targeting a wide range of biomolecules, including small ones, for two main reasons. First, the target-capturing module could be replaced with various other proteins, such as different nanobodies, single-chain variable fragments (scFv), protein A, dCas9, or avidin, to capture a wide range of biomolecules. Second, the sample requirement for MagIC-cryo-EM is a mere 5 ng per grid, which is comparable to or even lower than the requirements of widely used ChIP-seq *Zou et al., 2022*. Coupling next-generation sequencing with MagIC-cryo-EM beads would help the field determine structural features of functionally distinct

chromatin regions, such as heterochromatin, euchromatin, transcription start sites, telomeres, and centromeres. The low sample requirement of MagIC-cryo-EM also opens the door to structural analysis using limited specimens, including patient tissues.

Combining MS, MagIC-cryo-EM and DuSTER, we found that the majority of chromatin-bound H1.8 in interphase existed as a complex with NPM2 rather than with nucleosomes (*Figure 5C and D*). This contrasts to the reports suggesting that NAP1 is the major H1.8-bound chaperone in *Xenopus* egg extracts *Shintomi et al., 2005*; *Freedman et al., 2010*, while it is consistent with our previous MS analysis that also detected NPM2, but not NAP1, in fractions enriched with nucleosomes in interphase *Arimura et al., 2021*. Our observation is also in line with a previous report that NPM2 is able to remove linker histones but not core histones from somatic nuclei that are introduced to *Xenopus* egg extracts *Dimitrov and Wolffe, 1996*. Since the amounts of H1.8-associated NAP1 or NPM2 in the egg cytoplasm did not change between interphase and metaphase (*Figure 3—figure supplement 1*), a mechanism must exist such that NPM2 interacts with H1.8 on chromatin specifically in interphase and suppresses H1.8-nucleosome interaction (*Figure 5D*). Two basic patches at the C-terminal tail of NPM2 may contribute to cell-cycle-dependent DNA binding as they are flanked with potential Cdk1 phosphorylation sites. NPM2 may maintain nucleosome-bound H1.8 at a low level in interphase during early developmental cell cycles to support rapid DNA replication, while mitotic induction of H1.8 association with nucleosomes tunes condensin loading on chromosomes and ensures proper chromosome size to facilitate chromosome segregation (*Choppakatla et al., 2021*; *Figure 5D*).

Structural studies based on in vitro reconstitution previously suggested that NPM2 binds to its substrate as a homo-decamer *Dutta et al., 2001*; *Platonova et al., 2011*, or a homo-pentamer *Ramos et al., 2010*; *Taneva et al., 2009*. Our cryo-EM structure strongly suggests that the NPM2 binds to H1.8 as a homo-pentamer. Structure variation analyses suggest that NPM2 subunits can exhibit two structural configurations, open and closed forms, of which H1.8 is stably associated with only the open form. Since the closed form is more similar to the reported crystal structure and AF2-predicted structures (*Figure 5—figure supplement 1B–D*), both of which are determined in the absence of the substrates, our analysis points toward a possibility that substrate binding induces the structural transition of NPM2 to the open form. The conformational changes of the NPM family have been proposed in other studies, such as NMR and negative stain-EM *Lorton et al., 2023*; *Anson et al., 2015*; *González-Arzola et al., 2022*. Our cryo-EM structures of NPM2 indicate the potential mechanisms of NPM2 conformational changes and potential substrate binding sites. Among NPM2 acidic tracts A1, A2, and A3, which are important for substrate recognition, our atomic models visualize A1 and the edge of A2 at the petal side of the structure, where the density corresponding to the predicted H1.8 can be found (*Figure 5B*). As the A2 and A3 belong to the disordered C-terminal tail that extends from the petal side of the NPM2 complex, our data suggest that the open form provides a stable association platform by exposing the acidic surface at the petal side for the substrate recognition, while the C-terminal A2 and A3 at the flexible tail may facilitate recruitment and possibly also entrapment of the substrate. Since our structural analysis further suggests that each NPM2 subunit may independently adapt open and closed form within a pentamer, this flexibility in the core domain may enable the association of substrates with diverse sizes and structures to support its molecular chaperone functionality.

## Limitations of the study

While MagIC-cryo-EM is envisioned as a versatile approach suitable for various biomolecules from diverse sources, including cultured cells and tissues, it has thus far been tested only with H1.8-bound nucleosome and H1.8-bound NPM2, both using anti-GFP nanobodies to isolate GFP-tagged H1.8 from chromosomes assembled in *Xenopus* egg extracts after pre-fractionation of chromatin. To apply MagIC-cryo-EM for the other targets, the following factors must be considered: *(1) Pre-fractionation*. This step (e.g. density gradient or gel filtration) may be necessary to enrich the target protein in a specific complex from other diverse forms (such as monomeric forms, subcomplexes, and protein aggregates). *(2) Avoiding bead aggregation*. Beads may be clustered by targets (if the target complex contains multiple affinity tags or is aggregated), nonspecific binders, and target capture modules. To directly apply antibodies that recognize the native targets and specific modifications, optimization to avoid bead aggregation will be important. *(3) Stabilizing complexes*. The target complexes must be stable during the sample preparation. Crosslink was necessary for the H1.8-GFP-bound nucleosome.

*(4) Loading the optimum number of targets on the bead.* The optimal number of particles per bead differs depending on target sizes, as larger targets are more likely to overlap. For H1.8-GFP-bound nucleosomes, 500–2000 particles per bead were optimal. We expect that fewer particles should be coated for larger targets.

Regarding the cryo-EM data acquisition, the selection of data collection points is currently performed through the manual picking of magnetic beads on the MMM map. This method does not support image-shift-based data collection and serves as a bottleneck for data collection speed, limiting throughput to approximately 500–1000 micrographs per day. The development of machine learning-based software to automatically identify magnetic beads on MMM maps and establish parameters for image-shift-based multiple shots could substantially enhance data collection efficiency.

The efficiency of magnetic bead capture can be further improved. In the current MagIC-cryo-EM workflow, the cryo-EM grid is incubated on a magnet before being transferred to the Vitrobot for vitrification. However, since the Vitrobot cannot accommodate a strong magnet, the vitrification step occurs without the magnetic force, potentially resulting in bead loss. This limitation could be addressed by developing a new plunge freezer capable of maintaining magnetic force during vitrification.

While DuSTER enables the structural analysis of NPM2 co-isolated with H1.8-GFP, the resulting map quality is modest, and the reported numerical resolution may be overestimated. Furthermore, only partial density for H1.8 is observed. Although structural analysis of small proteins is inherently challenging, it is possible that halo-like scattering further hinders high-resolution structural determination by reducing the S/N ratio. More detailed structural analyses of the NPM2-substrate complex will be addressed in future studies.

## Methods

### *Xenopus laevis*

*Xenopus laevis* was purchased from *Xenopus* 1 (female, 4270; male, 4235). Vertebrate animal protocols (20031 and 23020) approved by the Rockefeller University Institutional Animal Care and Use Committee were followed.

### Purification of Biotin-3HB-SPYcatcher003

Biotin-3HB-SPYcatcher003 was bacterially expressed and purified using pQE80-His$_{14}$-bdSUMO-Cys-3HB-SPYcatcher003. To build the plasmid, a pQE80 derivative vector encoding an N-terminal His-tag was amplified by PCR from pSF1389 [Addgene plasmid # 104962] *Frey and Görlich, 2014*. gBlock DNAs encoding *Brachypodium distachyon* SUMO (bdSUMO) (*Frey and Görlich, 2014*) and a computationally designed monomeric three-helix bundle *Huang et al., 2014* were synthesized by IDT and used as a PCR template. DNA encoding SPYcatcher003 was amplified using pSpyCatcher003 [Addgene plasmid # 133447] *Keeble et al., 2019* as a PCR template. DNA fragments were assembled by the Gibson assembly method (*Gibson et al., 2009*). *E. coli* Rosetta (DE3) cells expressing His$_{14}$-bdSUMO-Cys-3HB-SPYcatcher003 were induced with 1 mM isopropyl-β-D-thiogalactopyranoside (IPTG) at 25 °C and then resuspended in 100 mL buffer A (8 mM Na$_2$HPO$_4$, 2 mM KH$_2$PO$_4$, 537 mM NaCl, 2.7 mM KCl, 10% glycerol, 2 mM β-mercaptoethanol, 1 mM PMSF, 20 mM imidazole with 1 x cOmplete Protease Inhibitor Cocktail EDTA-free [Roche]). The cells were disrupted by sonication, and the soluble fraction was collected by centrifugation at 20,000 rpm (46,502 rcf) at 4 °C for 30 min using a 45Ti rotor in Optima L80 (Beckman Coulter). This fraction was then mixed with Ni-NTA agarose beads (QIAGEN). Protein-bound Ni-NTA agarose beads were packed into an Econo-column (Bio-Rad) and washed with 170 column volumes (CV) of buffer B (8 mM Na$_2$HPO$_4$, 2 mM KH$_2$PO$_4$, 937 mM NaCl, 2.7 mM KCl, 10% glycerol, 2 mM β-mercaptoethanol, 1 mM PMSF, 40 mM imidazole with 1 x cOmplete EDTA-free Protease Inhibitor Cocktail [Roche], pH 7.4). The beads were further washed with 33 CV of Phosphate-Buffered Saline (PBS: 8 mM Na$_2$HPO$_4$, 2 mM KH$_2$PO$_4$, 137 mM NaCl, 2.7 mM KCl, pH 7.4) containing additional 5% glycerol to remove β-mercaptoethanol. The His$_{14}$-SUMO-tag was cleaved by incubating overnight at 4 °C with N-terminal His-tagged SENP1 protease, which was expressed and purified using the previously described method with pSF1389 [Addgene plasmid # 104962] *Frey and Görlich, 2014*. Ni-NTA agarose beads that bound the cleaved His$_{14}$-bdSUMO-tag and His$_{14}$-SENP1 were filtered out using an Econo-column (Bio-Rad). The cleaved 3HB-SPYcatcher003 with a cysteine residue at the N-terminal was concentrated using Amicon 30 K (Millipore), mixed with

EZ-link Maleimide-PEG2-Biotin (Thermo A39261), and left at 4 °C overnight. Biotinylated 3HB-SPY-catcher003 was dialyzed overnight against PBS at 4 °C. The dialyzed Biotin-3HB-SPYcatcher003 was further purified through a Hi-load Superdex75 16/600 column (Cytiva) and stored at –20 °C in PBS containing 47.5% glycerol.

## Purification of Biotin-60 nm-SAH-SPYcatcher003 and Biotin-90 nm-SAH-SPYcatcher003

Biotin-30 nm-SAH-SPYcatcher003 and Biotin-60 nm-SAH-SPYcatcher003 were bacterially expressed and purified using pQE80-His$_{14}$-bdSUMO-Cys-30nm-SAH-SPYcatcher003 and pQE80-His$_{14}$-bdSUMO-Cys-60 nm-SAH-SPYcatcher003. DNA encoding 30 nm SAH from *Trichomonas vaginalis* was amplified using pCDNA-FRT-FAK30 [Addgene plasmid # 59121] *Sivaramakrishnan and Spudich, 2011* as a PCR template. To extend the repeat to the desired length, MluI and AscI sites were inserted at the top and bottom of the DNA segment encoding 30 nm SAH, respectively. Although the target sequences for AscI (GG/CGCGCC) and MluI (A/CGCGT) are distinct, the DNA overhangs formed after the DNA digestion are identical. In addition, the DNA sequence formed by ligating these DNA overhangs translated into Lys-Ala-Arg, which does not disrupt an SAH. To generate pQE80-His$_{14}$-bdSUMO-Cys-60 nm-SAH-SPYcatcher3, two DNA fragments were prepared. The longer fragment was prepared by digesting pQE80-His$_{14}$-bdSUMO-Cys-30 nm-SAH-SPYcatcher003 with XhoI and MluI. The shorter fragment was prepared by digesting pQE80-His$_{14}$-bdSUMO-Cys-30 nm-SAH-SPYcatcher003 with XhoI and AscI. Target fragments were isolated by agarose gel extraction and ligated to form pQE80-His$_{14}$-bdSUMO-Cys-60nm-SAH-SPYcatcher003. Repeating these steps, pQE80-His$_{14}$-bdSUMO-Cys-90 nm-SAH-SPYcatcher003 was also generated.

*E. coli* Rosetta (DE3) cells expressing His$_{14}$-bdSUMO-Cys-SAH-SPYcatcher003 were induced with 1 mM IPTG at 18 °C and then resuspended in 100 mL of buffer A before being disrupted by sonication. The soluble fraction was collected by centrifugation at 20,000 rpm (46,502 rcf) at 4 °C for 30 min using a 45Ti rotor in Optima L80 (Beckman Coulter) and applied to a HisTrap HP column (Cytiva). The column was washed with 4 column volumes (CV) of buffer B. His$_{14}$-bdSUMO-Cys-SAH-SPYcatcher003 was eluted from the HisTrap column with buffer D (8 mM Na$_2$HPO$_4$, 2 mM KH$_2$PO$_4$, 137 mM NaCl, 2.7 mM KCl, 5% glycerol, 200 mM imidazole [pH 7.4]). The eluted His$_{14}$-bdSUMO-Cys-SAH-SPYcatcher003 was mixed with His$_{14}$-SENP1 and dialyzed against PBS containing 5% glycerol at 4 °C overnight. The dialyzed protein was applied to the HisTrap HP column (Cytiva) to remove the cleaved His$_{14}$-bdSUMO-tag and His$_{14}$-SENP1. The cleaved SAH-SPYcatcher003 was further purified through a MonoQ 5/50 column (Cytiva). The purified SAH-SPYcatcher003 with a cysteine residue at the N-terminus was concentrated with Amicon 10 K (Millipore), mixed with EZ-link Maleimide-PEG2-Biotin (Thermo A39261), and placed overnight at 4 °C. The biotinylated SAH-SPYcatcher003 was dialyzed against PBS at 4 °C overnight. The dialyzed Biotin-SAH-SPYcatcher003 was purified through a Hi-load Superdex200 16/600 column (Cytiva) and stored at –20 °C in PBS containing 47.5% glycerol.

## Purification of mono-SPYtag-avidin tetramer

Mono-SPYtag-avidin tetramer was purified using a modified version of the method described by *Howarth et al., 2006*. pET21-SPY-His$_6$-tag streptavidin and pET21-streptavidin were generated by using pET21a-Streptavidin-Alive [Addgene plasmid # 20860] *Howarth et al., 2006* as a PCR template. SPY-His$_6$-tag streptavidin and untagged avidin were expressed individually in *E. coli* BL21(DE3) as inclusion bodies by inducing with 1 mM IPTG at 37 °C. The cells expressing the proteins were resuspended in 100 mL of buffer E (50 mM Tris-HCl, 1 mM EDTA) and disrupted by sonication. Insoluble fractions were collected by centrifugation at 20,000 rpm at 4 °C for 30 min using a 45Ti rotor in Optima L80 (Beckman Coulter). The insoluble pellets were washed by resuspending them in 50 ml of buffer E and re-collecting them through centrifugation at 20,000 rpm at 4 °C for 30 min using a 45Ti rotor in Optima L80 (Beckman Coulter). The washed insoluble pellets were resuspended in 8 mL of 6 M guanidine HCl (pH 1.5) and dialyzed against 200 mL of 6 M guanidine HCl (pH 1.5) overnight at 4 °C. The denatured proteins were collected by centrifugation at 20,000 rpm at 4 °C for 30 min using a 45Ti rotor in Optima L80 (Beckman Coulter). Protein concentrations in soluble fractions were estimated based on the absorbance at 260 nm. Denatured SPY-His$_6$-tag streptavidin and untagged streptavidin were mixed at a 1:2.9 molar ratio and rapidly refolded by diluting them with 250 mL of PBS at 4 °C. After 6 hr of stirring at 4 °C, aggregated proteins were removed by centrifugation at

20,000 rpm at 4 °C for 30 min using a 45Ti rotor in Optima L80 (Beckman Coulter). The supernatant was mixed with 62.7 g of solid ammonium sulfate and stirred overnight at 4 °C. Insolubilized proteins were removed with centrifugation at 20,000 rpm at 4 °C for 30 min using a 45Ti rotor in Optima L80 (Beckman Coulter). The supernatant was loaded into the HisTrap HP column (Cytiva). Refolded avidin tetramers were eluted from the column by a linear gradient of imidazole (10 mM to 500 mM) in PBS. The peak corresponding to mono-SPY-His-tagged streptavidin tetramer was collected and concentrated using Amicon 10 K (Millipore). The concentrated mono-SPY-His$_6$-tagged streptavidin tetramer was further purified through Hiload superdex75 (Cytiva) and stored at –20 °C in PBS containing 47.5% glycerol.

## Purification of SPYtag-GFP nanobody

MagIC-cryo-EM beads were optimized by testing three different GFP nanobodies: tandem GFP nanobody, GFP enhancer nanobody, and LaG (llama antibody against GFP)–10 (*Figure 2—figure supplement 1*). To express SPYtag-GFP nanobodies, plasmids pSPY-GFP nanobody were built. The plasmid has a pQE80 backbone, and the DNA sequences that encode His$_{14}$-bdSUMO-SPYtag-GFP nanobody were inserted into the multiple cloning sites of the backbone. DNA encoding tandem GFP nanobody was amplified from pN8his-GFPenhancer-GGGGS4-LaG16 [Addgene plasmid # 140442] *Zhang et al., 2020*. DNA encoding GFP enhancer nanobody *Kirchhofer et al., 2010* was amplified from pN8his-GFPenhancer-GGGGS4-LaG16. DNA encoding the LaG10 nanobody was amplified from a plasmid provided by Dr. Michael Rout (*Fridy et al., 2014*). GFP nanobodies were expressed at 16 °C in *E. coli* Rosetta (DE3) by IPTG induction. The cells expressing His$_{14}$-bdSUMO-SPYtag-GFP nanobody were resuspended with 100 mL buffer A and disrupted by sonication. The soluble fraction was collected with centrifugation at 20,000 rpm (46,502 rcf) at 4 °C for 30 min using a 45Ti rotor in Optima L80 (Beckman Coulter) and applied to the HisTrap HP column (Cytiva). The protein was eluted from the column with a step gradient of imidazole (50, 200, 400 mM) in buffer F (50 mM Tris-HCl (pH 8), 100 mM NaCl, 800 mM Imidazole, 5% Glycerol). The eluted His$_{14}$-bdSUMO-SPYtag-GFP nanobody was mixed with His$_{14}$-SENP1 and dialyzed against PBS containing 5% glycerol at 4 °C overnight. The dialyzed protein was applied to the HisTrap HP column (Cytiva) to remove the cleaved His$_{14}$-bdSUMO-tag and His$_{14}$-SENP1. The cleaved SPYtag-GFP-nanobody was concentrated with Amicon 10 K (Millipore). The concentrated SPYtag-singular GFP nanobody was further purified through Hiload superdex75 (Cytiva) and stored at –20 °C in PBS containing 47.5% glycerol.

## Purification of H1.8-GFP

To purify *Xenopus laevis* H1.8-superfolder GFP (sfGFP, hereafter GFP), pQE80-His$_{14}$-bdSUMO-H1.8-GFP was generated by replacing bdSENP1 in pSF1389 vector to H1.8-GFP. Using this plasmid, His$_{14}$-bdSUMO-H1.8-GFP was expressed in *E. coli* Rosetta (DE3) at 18 °C with 1 mM IPTG induction. The soluble fraction was collected through centrifugation at 20,000 rpm (46,502 rcf) at 4 °C for 30 min using a 45Ti rotor in Optima L80 (Beckman Coulter) and applied to the HisTrap HP column (Cytiva). His$_{14}$-bdSUMO-H1.8-GFP was eluted from the column with a linear gradient of imidazole (100 mM to 800 mM) in PBS. The fractions containing His$_{14}$-bdSUMO-H1.8-GFP were collected, mixed with SENP1 protease, and dialyzed overnight against PBS containing 5% glycerol at 4 °C. The SENP1-treated sample was then applied to a Heparin HP column (Cytiva) and eluted with a linear gradient of NaCl (137 mM to 937 mM) in PBS containing 5% glycerol. The fractions containing H1.8-GFP were collected and concentrated using Amicon 30 K (Millipore) before being applied to a Hiload Superdex200 16/600 column (Cytiva) in PBS containing 5% glycerol. The fractions containing H1.8-GFP were collected, concentrated using Amicon 30 K (Millipore), flash-frozen, and stored at –80 °C.

## Purification of MNase

To purify MNase, pK19-His-bdSUMO-MNase was generated. Using this plasmid, His14-bdSUMO-MNase was expressed in *E. coli* JM101 at 18 °C with 2 mM IPTG induction. The soluble fraction was collected through centrifugation at 20,000 rpm (46,502 rcf) at 4 °C for 30 min using a 45Ti rotor in Optima L80 (Beckman Coulter) and applied to the HisTrap HP column (Cytiva). His14-bdSUMO-MNase was eluted from the column with a linear gradient of imidazole (100 mM to 500 mM) in PBS. The fractions containing His14-bdSUMO-MNase were collected, mixed with SENP1 protease, and dialyzed overnight against PBS containing 5% glycerol at 4 °C. The dialyzed protein was applied to the

HisTrap HP column (Cytiva) to remove the cleaved His14-bdSUMO-tag and His14-SENP1. The cleaved MNase was concentrated with Amicon 3 K (Millipore). The concentrated MNase was further purified through Hiload superdex75 (Cytiva) and stored at –80 °C in PBS containing 60% glycerol.

## Purification of *X. laevis* histones

All histones were purified using the method described previously (*Zierhut et al., 2014*). Bacterially expressed *X. laevis* H2A, H2B, H3.2, and H4 were purified from inclusion bodies. His-tagged histones (H2A, H3.2, and H4) or untagged H2B expressed in bacteria were resolubilized from the inclusion bodies by incubation with 6 M guanidine HCl. For His-tagged histones, the solubilized His-tagged histones were purified using Ni-NTA beads (QIAGEN). For untagged H2B, the resolubilized histones were purified using a MonoS column (Cytiva) under denaturing conditions before H2A-H2B dimer formation. To reconstitute the H3–H4 tetramer and H2A–H2B dimer, the denatured histones were mixed at an equal molar ratio and dialyzed to refold the histones by removing the guanidine. His-tags were removed by TEV protease treatment, and the H3–H4 tetramer and H2A–H2B dimer were isolated through a HiLoad 16/600 Superdex 75 column (Cytiva). The fractions containing the H3–H4 tetramer and H2A–H2B dimer were concentrated using Amicon 10 K, flash-frozen, and stored at –80 °C.

## Preparation of in vitro reconstituted poly-nucleosome

pAS696 containing the 19-mer of the 200 bp 601 nucleosome positioning sequence was digested using HaeII, DraI, EcoRI, and XbaI. Both ends of the 19-mer of the 200 bp 601 DNA were labeled with biotin by Klenow fragment (NEB) with biotin-14-dATP (*Guse et al., 2012*). The nucleosomes were assembled with the salt dialysis method (*Guse et al., 2012*). Purified DNAs were mixed with H3-H4 and H2A-H2B, transferred into a dialysis cassette, and placed into a high-salt buffer (10 mM Tris-HCl [pH 7.5], 1 mM EDTA, 2 M NaCl, 5 mM β-mercaptoethanol, and 0.01% Triton X-100). Using a peristaltic pump, the high-salt buffer was gradually exchanged with a low-salt buffer (10 mM Tris-HCl [pH 7.5], 1 mM EDTA, 50 mM NaCl, 5 mM β-mercaptoethanol, 0.01% Triton X-100) at roughly 2 mL/min overnight at 4 °C. In preparation for cryo-EM image collection, the dialysis cassette containing the sample was then placed in a buffer containing 10 mM HEPES-HCl (pH 7.4) and 30 mM KCl and dialyzed for 48 hr at 4 °C.

## Native PAGE and SDS-PAGE

For the native PAGE for nucleosome (*Figure 3C*), 15 µL of nucleosome fractions were loaded onto a 0.5 x TBE 6% native PAGE gel. For the native PAGE for nucleosomal DNA (*Figure 3—figure supplement 2B*), 15 µL of nucleosome fractions were mixed with 1 µL of 10 mg/mL RNaseA (Thermo Fisher Scientific) and incubated at 55 °C for 30 min. To deproteinize and reverse-crosslink DNA, RNaseA-treated samples were then mixed with 1 µL of 19 mg/mL Proteinase K solution (Roche) and incubated at 55 °C for overnight. Samples were loaded to 0.5 x TBE 6% native PAGE. Native PAGE gels were stained by SYTO-60 to detect DNA. SYTO-60 and GFP signals were scanned on a LI-COR Odyssey. For SDS-PAGE analysis (*Figure 3—figure supplement 2B*), 20 µL of nucleosome fractions were mixed with 5 µL of 4 x SDS-PAGE sample buffer (200 mM Tris- HCl pH 6.8, 8% SDS, 40% glycerol, 10% β-mercaptoethanol) and boiled for 10 min at 96 °C. Samples were loaded to a 4–20% gradient gel (Bio-Rad, # 5671095).

## Western blot

For the western blot of nucleosome fractions (*Figure 3D*), 20 µL of nucleosome fractions were mixed with 5 µL of 4 x SDS-PAGE sample buffer and boiled for 10 min at 96 °C. Samples were loaded to a 4–20% gradient gel (Bio-Rad, # 5671095).

For the H1.8-GFP complementation assay (*Figure 3—figure supplement 1*), 2 µL egg extract samples were added to 38 µL of 1 x SDS-PAGE sample buffer (50 mM Tris- HCl pH 6.8, 2% SDS, 10% glycerol, 2.5% β-mercaptoethanol) and boiled for 5 min at 96 °C. Samples were mixed by vortex and spun at 13,200 rpm for 1 min before gel electrophoresis. 10 µL out of 40 µL samples were separated in 4–20% gradient gel (Bio-Rad, # 5671095).

The SDS-PAGE gels were transferred into the western blot cassette and transferred to a nitrocellulose membrane (Cytiva, # 10600000) with 15 V at 4 °C overnight. The transferred membranes were

blocked with Intercept TBS Blocking Buffer (LI-COR Biosciences, # 927–60001). Primary and secondary antibodies were diluted in Intercept TBS Antibody Diluent (LI-COR Biosciences, #927–65001). For *Figure 3—figure supplement 1A*, as primary antibodies, mouse monoclonal antibody against GFP (Santa Cruz Biotechnology, # sc-9996, 1:1000 dilution) and rabbit polyclonal antibody against *X. laevis* H1.8 (*Jenness et al., 2018*; final: 1 µg/mL) were used. For *Figure 3—figure supplement 1*, as primary antibodies, rabbit polyclonal antibody against *X. laevis* H1.8 (*Jenness et al., 2018*), rabbit polyclonal antisera against *X. laevis* NAP1 1:500 dilution (*Lorton et al., 2023*), NPM2 (1:500 dilution; *Anson et al., 2015*), and rabbit polyclonal antibody against phosphorylated histone H3 Thr3 (MilliporeSigma, # 07–424, 1:5000 dilution) were used. NAP1 and NPM2 antibody are kind gifts of David Shechter. As secondary antibodies, IRDye 800CW goat anti-rabbit IgG (LI-COR, # 926–32211; 1:10,000) and IRDye 680RD goat anti-mouse IgG (LI-COR, # 926–68070; 1:15,000) were used. The images were taken with Odyssey M Infrared Imaging System (LI-COR Biosciences).

## Immunoprecipitation (IP) assay in *Xenopus* egg extract

For the IP assay (*Figure 3—figure supplement 1*), antibody against rabbit IgG, in-house purified from pre-immune rabbit serum by HiTrap Protein A HP (# 17040301), and antibody against *X. laevis* H1.8 (# RU2130) were conjugated to Protein-A coupled Dynabeads (Thermo Fisher Scientific, # 10001D) at 20 µg/mL beads at 4 °C for overnight on a rotator. rIgG and H1.8 antibody beads were crosslinked using 5 mM BS3 (Thermo Fisher Scientific, # A39266) resuspended in PBS (pH 7.4) at room temperature for 30 min and quenched by 50 mM Tris-HCl (pH 7.4) resuspended in PBS (pH 7.4) at room temperature for 20–30 min on a rotator. All antibody beads were washed extensi vely using wash/coupling buffer (10 mM K-HEPES (pH 8.0) and 150 mM KCl), followed by sperm dilution buffer (10 mM K-HEPES (pH 8.0), 1 mM MgCl$_2$, 100 mM KCl, 150 mM sucrose). The beads were left on ice until use.

Interphase egg extract (30 µL) was prepared by incubating at 20 °C for 60 min after adding CaCl$_2$ (final: 0.4 mM) and cycloheximide (final: 100 µg/mL) to fresh CSF egg extract. Mitotic egg extract (CSF egg extract, 30 µL) was also incubated at 20 °C for 60 min without any additives. After 60 min incubation, each mitotic and interphase egg extract was transferred to antibody-conjugated beads (10 µL) after removing sperm dilution buffer on a magnet stand (Sergi Lab Supplies, Cat# 1005). Beads-extract mixtures were mixed and incubated on ice for 45 min with flicking tubes every 15 min. After 45 min, beads were collected using a magnet stand at 4 °C and washed three times with beads wash buffer (sperm dilution buffer supplemented 1 x cOmplete EDTA-free protease inhibitor cocktail (Roche, # 4693132001), 1 x PhosSTOP (Roche, # 4906845001), and 0.1% (v/v) Triton-X (Bio-Rad, # 1610407)). Beads are resuspended in 20 µL of 1 x SDS sample buffer and loaded 10 µL out of 20 µL to a SDS-PAGE gel. Methods for SDS-PAGE and western blot are described above.

## Trial MagIC-cryo-EM with poly-nucleosome (used in Figure 1)

A total of 60 fmol of Absolute Mag streptavidin nano-magnetic beads (CD bioparticles: WHM-X047, 50 nM size) were mixed with 100 µL of EM buffer A (10 mM HEPES-KOH [pH 7.4], 30 mM KCl, 1 mM EGTA, 0.3 ng/µL leupeptin, 0.3 ng/µL pepstatin, 0.3 ng/µL chymostatin, 1 mM Sodium Butyrate, 1 mM beta-glycerophosphate, 1 mM MgCl$_2$, 2% trehalose, 0.2% 1,6-hexanediol). The beads were collected by incubation on two pieces of 40x20 mm N52 neodymium disc magnets (DIYMAG: D40x20–2P-NEW) at 4 °C for 30 min and then resuspended in 120 µL of EM buffer A. The two pieces of strong neodymium magnets have to be handled carefully as magnets can leap and slam together from several feet apart. Next, 60 µL of 34 nM nucleosome arrays formed on the biotinylated 19-mer 200 bp 601 DNA were mixed with the beads and rotated at 20 °C for 2 hr. To remove unbound nucleosomes, the biotin-poly-nucleosome-bound nano-magnetic beads were collected after 40 min of incubation on the N52 neodymium disc magnets and then resuspended in 300 µL EM buffer containing 10 µM biotin. A 100 µL portion of the biotin-poly-nucleosome-bound nano-magnetic beads solution was incubated on the N52 neodymium disc magnets for 30 min and then resuspended in 20 µL EM buffer A. Finally, 3 µL of biotin-poly-nucleosome-bound nano-magnetic beads solution was added onto a glow-discharged Quantifoil Gold R 1.2/1.3 300 mesh grid (Quantifoil). The samples were vitrified under 100% humidity, with a 20 s incubation and 5 s blotting time using the Vitrobot Mark IV (FEI).

The grid was imaged on a Talos Arctica (Thermo Fisher) equipped with a 200 kV field emission gun and K2 camera. A total of 657 movies were collected at a magnification of ×72,000 (1.5 Å/pixel) using super-resolution mode, as managed by SerialEM (*Mastronarde, 2003*). Movie frames are

motion-corrected and dose-weighted patch motion correction in CryoSPARC v3 with output Fourier cropping fac½ 1/2 (*Punjani et al., 2017*). Particles were picked by Topaz v0.2.3 with around 2000 manually picked nucleosome-like particles as training models (*Bepler et al., 2019a*). Picked particles were extracted using CryoSPARC v3 (extraction box size = 200 pixel). 2D classification of extracted particles was done using 100 classes in CryoSPARC v3. Using 2D classification results, particles were split into the nucleosome-like groups and the non-nucleosome-like groups. Four 3D initial models were generated for both groups with ab initio reconstruction in CryoSPARC v3 (Class similarity = 0). One nucleosome-like model was selected and used as a given model of heterogeneous reconstruction with all four of the 'decoy' classes generated from the non-nucleosome-like group. After the first round of 3D classification, the particles assigned to the 'decoy' classes were removed, and the remaining particles used for a second round of 3D classification using the same settings as the first round. These steps were repeated until more than 90% of particles wer classified as a nucleosome-like class. To isolate the nucleosome class that has visible H1.8 density, four to six 3D references were generated with ab initio reconstruction of CryoSPARC v3 using purified nucleosome-like particles (Class similarity = 0.9). Refined particles were further purified with the heterogeneous refinement using an H1.8-visible class and an H1.8-invisible class as decoys. The classes with reasonable extra density were selected and refined with homogeneous refinement. The final resolution was determined with the gold stand FSC threshold (FSC = 0.143).

## Preparation of in vitro reconstituted mono-nucleosome and H1.8-GFP-bound mono-nucleosome

The 193 bp 601 DNA fragment was amplified by a PCR reaction (*Arimura et al., 2012*; *Lowary and Widom, 1998*). The nucleosomes were assembled with the salt dialysis method described above. The reconstituted nucleosome was dialyzed into buffer XL (80 mM PIPES-KOH [pH 6.8], 15 mM NaCl, 60 mM KCl, 30% glycerol, 1 mM EGTA, 1 mM MgCl$_2$, 10 mM β-glycerophosphate, 10 mM sodium butyrate). H1.8-GFP was mixed with nucleosome with a 1.25 molar ratio in the presence of 0.001% poly L-glutamic acid (wt 3000–15,000; Sigma-Aldrich) and incubated at 37 °C for 30 min. As a control nucleosome sample without H1.8-GFP, the sample without H1.8-GFP was also prepared. The samples were then crosslinked adding a 0.5-time volume of buffer XL containing 3% formaldehyde and incubating for 90 min on ice. The crosslink reaction was quenched by adding 1.7 volume of quench buffer (30 mM HEPES-KOH (pH 7.4), 150 mM KCl, 1 mM EGTA, 10 ng/µL leupeptin, 10 ng/µL pepstatin, 10 ng/µL chymostatin, 10 mM sodium butyrate, 10 mM β-glycerophosphate, 400 mM glycine, 1 mM MgCl$_2$, 5 mM DTT). The quenched sample was layered onto the 10–25% linear sucrose gradient solution with buffer SG (15 mM HEPES-KOH [pH 7.4], 50 mM KCl, 10–22% sucrose, 10 µg/mL leupeptin, 10 µg/mL pepstatin, 10 µg/mL chymostatin, 10 mM sodium butyrate, 10 mM β-glycerophosphate, 1 mM EGTA, 20 mM glycine) and spun at 32,000 rpm (max 124,436 rcf) and 4 °C for 13 hr using SW55Ti rotor in Optima L80 (Beckman Coulter). The centrifuged samples were fractionated from the top of the sucrose gradient. The concertation of H1.8-GFP-bound nucleosome in each fraction is calculated based on the 260 nm light absorbance detected by Nanodrop 2000 (Thermo Fisher Scientific).

## Preparation of GFP nanobody attached MagIC-cryo-EM beads

A total of 25 fmol of Absolute Mag streptavidin nanomagnetic beads (CD Bioparticles: WHM-X047) were transferred to a 0.5 mL protein LoBind tube (Eppendorf) and mixed with 200 pmol of inner spacer module protein (biotin-3HB-SPYcatcher003 or biotin-60nm-SAH-SPYcatcher003) in 200 µL of EM buffer A (10 mM HEPES-KOH [pH 7.4], 30 mM KCl, 1 mM EGTA, 10 ng/µL leupeptin, 10 ng/µL pepstatin, 10 ng/µL chymostatin, 1 mM Sodium Butyrate, and 1 mM beta-glycerophosphate) and the mixture was incubated at 4 °C for 10 hr. To wash the beads, the mixture was spun at 13,894 rpm (16,000 rcf) at 4 °C for 10 min using the SX241.5 rotor in an Allegron X-30R centrifuge (Beckman Coulter). The beads that accumulated at the bottom of the tube were resuspended in 200 µL of EM buffer A. Subsequently, 200 pmol of mono-SPYtag-avidin tetramer was added to the beads in 200 µL of EM buffer A, and the mixture was incubated at 4 °C for 10 hr. Again, the beads were washed by collecting them via centrifugation and resuspending them in 200 µL of EM buffer A. This washing step was repeated once more, and 800 pmol of outer spacer module protein (biotin-30 nm-SAH-SPYcatcher003, biotin-60 nm-SAH-SPYcatcher003 or biotin-90 nm-SAH-SPYcatcher003) were added and incubated at 4 °C for 10 hr. The beads were washed twice and resuspended with 25 µL of EM buffer A. 20 µL of this mixture

was transferred to a 0.5 ml protein LoBind tube and mixed with 640 pmol of SPYtag-GFP nanobody and incubated at 4 °C for 10 hr. The beads were washed twice and resuspended with 25 µL of EM buffer A. The assembled GFP nanobody attached MagIC-cryo-EM beads can be stored in EM buffer A containing 50% glycerol at –20 °C for several weeks.

## Graphene grids preparation

Graphene grids were prepared using the method established by *Han et al., 2020* with minor modifications. Briefly, monolayer graphene grown on the copper foil (Grolltex) was coated by polymethyl methacrylate (Micro chem, EL6) with the spin coat method. The copper foil was removed by 1 M of ammonium persulfate. The graphene monolayer coated by polymethyl methacrylate was attached to gold or copper grids with carbon support film (Quantifoil) and baked for 30 min at 130 °C. The polymethyl methacrylate was removed by washing with 2-butanone, water, and 2-propanol on a hotplate.

## Optimization of the spacer module length by the MagIC-cryo-EM of in vitro reconstituted H1.8-GFP bound nucleosome

To prepare the MagIC-cryo-EM beads capturing H1.8-GFP bound mono-nucleosome, 4 fmol of GFP nanobody-attached MagIC-cryo-EM beads with different spacer lengths were mixed with 100 nM (28 ng/µL) of in vitro reconstituted crosslinked H1.8-GFP-bound mono-nucleosome in 100 µL of PBS containing 15~30% glycerol and incubated at 4 °C for 12 hr. To wash the beads, the beads were collected with centrifugation at 13,894 rpm (16,000 rcf) at 4 °C for 20 min using SX241.5 rotor in Allegron X-30R (Beckman Coulter) and resuspended with 200 µL of PBS containing 15~30% glycerol. This washing step was repeated once again, and the beads were resuspended with 100 µL of EM buffer C (10 mM HEPES-KOH [pH 7.4], 30 mM KCl, 1 mM EGTA, 10 ng/µL leupeptin, 10 ng/µL pepstatin, 10 ng/µL chymostatin, 1 mM sodium butyrate, 1 mM β-glycerophosphate, 1.2% trehalose, and 0.12% 1,6-hexanediol). This washing step was repeated once again, and the beads were resuspended with 100~200 µL of EM buffer C (theoretical beads concentration: 20~40 pM).

To vitrify the grids, a plasma-cleaned graphene-coated Quantifoil gold R1.2/1.3 400 mesh grid (Quantifoil) featuring a monolayer graphene coating *Han et al., 2020* was held using a pair of sharp non-magnetic tweezers (SubAngstrom, RVT-X). The two pieces of strong neodymium magnets have to be handled carefully as magnets can leap and slam together from several feet apart. Subsequently, 4 µL of MagIC-cryo-EM beads capturing H1.8-GFP-nucleosomes were applied to the grid. The grid was then incubated on the 40×20 mm N52 neodymium disc magnets for 5 min within an in-house high-humidity chamber to facilitate magnetic bead capture. Once the capture was complete, the tweezers anchoring the grid were transferred and attached to the Vitrobot Mark IV (FEI), and the grid was vitrified by employing a 2 s blotting time at room temperature under conditions of 100% humidity.

We found that gold grids are suitable for MagIC-cryo-EM, whereas copper grids worsened the final resolution of the structures presumably due to magnetization of the copper grids during the concentration process which then interfered with the electron beam and caused the grid to vibrate during data collection (*Figure 2—figure supplement 1*, Test 7).

The vitrified grids were loaded onto the Titan Krios (Thermo Fisher), equipped with a 300 kV field emission gun and a K3 direct electron detector (Gatan). A total of 1890 movies were collected at a magnification of ×64,000 (1.33 Å/pixel) using super-resolution mode, as managed by SerialEM (*Mastronarde, 2003*).

Movie frames were corrected for motion using MotionCor2 (*Zheng et al., 2017*) installed in Relion v4 (*Scheres, 2012*) or patch motion correction implemented in CryoSPARC v4. Particles were picked with Topaz v0.2 (*Bepler et al., 2019a*), using approximately 2000 manually picked nucleosome-like particles as training models. The picked particles were then extracted using CryoSPARC v4 (extraction box size = 256 pixels) (*Punjani et al., 2017*). Nucleosome-containing particles were isolated through decoy classification using heterogeneous reconstruction with one nucleosome-like model and four decoy classes generated through ab initio reconstruction in CryoSPARC v4. CTF refinement and Bayesian polishing were applied to the nucleosome-containing particles in the Relion v4 (*Scheres, 2012*; *Zivanov et al., 2019*). To isolate the nucleosome class with visible H1.8 density, four 3D references were generated through ab initio reconstruction in CryoSPARC v4 using purified nucleosome-like particles (Class similarity = 0.9). These four 3D references were used for heterogeneous reconstruction. Two of the classes had strong H1.8 density. Using the particles assigned in these

classes, non-uniform refinement was performed in CryoSPARC v4. The final resolution was determined using the gold standard FSC threshold (FSC = 0.143).

## MagIC-cryo-EM of in vitro reconstituted H1.8-GFP-bound nucleosome using the mixture of the H1.8-GFP-bound and unbound nucleosomes (shown in Figure 2)

A total of 0.5 fmol of GFP-singular nanobodies conjugated to 3HB-60nm-SAH magnetic beads were mixed with 1.7 nM (0.5 ng/µL) of H1.8-GFP-bound nucleosome and 53 nM (12 ng/µL) of H1.8-free nucleosome in 100 µL of buffer SG (15 mM HEPES-KOH [pH 7.4], 50 mM KCl, 12% sucrose, 1 x LPC, 10 mM Sodium Butyrate, 10 mM β-glycerophosphate, 1 mM EGTA) containing approximately 17% sucrose. The mixture was then incubated at 4 °C for 10 hr. To wash the beads, they were collected by centrifugation at 13,894 rpm (16,000 rcf) at 4 °C for 20 min using the SX241.5 rotor in an Allegron X-30R centrifuge (Beckman Coulter). Subsequently, the beads were resuspended in 200 µL of EM buffer C. This washing step was repeated twice, and the beads were finally resuspended in approximately 80 µL of EM buffer C, resulting in a theoretical bead concentration of 6.25 pM.

To vitrify the grids, 4 µL of the samples were applied to plasma-cleaned graphene-coated Quantifoil gold R1.2/1.3 300-mesh grids (Quantifoil). The grid was then incubated on the 40x20 mm N52 neodymium disc magnets for 5 min and vitrified using the Vitrobot Mark IV (FEI) with a 2 s blotting time at room temperature under 100% humidity. The vitrified grids were loaded onto the Titan Krios (Thermo Fisher), equipped with a 300 kV field emission gun and a K3 direct electron detector (Gatan). A total of 1890 movies were collected at a magnification of x 64,000 (1.33 Å/pixel) using super-resolution mode, as managed by SerialEM (*Mastronarde, 2003*).

The analysis pipeline is described in *Figure 2—figure supplement 2*. Movie frames were corrected for motion using MotionCor2 (*Zheng et al., 2017*), which was installed in Relion v4 (*Scheres, 2012*). Particles were picked with Topaz v0.2.3 (*Bepler et al., 2019a*), using approximately 2000 manually picked nucleosome-like particles as training models. The picked particles were then extracted using CryoSPARC v4 (extraction box size = 256 pixels; *Punjani et al., 2017*). Nucleosome-containing particles were isolated through decoy classification using heterogeneous reconstruction with one nucleosome-like model and four decoy classes generated through ab initio reconstruction in CryoSPARC v3.3. CTF refinement and Bayesian polishing were applied to the nucleosome-containing particles in Relion v4 (*Scheres, 2012*; *Zivanov et al., 2019*). To isolate the nucleosome class with visible H1.8 density, four 3D references were generated through ab initio reconstruction in CryoSPARC v3.3 using purified nucleosome-like particles (Class similarity = 0.9). These four 3D references were used for heterogeneous reconstruction. Two of the classes had strong H1.8 density. Using the particles assigned in these classes, non-uniform refinement was performed in CryoSPARC v3.3. The final resolution was determined using the gold standard FSC threshold (FSC = 0.143).

## Assessment of the efficiency of the magnetic concentration of the MagIC-cryo-EM on cryo-EM grid (shown in Figure 2)

A plasma-cleaned graphene-coated Quantifoil copper R1.2/1.3 400 mesh grid (Quantifoil) was held using non-magnetic Vitrobot tweezers (SubAngstrom). Subsequently, 4 µL of 12.5 pM GFP-nanobody attached MagIC-cryo-EM beads were applied to the grid. The grid was then incubated on the 40x20 mm N52 neodymium disc magnets for 5 min within a high-humidity chamber. As a control experiment, several grids were frozen by omitting the magnetic incubation steps. Once the capture was complete, the tweezers anchoring the grid were attached to the Vitrobot Mark IV (FEI), and the grid was vitrified by employing a 2 s blotting time at room temperature under conditions of 100% humidity. The vitrified grids were subjected to cryo-EM to collect 8×8 or 9×9 montage maps at ×2600 magnification on Talos Arctica to capture the whole area of each square mesh. The efficiency of the magnetic concentration of the MagIC-cryo-EM beads was quantitatively assessed by counting the percentage of holes containing MagIC-cryo-EM beads and counting the average number of MagIC-cryo-EM beads per hole. For the quantification, 11 square meshes with 470 holes were used for the condition without magnetic concentration. For the condition with 5 min incubation on magnets, 11 square meshes with 508 holes were used. The boxplots and the scatter plots were calculated by the seaborn.boxplot and seaborn.stripplot tools in the Seaborn package (*Waskom, 2021*) and visualized

by Matplotlib (*Hunter, 2007*). Outlier data points that are not in 1.5 times of the interquartile range, the range between the 25th and 75th percentile, were excluded.

## Functional assessment of H1.8-GFP in *Xenopus* egg extract

The functional replaceability of H1.8-GFP in *Xenopus* egg extracts was assessed through whether H1.8-GFP could rescue the chromosome morphological defect caused by depletion of endogenous H1.8. Mitotic chromosome morphology and length were assessed through the previously described method (23) with some modifications.

The cytostatic factor (CSF)-arrested metaphase *Xenopus laevis* egg extracts were prepared using the method as described *Murray, 1991*). Anti-rabbit IgG (SIGMA, Cat# I5006) and rabbit anti-H1.8 custom antibodies (*Choppakatla et al., 2021* (Identification# RU2130) were conjugated to Protein-A coupled Dynabeads (Thermo Fisher Scientific, # 10001D) at 250 µg/mL beads at 4 °C for overnight on a rotator. IgG and H1.8 antibody beads were crosslinked using 4 mM BS3 (Thermo Fisher Scientific, # A39266) resuspended in PBS (pH 7.4) at room temperature for 45 min and quenched by 50 mM Tris-HCl (pH 7.4) resuspended in PBS (pH 7.4) at room temperature for 20–30 min on a rotator. All antibody beads were washed extensively using wash/coupling buffer (10 mM K-HEPES (pH 8.0) and 150 mM KCl), followed by sperm dilution buffer (10 mM K-HEPES (pH 8.0), 1 mM $MgCl_2$, 100 mM KCl, 150 mM sucrose). After the two rounds of depletion at 4 °C for 45 min using 2 volumes of antibody-coupled beads on a rotator, the beads were separated using a magnet (Sergi Lab Supplies, Cat# 1005). For the complementation of H1.8, 1.5 µM of recombinantly purified H1.8 or H1.8-GFP was supplemented into H1.8-depleted CSF egg extract.

To assess chromosome morphology in the metaphase chromosomes with spindles, 0.4 mM $CaCl_2$ was added to CSF-arrested egg extracts containing *X. laevis* sperm (final concentration 2000 /µL) to cycle the extracts into interphase at 20 °C for 90 min. To induce mitotic entry, half the volume of fresh CSF extract and 40 nM of the non-degradable cyclin BΔ90 fragment were added after 90 min and incubated at 20 °C for 60 min.

Metaphase spindles for fluorescent imaging were collected by a published method (*Desai et al., 1998*). 15 µL metaphase extracts containing mitotic chromosomes were diluted into 2 mL of fixing buffer (80 mM K-PIPES pH 6.8, 1 mM $MgCl_2$, 1 mM EGTA, 30% (v/v) glycerol, 0.1% (v/v) Triton X-100, 2% (v/v) formaldehyde) and incubated at room temperature for 5 min. The fixed samples were layered onto a cushion buffer (80 mM K-PIPES pH 6.8, 1 mM $MgCl_2$, 1 mM EGTA, 50% (v/v) glycerol) with a coverslip (Fisher Scientific, Cat# 12CIR-1.5) placed at the bottom of the tube and centrifuged at 5000 x *g* for 15 min at 16 °C in a swinging bucket rotor (Beckman Coulter, JS-5.3 or JS-7.5). The coverslips were recovered and fixed with pre-chilled methanol (–20 °C) for 5 min. The coverslips were extensively washed with TBST (TBS supplemented 0.05% Tween-20) and then blocked with antibody dilution buffer (AbDil; 50 mM Tris-HCl pH 7.5, 150 mM NaCl, 2% BSA, 0.02% $NaN_3$) at 4 °C for overnight.

Individualized mitotic chromosome samples were prepared as described previously (*Choppakatla et al., 2021*). 10 µL of metaphase extracts containing mitotic chromosomes were diluted into 60 µL of chromosome dilution buffer (10 mM K-HEPES pH 8, 200 mM KCl, 0.5 mM EGTA, 0.5 mM $MgCl_2$, 250 mM sucrose), mixed by gentle flicking, and incubated at room temperature for 8 min. Diluted samples were transferred into 3 mL of fixing buffer (80 mM K-PIPES pH 6.8, 1 mM $MgCl_2$, 1 mM EGTA, 30% (v/v) glycerol, 0.1% (v/v) Triton X-100, 2% (v/v) formaldehyde), mixed by inverting tubes, and incubated for total 6 min at room temperature. Similar to mitotic chromosome preparation, the fixed samples were subjected to glycerol cushion centrifugation (7000 × *g* for 20 min at 16 °C) using a swinging bucket rotor (Beckman, JS-7.5). Coverslips were recovered, fixed with pre-chilled methanol (–20 °C) for 5 min, extensively washed with TBST, and then blocked with AbDil buffer at 4 °C overnight.

For immunofluorescence microscopy, primary and secondary antibodies were diluted in AbDil buffer. Coverslips were incubated in primary antibody solution at room temperature for 60 min and secondary antibody at room temperature for 45 min. DNA was stained using NucBlue Fixed Cell ReadyProbes Reagent (Thermo Fisher Scientific, Cat# R37606) following manufacture's protocol. Coverslips were extensively washed using TBST between each incubation and sealed on the slide glass using ProLong Diamond Antifade Mountant (Thermo Fisher Scientific, Cat# P36965). For primary antibodies, mouse monoclonal antibody against α-tubulin (MilliporeSigma, Cat# T9026, 1:1000 dilution) and rabbit polyclonal antibody against *X. laevis* CENP-A (*Wynne and Funabiki, 2015*; Identification#

RU1286), 1:1000 dilution. For secondary antibodies, mouse IgG was detected using Cy3 AffiniPure F(ab')₂ Fragment Donkey Anti-Mouse IgG (H+L; Jackson ImmunoResearch, Cat# 715-166-150; 1:500 dilution) and rabbit IgG was detected using Cy5 AffiniPure Donkey Anti-Rabbit IgG (H+L; Jackson ImmunoResearch, Cat# 711-175-152; 1:500 dilution).

The immunofluorescence imaging was performed on a DeltaVision Image Restoration microscope (Applied Precision), which is a widefield inverted microscope equipped with a pco. edge sCMOS camera (pco). Immunofluorescence samples were imaged with 1 µm z-sections using a 60×Olympus UPlan XApo (1.42 NA) oil objective, and were processed with a iterative processive deconvolution algorithm using the Soft-WoRx (Applied Precision).

For chromosome length measurements, the length of individualized mitotic chromosomes were manually traced on a single maximum intensity slice using segmented line tool in Fiji software (ver. 2.9.0). Data was summarized using R (ver. 4.2.2) and visualized as SuperPlots *Lord et al., 2020* using ggplot2 package in R and RStudio (ver. RSTUDIO-2023.09.1–494). For the representative images in *Figure 3—figure supplement 1*, max projection images were prepared in Fiji using z-stuck function. For the visibility, the brightness and contrast of representative images were adjusted using GIMP software (ver. 4.2.2). Adjustment was done using a same setting among all images.

## Fractionation of chromosomes isolated from *Xenopus* egg extracts Figure 3

Nucleosomes were isolated from *Xenopus* egg extract chromosomes using the previously described method (*Arimura et al., 2021*). To prevent the spontaneous cycling of egg extracts, 0.1 mg/mL cycloheximide was added to the CSF extract. H1.8-GFP was added to the CSF extract at a final concentration of 650 nM, equivalent to the concentration of endogenous H1.8 (*Wühr et al., 2014*). For interphase chromosome preparation, *Xenopus laevis* sperm nuclei (final concentration 2000 /µL) were added to 5 mL of CSF extracts, which were then incubated for 90 min at 20 °C after adding 0.3 mM CaCl₂ to release the CSF extracts into interphase. For metaphase sperm chromosome preparation, cyclin B Δ90 (final concentration 24 µg/mL) and 1 mL of fresh CSF extract were added to 2 mL of the extract containing interphase sperm nuclei prepared using the method described above. To make up for the reduced H1.8-nucleosome formation in interphase, we used 5 mL of egg extracts for preparing interphase chromosomes and 2 mL of extracts for metaphase chromosomes. The extracts were incubated for 60 min at 20 °C, with gentle mixing every 10 min. To crosslink the *Xenopus* egg extracts chromosomes, nine times the volume of ice-cold buffer XL (80 mM PIPES-KOH [pH 6.8], 15 mM NaCl, 60 mM KCl, 30% glycerol, 1 mM EGTA, 1 mM MgCl₂, 10 mM β-glycerophosphate, 10 mM sodium butyrate, 2.67% formaldehyde, 0.001% digitonin) was added to the interphase or metaphase extract containing chromosomes, which was further incubated for 60 min on ice. These fixed chromosomes were then layered on 3 mL of fresh buffer SC (80 mM HEPES-KOH [pH 7.4], 15 mM NaCl, 60 mM KCl, 1.17 M sucrose, 50 mM glycine, 0.15 mM spermidine, 0.5 mM spermine, 1.25 x cOmplete EDTA-free Protease Inhibitor Cocktail (Roche), 10 mM beta-glycerophosphate, 10 mM sodium butyrate, 1 mM EGTA, 1 mM MgCl₂) in 50 mL centrifuge tubes (Falcon, #352070). The tubes were spun at 3300 (2,647 rcf) rpm at 4 °C for 40 min using a JS 5.3 rotor in an Avanti J-26S centrifuge (Beckman Coulter). Pellets containing fixed chromosomes were resuspended with 10 mL of buffer SC, layered on 3 mL of fresh buffer SC in 14 mL centrifuge tubes (Falcon, #352059), and spun at 3300 (2647 rcf) rpm at 4 °C for 40 min using a JS 5.3 rotor in an Avanti J-26S centrifuge (Beckman Coulter). The chromosomes were collected from the bottom of the centrifuge tube and resuspended with buffer SC. Chromosomes were pelleted by centrifugation at 5492 rpm (2500 rcf) using an SX241.5 rotor in an Allegron X-30R centrifuge (Beckman Coulter). The chromosome pellets were resuspended with 200 µL of buffer SC. To digest chromatin, MNase concentration and reaction time were tested on a small scale and optimized to the condition that produce 180–200 bp DNA fragments. After the optimization, 0.6 and 0.3 U/µL of MNase were added to interphase and metaphase chromosomes, respectively. Then, CaCl₂ was added to a final concentration of 7.4 mM, and the mixture was incubated at 4 °C for 4 hr. The MNase reaction was stopped by adding 100 µL MNase stop buffer B (80 mM PIPES-KOH (pH 6.8), 15 mM NaCl, 60 mM KCl, 30% glycerol, 20 mM EGTA, 1 mM MgCl₂, 10 mM β-glycerophosphate, 10 mM sodium butyrate, 3.00% formaldehyde). The mixtures were incubated on ice for 1 hr and then diluted with 700 µL of quench buffer (30 mM HEPES-KOH (pH 7.4), 150 mM KCl, 1 mM EGTA 1 x LPC, 10 mM sodium butyrate, 10 mM β-glycerophosphate,

400 mM glycine, 1 mM MgCl$_2$, 5 mM DTT). The soluble fractions released by MNase were isolated by taking supernatants after centrifugation at 13,894 rpm (16,000 rcf) at 4 °C for 30 min using an SX241.5 rotor in an Allegron X-30R centrifuge (Beckman Coulter). The supernatants were collected and layered onto a 10–22% linear sucrose gradient solution with buffer SG (15 mM HEPES-KOH [pH 7.4], 50 mM KCl, 10–22% sucrose, 10 µg/mL leupeptin, 10 µg/mL pepstatin, 10 µg/mL chymostatin, 10 mM sodium butyrate, 10 mM β-glycerophosphate, 1 mM EGTA, 20 mM glycine) and spun at 32,000 rpm (max 124,436 rcf) and 4 °C for 13 hr using an SW55Ti rotor in an Optima L80 centrifuge (Beckman Coulter). The samples were fractionated from the top of the sucrose gradient. The concentration of H1.8 in each fraction was determined by western blot. 15 µL of each sucrose gradient fraction was incubated at 95 °C with 1% sodium dodecyl sulfate (SDS) and applied for SDS-PAGE with a 4–20% gradient SDS-PAGE gel (Bio-Rad). The proteins were transferred to a nitrocellulose membrane (Cytiva) from the SDS-PAGE gel using TE42 Tank Blotting Units (Hoefer) at 15 V, 4 °C for 4 hr. As primary antibodies, 1 µg/mL of mouse monoclonal Anti-GFP Antibody sc-9996 (Santa Cruz Biotechnology) and as secondary antibodies, IR Dye 800CW goat anti-mouse IgG (Li-Cor 926–32210; 1:15,000) were used. The images were taken with an Odyssey Infrared Imaging System (Li-Cor). The existence of the H1.8-GFP-bound nucleosomes was confirmed by native PAGE. 15 µL of each sucrose gradient fraction was applied for a 6 % x0.5 TEB native PAGE gel. The DNA was stained with SYTO-60 (Invitrogen S11342: 1:10,000). The images of SYTO-60 signal and GFP signal were taken with an Odyssey Infrared Imaging System (Li-Cor).

## MagIC-cryo-EM of H1.8-GFP-bound nucleosomes isolated from chromosomes assembled in *Xenopus* egg extract (used in Figure 3)

Tween 20 was added to a final concentration of 0.01% to the 350 µL of fraction 5 from the interphase or metaphase sucrose gradient fractions shown in *Figure 3*, *Figure 3—figure supplement 2*. These samples were then mixed with 1 fmol of GFP nanobody-conjugated MagIC-cryo-EM beads. The mixture was incubated at 4 °C for 10 hr. The beads were washed four times with EM buffer C containing 0.01% Tween 20, as described above. Finally, the beads were resuspended in approximately 80 µL of EM buffer C containing 0.001% Tween 20.

To vitrify the grids, 4 µL of the samples were applied to plasma-cleaned graphene-coated Quantifoil gold R1.2/1.3 300-mesh grids (Quantifoil). The grid was then incubated on the 40x20 mm N52 neodymium disc magnets for 5 min and vitrified using the Vitrobot Mark IV (FEI) with a 2 s blotting time at room temperature under 100% humidity. The vitrified grids were loaded onto the Titan Krios (Thermo Fisher), equipped with a 300 kV field emission gun and a K3 direct electron detector (Gatan). A total of 677 movies for the interphase and 965 movies for the metaphase were collected at a magnification of ×64,000 (1.33 Å/pixel) using super-resolution mode, as managed by SerialEM (*Mastronarde, 2003*).

The analysis pipeline is described in *Figure 3—figure supplement 3*. Movie frames were corrected for motion using MotionCor2 (*Zheng et al., 2017*), which was installed in Relion v4 (*Scheres, 2012*). The micrographs for interphase and metaphase MagIC-cryo-EM were combined and subjected to particle picking. Particles were picked with Topaz v0.2.3 (*Bepler et al., 2019a*), using approximately 2000 manually picked nucleosome-like particles as training models. The picked particles were then extracted using CryoSPARC v4 (extraction box size = 256 pixels; *Punjani et al., 2017*). Nucleosome-containing particles were isolated through decoy classification using heterogeneous reconstruction with one nucleosome-like model and four decoy classes generated through ab initio reconstruction in CryoSPARC v4. CTF refinement and Bayesian polishing were applied to the nucleosome-containing particles in Relion v4 (*Scheres, 2012*; *Zivanov et al., 2019*). To isolate the nucleosome class with visible H1.8 density, three 3D references were generated through ab initio reconstruction in CryoSPARC v4 using purified nucleosome-like particles (Class similarity = 0.9). This step was repeated for the class with weak H1.8 density (Class A). Non-uniform refinement was performed in CryoSPARC v4 for each class. Subsequently, to isolate the H1.8-bound nucleosome structures in interphase and metaphase, the particles were separated based on their original movies. Using these particle sets, the 3D maps of the interphase and metaphase H1.8-bound nucleosomes were refined individually through non-uniform refinement in CryoSPARC v4. The final resolution was determined using the gold standard FSC threshold (FSC = 0.143).

## Isolation of interphase-specific H1.8-GFP containing complex by MagIC-cryo-EM (used in Figure 4)

Tween20 was added to a final concentration of 0.01%–350 µL of fraction 4 from the interphase sucrose gradient fractions shown in *Figure 3C*. The sample was then mixed with 1 fmol of GFP nanobody-conjugated MagIC-cryo-EM beads. The mixture was incubated at 4 °C for 10 hr. The beads were washed four times with EM buffer C containing 0.01% Tween 20, as described above. Finally, the beads were resuspended in approximately 80 µL of EM buffer C containing 0.001% Tween 20. The resuspended MagIC-cryo-EM beads solution was subjected to the MS and cryo-EM.

### Mass spectrometry

For the MS analysis, 20 µL of the resuspended solution containing the MagIC-cryo-EM beads isolating interphase-specific H1.8-GFP containing complex was incubated at 95 °C for 10 min to reverse the crosslink. The 20 µL each of the sucrose gradient fractions 4 and 5 (interphase and metaphase) was also incubated at 95 °C. The samples were then applied to an SDS-PAGE (4–20% gradient gel, Bio-Rad). The gel was stained with Coomassie Brilliant Blue G-250 (Thermo Fisher). The corresponding lane was cut into pieces approximately 2 mm × 2 mm in size. The subsequent destaining, in-gel digestion, and extraction steps were carried out as described (*Shevchenko et al., 2006*). In brief, the cut gel was destained using a solution of 30% acetonitrile and 100 mM ammonium bicarbonate in water. Gel pieces were then dehydrated using 100% acetonitrile. Disulfide bonds were reduced with dithiothreitol, and cysteines were alkylated using iodoacetamide. Proteins were digested by hydrating the gel pieces in a solution containing sequencing-grade trypsin and endopeptidase LysC in 50 mM ammonium bicarbonate. Digestion proceeded overnight at 37 °C. The resulting peptides were extracted three times with a solution of 70% acetonitrile and 0.1% formic acid. These extracted peptides were then purified using in-house constructed micropurification C18 tips. The purified peptides were subsequently analyzed by LC-MS/MS using a Dionex 3000 HPLC system equipped with an NCS3500RS nano- and microflow pump, coupled to an Orbitrap ASCEND mass spectrometer from Thermo Fisher Scientific. Peptides were separated by reversed-phase chromatography using solvent A (0.1% formic acid in water) and solvent B (80% acetonitrile, 0.1% formic acid in water) across a 70 min gradient. Spectra were recorded in positive ion data-dependent acquisition mode, with fragmentation of the 20 most abundant ions within each duty cycle. MS1 spectra were recorded with a resolution of 120,000 and an AGC target of 2e5. MS2 spectra were recorded with a resolution of 30,000 and an AGC target of 2e5. The spectra were then queried against a *Xenopus laevis* database (*Wühr et al., 2014*; *Peshkin et al., 2019*), concatenated with common contaminants, using MASCOT through Proteome Discoverer v.1.4 from Thermo Fisher Scientific. The abundance value for each protein is calculated as the average of the three most abundant peptides belonging to each protein (*Silva et al., 2006*). All detected proteins are listed in *Supplementary file 2*. The keratin-related proteins that were considered to be contaminated during sample preparation steps and the proteins with less than 5% coverage that were considered to be misannotation were not shown in *Figure 4D* and *Table 3*.

## Cryo-EM data collection of interphase-specific H1.8-GFP containing complex isolated by MagIC-cryo-EM beads (used in Figure 4)

To vitrify the grids, 4 µL of the resuspended solution containing the MagIC-cryo-EM beads isolated interphase-specific H1.8-GFP containing complex were applied to plasma-cleaned in-house graphene attached Quantifoil gold R1.2/1.3 300-mesh grids (Quantifoil). The grid was then incubated on the 40×20 mm N52 neodymium disc magnets for 5 min and vitrified using the Vitrobot Mark IV (FEI) with a 2 s blotting time at room temperature under 100% humidity. The vitrified grids were loaded onto the Titan Krios (ThermoFisher), equipped with a 300 kV field emission gun and a K3 direct electron detector (Gatan). At a magnification of ×105,000 (0.86 Å/pixel), 4543 movies were collected. At a magnification of ×105,000 (1.08 Å/pixel), 1807 movies were collected.

## Application of DuSTER for Cryo-EM analysis of interphase-specific H1.8-GFP containing complex isolated by MagIC-cryo-EM beads (used in Figure 4)

The pipeline to generate the initial 3D model is described in *Figure 4—figure supplement 3*. Movie frames are motion-corrected and dose-weighted patch motion correction in CryoSPARC v4 with output Fourier cropping factor 1/2 (*Punjani et al., 2017*). To remove low S/N ratio particles that are not reproducibly recentered during 2D classification, through DuSTER, particles picking with Topaz v0.2 *Bepler et al., 2019a* were repeated twice to assign two picked points for each protein particle on micrographs. Training of Topaz was performed individually for each picked particle set using the same approximately 2000 manually picked particles as training models. The particles in these two picked particle sets were then extracted using CryoSPARC v4 (extraction box size = 185.8 Å; *Punjani et al., 2017*). These two extracted particle sets were individually applied to 2D classification in CryoSPARC v4 (600 classes). These 2D classifications did not generate any reasonable 2D classes of interphase-specific H1.8-GFP containing complex that was expected from the particle images on the original motion-corrected micrographs. The reproducibility of the particles recentering can be assessed by the $D$. Smaller value of $D$ indicates that two pick points on each particle are reproducibly recentered during 2D classification. To remove duplicate particles at closed distances, we used this tool to keep the recentered points whose $D$ are shorter than $D_{TH}$. The DuSTER curation can be achieved by using the 'Remove Duplicate Particles' tool in CryoSPARC. Although the tool was originally designed to remove duplicate particles at closed distances, we used this tool to keep the recentered points whose $D$ are shorter than $D_{TH}$. All particles from two individual particle sets after the 2D classification were applied to the 'Remove Duplicate Particles' tool in CryoSPARC v4 using the 'Remove Duplicates Entirely' option (Minimum separation distance: 20 Å). Although the tool was originally designed to remove duplicate particles at closed distances, we used this tool to keep the recentered points whose $D$ are shorter than $D_{TH}$. The particles whose recentered points whose $D$ are shorter than $D_{TH}$ and were the particles used in further downstream processing, were sorted as 'rejected particles'. These particles were applied to the Particle Sets Tool in CryoSPARC v4 to split them into two individual particle sets. 2D DuSTER, including particle re-centering, particle extraction, and particle splitting steps, was repeated seven times. After seven rounds of 2D DuSTER, the particles were manually curated by removing the 2D classes with unreasonable sizes or shapes for the interphase-specific H1.8-GFP containing complex. The 2D images of removed classes are shown in *Figure 4—figure supplement 3*. After manual curation, the particles were further cleaned by an additional four rounds of 2D DuSTER. The particles were further cleaned by the Class Probability Filtering Tool in CryoSPARC v4. 2D classification was performed twice for one of the cleaned particle sets. The particles whose 2D class probability scores were lower than 0.3 in both replicates of 2D classification were removed. The redundant 2D classifications were necessary to prevent unintentional loss of high S/N particles. The duplicated class probability filtering was repeated six times. Using the filtered particles, 2D classification was performed twice. The high-resolution classes with reasonable protein-like features were manually selected from both 2D classification results. To prevent unintentional contamination of low S/N particles, the 92,382 particles that were selected in both 2D classification runs were used for ab initio 3D reconstruction (C5 symmetry applied). The 3D structure was highly similar to NPM2, and we were convinced that the interphase-specific H1.8-GFP containing complex is NPM2-H1.8-GFP complex.

The pipeline for the particle cleaning using 3D DuSTER is described in *Figure 4—figure supplement 5*. After seven rounds of 2D DuSTER for the particles picked by Topaz, decoy 3D classification was employed to remove nucleosomes and GFP complexed with GFP-nanobody. The nucleosome 3D model was generated by ab initio 3D reconstruction using the particles assigned to nucleosome-like 2D classes. The 3D model of GFP complexed with GFP-nanobody was modeled from the crystal structure of the complex (PDB ID: 3k1k; *Kirchhofer et al., 2010*) using EMAN2 (*Tang et al., 2007*). Noise 3D models were generated by ab initio 3D reconstruction using the low S/N particles that were removed during 2D DuSTER. Using these models and the initial 3D model of NPM2-H1.8-GFP, heterogeneous 3D refinement was performed twice in CryoSPARC v4. To prevent unintentional loss of high S/N particles, particles that were assigned to the nucleosome and GFP complexed with GFP-nanobody class in both heterogeneous 3D refinement results were removed. By using the Remove Duplicate Particles and Particle Sets tools in CryoSPARC v4, the particles in picked particle set 2

that corresponded to the particles cleaned by decoy classification were selected. Using both picked particle sets, heterogeneous 3D refinement of CryoSPARC v4 was performed individually. Using the same procedure as 2D DuSTER, the particles that were reproducibly centered in each particle set were selected (Minimum separation distance: 15 Å). 3D DuSTER was repeated six times. To conduct 3D DuSTER more comprehensively, 3D refinements were performed for each picked particle set three times. Particle curation based on the distance was performed for all nine combinations of these 3D refinement results, and this comprehensive 3D DuSTER was repeated once again. Using the particles in picked particle set 1 after 3D DuSTER, 2D classification was performed twice. The noise classes were manually selected from both 2D classification results. To prevent unintentional loss of high S/N particles, particles that were assigned to the noise class in both 2D classification runs were removed. This duplicated 2D classification and manual selection was repeated twice. During the 2D classification, 2D classes that represent GFP-nanobody were found. To remove the particles, duplicated decoy 3D classification was employed once again. The remaining 162,995 particles were used for the 3D structure reconstruction.

The pipeline for 3D structure reconstruction using the particle curated by 3D DuSTER is described in *Figure 4—figure supplement 6*. Using the 162,995 particles after the 3D DuSTER, *ab initio* 3D reconstruction (5 classes, C5) was performed five times. The particles assigned to the NPM2-like classes were selected. To prevent unintentional loss of high S/N particles, particles that were assigned to the noise class in all five ab initio 3D reconstruction runs were removed. For the 'averaged' NPM2 structure, a single 3D map was built by ab initio 3D reconstruction (1 class, C5) using the remaining 92,428 particles. The 3D map was refined by local refinement using the particles after symmetry expansion. For the structural variants of the NPM2, particles were split into the two classes by ab initio 3D reconstruction (2 class, C5). The ab initio 3D reconstruction (3 class, C5) was performed again for each class, and the particles were manually split into the three groups to generate 'open', 'half-open', and 'closed' NPM2 structures.

The initial atomic model of *Xenopus laevis* NPM2 pentamer was built by ColabFold v1.5.5, which implements AlphaFold2 and MMseqs2 (*Mirdita et al., 2022*; *Jumper et al., 2021*; *Steinegger and Söding, 2017*). The full-length *Xenopus laevis* NPM2 pentamer structure was docked on the cryo-EM maps by the Dock-in-map tool in Phenix v1.21 (*Afonine et al., 2018*). The atomic coordinates of the disordered regions were removed. The atomic model was refined using the Starmap v1.2.15 (*Lugmayr et al., 2023a*). The refined models were further refined using the real-space refinement in Phenix v1.21 (*Afonine et al., 2018*).

For reconstituting the 3D maps without applying symmetry, the particles used for reconstituting 'open', 'half-open', and 'closed' NPM2 structures were applied to the manual picking tool in cryoSPARC to remove the 3D alignment information attached to the particle images. The particle images were extracted and applied to the ab initio 3D reconstruction (1 class, C1).

3D FSC was plotted by the Orientation Diagnostics tool integrated in the cryoSPARC v4.4.

## AlphaFold2 prediction of the NPM2-H1.8 complex structure

The AF2 models of the *Xenopus laevis* NPM2-H1.8 complex were built by ColabFold v1.5.5, by submitting five NPM2 and one H1.8 amino acid sequence as input (*Mirdita et al., 2022*; *Jumper et al., 2021*; *Steinegger and Söding, 2017*).

## 3D structure visualization

Local resolution was estimated by cryoSPARC v4.4. All 3D structures, including cryo-EM density maps, cartoon depictions, and surface depictions with electrostatic potential, were visualized by the UCSF ChimeraX software (*Goddard et al., 2018*).

## Acknowledgements

This research was supported by a National Institutes of Health Grants (R35GM132111) to HF, Japan Society for the Promotion of Science Overseas Research Fellowships to HAK, and Osamu Hayaishi Memorial Scholarship for Study Abroad to YA. This research was also supported by the Stavros Niarchos Foundation (SNF) as part of its grant to the SNF Institute for Global Infectious Disease Research at The Rockefeller University. We are grateful to Mark Ebrahim, Johanna Sotiris, and Honkit Ng for their technical advice and assistance for the Cryo-EM and Soeren Heissel and Henrik Molina for MS

analysis, Amalia Pasolli assistance for EM, David Shechter for providing NPM2 and NAP1 antibodies, Genzhe Lu and Daniil Tagaev for their contributions to the optimization of MagIC-cryo-EM, and Rochelle Shih, Nick Prescott, Yiming Niu, and Isabel Wassing for comments on the manuscript. We also thank Seth Darst, Elizabeth Campbell, Thomas Huber, Michael Rout, Peter Fridy, Christopher Caffalette, Trevor Van Eeuwen, Hiro Furukawa, Takashi Onikubo, Sue Biggins, Daniel Barrero, and Mengqiu Jiang for consulting on the project. This work was conducted with the help of the High-Performance Computing Resource Center, Proteomics Resource Center, the Evelyn Gruss Lipper Cryo-Electron Microscopy Resource Center, Electron Microscopy Resource Center, and Bio-Imaging Resource Center at the Rockefeller University.

## Additional information

### Competing interests

Yasuhiro Arimura, Hide A Konishi, Hironori Funabiki: filed a patent application encompassing aspects of MagIC-cryo-EM (PCT/US2023/03315).

### Funding

| Funder | Grant reference number | Author |
| --- | --- | --- |
| National Institute of General Medical Sciences | R35GM132111 | Hironori Funabiki |
| Japan Society for the Promotion of Science | Overseas Research Fellowships | Hide A Konishi |
| Osamu Hayaishi Memorial Scholarship | Scholarship for Study Abroad | Yasuhiro Arimura |
| Stavros Niarchos Foundation | SNF Institute for Global Infectious Disease Research | Hironori Funabiki |

The funders had no role in study design, data collection and interpretation, or the decision to submit the work for publication.

### Author contributions

Yasuhiro Arimura, Conceptualization, Data curation, Formal analysis, Funding acquisition, Validation, Investigation, Visualization, Methodology, Writing – original draft, Project administration, Writing – review and editing; Hide A Konishi, Conceptualization, Funding acquisition, Investigation, Writing – review and editing; Hironori Funabiki, Conceptualization, Resources, Supervision, Funding acquisition, Project administration, Writing – review and editing

### Author ORCIDs

Yasuhiro Arimura ⓘ https://orcid.org/0000-0002-1903-6076
Hide A Konishi ⓘ https://orcid.org/0000-0001-9529-9321
Hironori Funabiki ⓘ https://orcid.org/0000-0003-4831-4087

### Ethics

Xenopus laevis was purchased from Xenopus 1 (female, 4270; male, 4235). Vertebrate animal protocols (20031 and 23020) approved by the Rockefeller University Institutional Animal Care and Use Committee were followed.

Reviewer #1 (Public review): https://doi.org/10.7554/eLife.103486.3.sa1
Reviewer #2 (Public review): https://doi.org/10.7554/eLife.103486.3.sa2
Author response https://doi.org/10.7554/eLife.103486.3.sa3

## Additional files

### Supplementary files
Supplementary file 1. Statistics of the cryo-EM structures.

Supplementary file 2. Mass spectrometry data.

MDAR checklist

### Data availability
Cryo-EM density maps have been deposited in the EM Data Resource under accession codes EMD-42599 (in vitro reconstituted poly-nucleosome), EMD-42598 (in vitro reconstituted H1-GFP bound nucleosome), EMD-42594 (*Xenopus* egg extract H1-GFP bound nucleosome structure containing both interphase and metaphase particles), EMD-42596 (interphase Xenopus egg extract H1-GFP bound nucleosome), EMD-42597 (metaphase *Xenopus* egg extract H1-GFP bound nucleosome), EMD-43238 (Averaged NPM2-H1.8-GFP structure), EMD- 43239 (open NPM2-H1.8-GFP structure), and EMD-43240 (closed NPM2-H1.8-GFP structure). The atomic coordinates have been deposited in the Protein Data Bank under accession codes PDB 8VHI (averaged NPM2-H1.8-GFP structure), PDB 8VHJ (open NPM2-H1.8-GFP structure), and PDB 8VHK (closed NPM2-H1.8-GFP structure). The plasmids for generating MagIC-cryo-EM beads were deposited to Addgene under accession codes #214835 (Non tagged Avidin), #214836 (SPYtag-Histag-Avidin), #214837 (SPYtag-GFPnanobody), #214838 (Cys-3HB-SPYcatcher), #214839 (Cys-30nmSAH-SPYcatcher), and #214840 (Cys-60nmSAH-SPYcatcher).

The following datasets were generated:

| Author(s) | Year | Dataset title | Dataset URL | Database and Identifier |
|---|---|---|---|---|
| Arimura Y, Funabiki H | 2024 | Nucleosome in polynucleosome attached on magnetic beads | https://www.ebi.ac.uk/emdb/EMD-42599 | Electron Microscopy Data Bank, EMD-42599 |
| Arimura Y, Funabiki H | 2024 | MagIC-cryo-EM of the *Xenopus* H1.8-GFP nucleosome reconstituted in vitro | https://www.ebi.ac.uk/emdb/EMD-42598 | Electron Microscopy Data Bank, EMD-42598 |
| Arimura Y, Funabiki H | 2024 | MagIC-cryo-EM of H1.8-GFP nucleosome isolated from *Xenopus* egg extract (interphase and metaphase data are mixed) | https://www.ebi.ac.uk/emdb/EMD-42594 | Electron Microscopy Data Bank, EMD-42594 |
| Arimura Y, Funabiki H | 2024 | MagIC-cryo-EM of the H1.8-GFP nucleosome formed in the *Xenopus* egg extract interphase chromosomes | https://www.ebi.ac.uk/emdb/EMD-42596 | Electron Microscopy Data Bank, EMD-42596 |
| Arimura Y, Funabiki H | 2024 | MagIC-cryo-EM of the H1.8-GFP nucleosome formed in the *Xenopus* egg extract metaphase chromosomes | https://www.ebi.ac.uk/emdb/EMD-42597 | Electron Microscopy Data Bank, EMD-42597 |
| Arimura Y, Funabiki H | 2024 | NPM2-H1.8 isolated from *Xenopus* egg extract | https://www.ebi.ac.uk/emdb/EMD-43238 | Electron Microscopy Data Bank, EMD-43238 |
| Arimura Y, Funabiki H | 2024 | NPM2-H1.8 isolated from *Xenopus* egg extract | https://www.ebi.ac.uk/emdb/EMD-43239 | Electron Microscopy Data Bank, EMD-43239 |
| Arimura Y, Funabiki H | 2024 | NPM2-H1.8 isolated from *Xenopus* egg extract (Stretched form) | https://www.ebi.ac.uk/emdb/EMD-43240 | Electron Microscopy Data Bank, EMD-43240 |
| Arimura Y, Funabiki H | 2024 | NPM2-H1.8 isolated from *Xenopus* egg extract | https://www.rcsb.org/structure/8VHI | RCSB Protein Data Bank, 8VHI |

*Continued on next page*

*Continued*

| Author(s) | Year | Dataset title | Dataset URL | Database and Identifier |
|---|---|---|---|---|
| Arimura Y, Funabiki H | 2024 | NPM2-H1.8 isolated from *Xenopus* egg extract (Bent form) | https://www.rcsb.org/structure/8VHJ | RCSB Protein Data Bank, 8VHJ |
| Arimura Y, Funabiki H | 2024 | NPM2-H1.8 isolated from *Xenopus* egg extract (Stretched form) | https://www.rcsb.org/structure/8VHK | RCSB Protein Data Bank, 8VHK |

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

# Appendix 1

**Appendix 1—key resources table**

| Reagent type (species) or resource | Designation | Source or reference | Identifiers | Additional information |
|---|---|---|---|---|
| Antibody | Anti-histone H1.8 (*Xenopus laevis*) Rabbit polyclonal antibody | *Jenness et al., 2018* | RU2130 | 1 µg/mL |
| Antibody | Anti-histone H3T3ph mouse monoclonal antibody | *Kelly et al., 2008* | 16B2 | 1:5000 dilution |
| Antibody | IRDye 680LT goat anti-rabbit IgG | LI-COR | 926–68021; RRID:AB_10706309 | 1:15,000 dilution |
| Antibody | IR Dye 800CW goat anti-mouse IgG | LI-COR | 926–32210; RRID:AB_621842 | 1:10,000 dilution |
| Antibody | Anti-histone NAP1 (*Xenopus laevis*) Rabbit polyclonal antibody | *Lorton et al., 2023* | N/A | 1:500 dilution |
| Antibody | Anti-histone NPM2 (*Xenopus laevis*) Rabbit polyclonal antibody | *Lorton et al., 2023* | N/A | 1:500 dilution |
| Antibody | Anti-rabbit IgG | SIGMA | I5006, RRID:AB_1163659 | |
| Antibody | Anti-rabbit IgG (purified from pre-bleed sera) | This work | N/A | |
| Antibody | Cy3 AffiniPure F(ab')$_2$ Fragment Donkey Anti-Mouse IgG (H+L) | Jackson ImmunoResearch | 715-166-150, RRID:AB_2340816 | 1:500 dilution |
| Antibody | Cy5 AffiniPure Donkey Anti-Rabbit IgG (H+L) | Jackson ImmunoResearch | 711-175-152, RRID:AB_2340607 | 1:500 dilution |
| Strain, strain background (*E. coli*) | *E. coli* Rosetta (DE3) | Novagen | 70954–3 | |
| Strain, strain background (*E. coli*) | *E. coli* BL21(DE3) | Novagen | 69450–3 | |
| Strain, strain background (*E. coli*) | *E. coli* JM101 | Agilent | 200234 | |
| Biological sample (*Xenopus laevis*) | *Xenopus laevis* sperm nuclei | Nasco, Isolated from male *Xenopus laevis* | LM00715 (male *Xenopus laevis*) | |
| Biological sample (*Xenopus laevis*) | *Xenopus laevis* egg | Nasco, Laid by female *Xenopus laevis* | LM00535 (female *Xenopus laevis*) | |
| Biological sample (*Xenopus laevis*) | Alexa594-labeled-tubulin | *Hyman et al., 1991*; Isolated from bovine brain | n/a | |
| Chemical compound, drug | Chorionic gonadotropin human | Sigma | CG10-1VL | |
| Chemical compound, drug | Pregnant Mare Serum Gonadotropin (PMSG) | Prospec | HOR-272 | |
| Chemical compound, drug | Cycloheximide | Sigma | C-7698 | |
| Chemical compound, drug | CaCl2 | Sigma Aldrich | C7902-500G | |

*Appendix 1 Continued on next page*

*Appendix 1 Continued*

| Reagent type (species) or resource | Designation | Source or reference | Identifiers | Additional information |
|---|---|---|---|---|
| Peptide, recombinant protein | Cyclin B Δ90 | *Glotzer et al., 1991* | N/A | |
| Chemical compound, drug | Cysteine | Sigma | C7352-1KG | |
| Chemical compound, drug | PIPES | Alfa Aesar | A16090 | |
| Chemical compound, drug | NaCl | Millipore Sigma | SX0420-5 | |
| Chemical compound, drug | Glycerol | Alfa Aesar | 36646 | |
| Chemical compound, drug | EGTA | Sigma Aldrich | E4378-250G | |
| Chemical compound, drug | MgCl2 | Acros organic | 197530010 | |
| Chemical compound, drug | β-glycerophosphate | Acros organic | 410991000 | |
| Chemical compound, drug | Sodium butyrate | Aldrich | 303410–100 G | |
| Chemical compound, drug | Formaldehyde | Fisher bioreagents | BP531-25 | |
| Chemical compound, drug | HEPES | Akron biotech | AK1069-1000 | |
| Chemical compound, drug | Sucrose | Fisher chemical | S5-3 | |
| Chemical compound, drug | KCl | Fisher chemical | P217-3 | |
| Chemical compound, drug | Spermidine | Sigma | S-2501 | |
| Chemical compound, drug | Spermine | Sigma | S1141-5G | |
| Chemical compound, drug | cOmplete EDTA-free Protease Inhibitor Cocktail | Roche | 11873580001 | |
| Chemical compound, drug | Leupeptin | Millipore Sigma | El8 | |
| Chemical compound, drug | Pepstatin | Sigma | P5318-25MG | |
| Chemical compound, drug | Chymostatin | Millipore Sigma | El6 | |
| Peptide, recombinant protein | RNaseA | Thermo Scientific | EN0531 | |
| Peptide, recombinant protein | Proteinase K solution | Roche | 3115828001 | |
| Chemical compound, drug | SYBR-safe | Invitrogen | S33102 | |
| Chemical compound, drug | 4–20% gradient SDS-PAGE gel | BioRad | 4561096 | |
| Chemical compound, drug | SYTO-60 | Invitrogen | S11342 | |

*Appendix 1 Continued on next page*

*Appendix 1 Continued*

| Reagent type (species) or resource | Designation | Source or reference | Identifiers | Additional information |
|---|---|---|---|---|
| Chemical compound, drug | Tris | Sigma | T1503-5KG | |
| Chemical compound, drug | Glycine | Sigma | G7126-5KG | |
| Chemical compound, drug | Amicon Ultra centrifugal filter 100 K | Millipore Sigma | UFC510024 | |
| Chemical compound, drug | Methanol | Fisher chemical | A452SK-4 | |
| Chemical compound, drug | Tube-o-dialyzer 15 kDa | G-Biosciences | 786–618 | |
| Chemical compound, drug | Amicon Ultra centrifugal filters 3 K | Millipore Sigma | UFC500324 | |
| Chemical compound, drug | Coomassie Brilliant Blue G-250 | Calbiochem | 3340 | |
| Peptide, recombinant protein | *Xenopus laevis* H2A, H2B, H3.2, H4 | *Zierhut et al., 2014* | N/A | |
| Chemical compound, drug | Ni-NTA beads | QIAGEN | 30210 | |
| Chemical compound, drug | Carbenicillin | Alfa Aesar | J61949 | |
| Chemical compound, drug | Methanol | Fisher chemical | A452SK-4 | |
| Chemical compound, drug | Trehalose | Sigma | T0167-100G | |
| Chemical compound, drug | 1,6,-hexanediol | Alfa Aesar | A12439 | |
| Peptide, recombinant protein | MluI | New England Biolabs | R3198S | |
| Peptide, recombinant protein | AscI | New England Biolabs | R0558S | |
| Peptide, recombinant protein | XhoI | New England Biolabs | R0146S | |
| Chemical compound, drug | EZ-link Maleimide-PEG2-Biotin | Thermo | A39261 | |
| Chemical compound, drug | Isopropyl-β-D-thiogalactopyranoside (IPTG) | RPI research products | I56000-25.0 | |
| Peptide, recombinant protein | Mono-SPYtag-avidin tetramer | This work | | |
| Peptide, recombinant protein | Biotin-30 nm-SAH-SPYcatcher003 | This work | | |
| Peptide, recombinant protein | Biotin-60 nm-SAH-SPYcatcher003 | This work | | |
| Peptide, recombinant protein | Biotin-90 nm-SAH-SPYcatcher003 | This work | | |
| Peptide, recombinant protein | Biotin-3HB-SPYcatcher003 | This work | | |
| Peptide, recombinant protein | SPYtag-GFPenhancer-GGGGS4-LaG16 | This work | | |

*Appendix 1 Continued on next page*

*Appendix 1 Continued*

| Reagent type (species) or resource | Designation | Source or reference | Identifiers | Additional information |
|---|---|---|---|---|
| Peptide, recombinant protein | SPYtag-GFP enhancer nanobody | This work | | |
| Peptide, recombinant protein | SPYtag-LaG (llama antibody against GFP)–10 | This work | | |
| Peptide, recombinant protein | H1.8-GFP | This work | | |
| Peptide, recombinant protein | MNase | This work | | |
| Peptide, recombinant protein | HaeII | New England Biolabs | R0107S | |
| Peptide, recombinant protein | DraI | New England Biolabs | R0129S | |
| Peptide, recombinant protein | EcoRI | New England Biolabs | R3101S | |
| Peptide, recombinant protein | XbaI | New England Biolabs | R0145S | |
| Peptide, recombinant protein | Klenow fragment | New England Biolabs | M0212S | |
| Chemical compound, drug | biotin-14-dATP | Jena Bioscience GmbH | NU-835-BIO14-S | |
| Chemical compound, drug | Intercept TBS Blocking Buffer | LI-COR Biosciences | 927–60001 | |
| Chemical compound, drug | nitrocellulose membrane | Cytiva | 10600000 | |
| Chemical compound, drug | Protein-A coupled Dynabeads | Thermo Fisher Scientific | 10001D | |
| Chemical compound, drug | Hi-load Superdex200 16/600 column (Cytiva) | Cytiva | 28989335 | |
| Chemical compound, drug | Hi-load Superdex75 16/600 column (Cytiva) | Cytiva | 28989333 | |
| Chemical compound, drug | BS3 (bis(sulfosuccinimidyl) suberate) | Thermo Fisher Scientific | A39266 | |
| Chemical compound, drug | 1 x PhosSTOP | Roche | 4906845001 | |
| Chemical compound, drug | Absolute Mag streptavidin nano-magnetic beads | CD bioparticles | WHM-X047 | |
| Chemical compound, drug | monolayer graphene grown on the copper foil | Grolltex | GRF23-L062–6x6 | |
| Chemical compound, drug | polymethyl methacrylate | Micro chem | EL6 | |
| Chemical compound, drug | Ammonium Persulfate | Thermo Scientific | AC401165000 | |
| Chemical compound, drug | 2-butanone | Thermo Scientific | AA39119K7 | |
| Chemical compound, drug | 2-propanol | Fisher Scientific | BP26184 | |
| Chemical compound, drug | NucBlue Fixed Cell ReadyProbes Reagent | Thermo Fisher | R37606 | |

*Appendix 1 Continued*

| Reagent type (species) or resource | Designation | Source or reference | Identifiers | Additional information |
|---|---|---|---|---|
| Other | in vitro reconstituted poly-nucleosome on magnetic beads | This work | EMD-42599 | |
| Other | in vitro reconstituted H1-GFP bound nucleosome (MagIC-cryo-EM) | This work | EMD-42598 | |
| Other | *Xenopus* egg extract H1-GFP bound nucleosome structure containing both interphase and metaphase particles (MagIC-cryo-EM) | This work | EMD-42594 | |
| Other | interphase *Xenopus* egg extract H1-GFP bound nucleosome (MagIC-cryo-EM) | This work | EMD-42596 | |
| Other | metaphase *Xenopus* egg extract H1-GFP bound nucleosome (MagIC-cryo-EM) | This work | EMD-42597 | |
| Other | Averaged NPM2-H1.8-GFP structure (MagIC-cryo-EM) | This work | EMD-43238 | |
| Other | open NPM2-H1.8-GFP structure (MagIC-cryo-EM) | This work | EMD-43239 | |
| Other | closed NPM2-H1.8-GFP structure (MagIC-cryo-EM) | This work | EMD- 43240 | |
| Other | averaged NPM2-H1.8-GFP structure | This work | PDB 8VHI | |
| Other | open NPM2-H1.8-GFP structure | This work | PDB 8VHJ | |
| Other | closed NPM2-H1.8-GFP structure | This work | PDB 8VHK | |
| Recombinant DNA reagent | pET21-SPY-His6-tag streptavidin | This work | Addgene 214836 | |
| Recombinant DNA reagent | pET21-streptavidin | This work | Addgene 214835 | |
| Recombinant DNA reagent | pQE80-His14-bdSUMO-Cys-30nm-SAH-SPYcatcher003 | This work | Addgene 214839 | |
| Recombinant DNA reagent | pQE80-His14-bdSUMO-Cys-60nm-SAH-SPYcatcher003 | This work | Addgene 214840 | |
| Recombinant DNA reagent | pQE80-His14-bdSUMO-Cys-3HB-SPYcatcher003 | This work | Addgene 214838 | |
| Recombinant DNA reagent | pQE80-SPYtag-GFP enhancer nanobody | This work | Addgene 214837 | |
| Recombinant DNA reagent | *Xenopus laevis* | NASCO | Wild type | |
| Recombinant DNA reagent | pSF1389 | *Frey and Görlich, 2014* | Addgene 104962 | |
| Recombinant DNA reagent | pCDNA-FRT-FAK30 | *Ritt et al., 2013* | Addgene 59121 | |
| Recombinant DNA reagent | pET21a-Streptavidin-Alive | *Howarth et al., 2006* | Addgene 20860 | |

*Appendix 1 Continued on next page*

*Appendix 1 Continued*

| Reagent type (species) or resource | Designation | Source or reference | Identifiers | Additional information |
|---|---|---|---|---|
| Software, algorithm | MASCOT through Proteome Discoverer v.1.4 | Thermo Scientific | N/A | |
| Software, algorithm | MOTIONCOR2 | *Zheng et al., 2017* | https://emcore.ucsf.edu/ucsf-software | |
| Software, algorithm | RELION v4 | *Scheres, 2012*; *Scheres, 2022* | https://github.com/3dem/relion | |
| Software, algorithm | CryoSPARC v3 and v4 | *Punjani et al., 2017* | https://cryosparc.com | |
| Software, algorithm | Chimera | *Pettersen et al., 2004* | https://www.cgl.ucsf.edu/chimera/ | |
| Software, algorithm | ChimaraX | *Pettersen et al., 2004* | https://www.rbvi.ucsf.edu/chimerax/ | |
| Software, algorithm | Topaz v0.2 | *Bepler et al., 2019a*; *Bepler et al., 2019b* | https://github.com/tbepler/topaz | |
| Software, algorithm | Bsoft | *Cardone et al., 2013* | https://lsbr.niams.nih.gov/bsoft/ | |
| Software, algorithm | Coot | *Emsley and Cowtan, 2004* | https://www2.mrc-lmb.cam.ac.uk/personal/pemsley/coot/ | |
| Software, algorithm | PHENIX | *Liebschner et al., 2019* | https://phenix-online.org/documentation/reference/refinement.html | |
| Software, algorithm | APBS | *Baker et al., 2001* | https://www.cgl.ucsf.edu/chimera/docs/ContributedSoftware/apbs/apbs.html | |
| Software, algorithm | PDB2PQR | *Dolinsky et al., 2007* | https://www.cgl.ucsf.edu/chimera/docs/ContributedSoftware/apbs/pdb2pqr.html | |
| Software, algorithm | SWISS-MODEL | *Waterhouse et al., 2018* | https://swissmodel.expasy.org | |
| Software, algorithm | RING 2.0 webserver | *Piovesan et al., 2016* | http://old.protein.bio.unipd.it/ring/ | |
| Software, algorithm | Pyem v0.5 | *Asarnow et al., 2019* | https://github.com/asarnow/pyem | |
| Software, algorithm | ImageJ | *Schneider et al., 2012* | RRID:SCR_003070, https://imagej.nih.gov/ij/ | |
| Software, algorithm | Python | Python Software Foundation | https://www.python.org | |
| Software, algorithm | Jupyter Notebook | Project Jupyter | https://jupyter.org | |
| Software, algorithm | Microsoft Excel | Microsoft | https://www.microsoft.com/en-us/microsoft-365/excel | |
| Software, algorithm | Soft-WoRx (Applied Precision) | SoftWoRx software | RRID:SCR_019157 | |

*Appendix 1 Continued on next page*

*Appendix 1 Continued*

| Reagent type (species) or resource | Designation | Source or reference | Identifiers | Additional information |
|---|---|---|---|---|
| Software, algorithm | Fiji (ver. 2.9.0) | *Schindelin et al., 2012* | RRID:SCR_002285, https://imagej.net/software/fiji/downloads | |
| Software, algorithm | RStudio ver. RSTUDIO-2023.09.1–494 | *RStudio Team, 2020* | https://posit.co/download/rstudio-desktop/ | |
| Software, algorithm | R (ver. 4.2.2) | *R Development Core Team, 2021* | https://www.R-project.org/ | |
| Software, algorithm | ggplot2 | *Wickham et al., 2016* | https://ggplot2.tidyverse.org/ | |
| Software, algorithm | ggthemes (v5.1.0.9000) | *Arnold, 2024* | https://jrnold.github.io/ggthemes/ | |
| Software, algorithm | ggpubr | *Kassambara, 2023* | https://rpkgs.datanovia.com/ggpubr/ | |
| Software, algorithm | dplyr | *Wickham et al., 2023* | https://dplyr.tidyverse.org/ | |
| Software, algorithm | lawstat | *Hui et al., 2008* | https://doi.org/10.18637/jss.v028.i03 | |
| Software, algorithm | see | *Lüdecke et al., 2021* | https://joss.theoj.org/papers/10.21105/joss.03393 | |
| Software, algorithm | scales | *Wickham et al., 2025* | https://scales.r-lib.org | |
| Software, algorithm | GIMP software (ver. 4.2.2). | GIMP (GNU Image Manipulation Program) | https://gimp.org | |
| Software, algorithm | Inkscape 1.2 | Inkscape Developers 2022 | https://inkscape.org/ | |
| Software, algorithm | ColabFold v1.5.5 | *Mirdita et al., 2022*; *Mirdita et al., 2025* | https://github.com/sokrypton/ColabFold | |
| Software, algorithm | Starmap v1.2.15 | *Lugmayr et al., 2023a*; *Lugmayr et al., 2023b* | https://github.com/wlugmayr/chimerax-starmap | |
| Other | *Xenopus laevis* MS database | *Wühr et al., 2014* | https://scholar.princeton.edu/wuehr/protein-concentrations-xenopus-egg | |
| Other | Centrifuge rotor | Beckman Coulter | SX241.5 | |
| Other | Ultra centrifuge rotor | Beckman Coulter | SW55Ti | |
| Other | Ultra centrifuge rotor | Beckman Coulter | SW40Ti | |
| Other | Centrifuge rotor | Beckman Coulter | JS 5.3 | |
| Other | Centrifuge | Beckman Coulter | Allegron X-30R | |
| Other | Ultra centrifuge | Beckman Coulter | Optima L80 | |
| Other | Centrifuge | Beckman Coulter | Avanti J-26S | |
| Other | Imaging System | Li-Cor | Odyssey Infrared Imaging System | |

*Appendix 1 Continued on next page*

*Appendix 1 Continued*

| Reagent type (species) or resource | Designation | Source or reference | Identifiers | Additional information |
|---|---|---|---|---|
| Other | LC-MS/MS system | Thermo Scientific | Dionex3000 HPLC, NCS3500RS nano- and microflow pump, Q-Exactive HF mass spectrometer | |
| Other | Plunge freezer | FEI | Vitrobot Mark IV | |
| Other | Plasma Cleaner | Gatan | Solarus II | |
| Other | Cryo-EM | FEI | Talos Arctica TEM | |
| Other | Cryo-EM | FEI | Titan Krios TEM | |
| Other | Cryo-EM detector | GATAN | K2 Camera | |
| Other | Cryo-EM detector | GATAN | K3 Camera | |
| Other | N52 neodymium magnets | DIYMAG | D40x20–2P-NEW | |
| Other | Cryo-EM grid | Quantifoil | Gold R 1.2/1.3 300 | |
| Other | Spectrophotometers | Thermo Scientific | Nanodrop ND-2000C | |
| Other | Centrifuge rotor | Beckman Coulter | JS 7.5 | |
| Other | Fluorescence microscope | Applied Precision | DeltaVision Image Restoration microscope | |

