## [Editor Report · eLife Assessment]

This study follows up on Arimura et al's powerful new method MagIC-Cryo-EM for imaging native complexes at high resolution. Using a clever design embedding protein spacers between the antibody and the nucleosomes purified, thereby minimizing interference from the beads, the authors concentrate linker histone variant H1.8 containing nucleosomes. From these samples, the authors obtain **convincing** atomic structures of the H1.8 bound chromatosome purified from interphase and metaphase cells, finding a NPM2 chaperone bound form exists as well. Caveats previously noted have been addressed nicely in the revision, strengthening the overall conclusions. This is an **important** new tool in the arsenal of single molecule biologists, permitting a deep dive into structure of native complexes, and will be of high interest to a broad swathe of scientists studying native macromolecules present at low concentrations in cells.

---

## [Referee Report · Reviewer #1 (Public review)]

Summary:

In this manuscript, Arimura et al describe MagIC-Cryo-EM, an innovative method for immune-selective concentrating of native molecules and macromolecular complexes for Cryo-EM imaging and single-particle analysis. Typically, Cryo-EM imaging requires much larger concentrations of biomolecules than those that are feasible to achieve by conventional biochemical fractionation. This manuscript is meticulously and clearly written and the new technique is likely to become a great asset to other electron microscopists and chromatin researchers.

Strengths:

Previously, Arimura et al. (Mol. Cell 2021) isolated from *Xenopus* extract and resolved by Cryo-EM a sub-class of native nucleosomes conjugated containing histone H1.8 at the on-dyad position, similar to that previously observed by other researchers with reconstituted nucleosomes. Here they sought to analyze immuno-selected nucleosomes aiming to observe specific modes of H1.8 positioning (e.g. on-dyad and off-dyad) and potentially reveal structural motifs responsible for the decreased affinity of H1.8 for the interphase chromatin compared to metaphase chromosomes. The main strength of this work is a clever and novel methodological design, in particular the engineered protein spacers to separate captured nucleosomes from streptavidin beads for clear imaging. The authors provide a detailed step-by-step description of MagIC-Cryo-EM procedure including nucleosome isolation, preparation of GFP nanobody attached magnetic beads, optimization of the spacer length, concentration of the nucleosomes on graphene grids, data collection and analysis, including their new DUSTER method to filter-out low signal particles. This tour de force methodology should facilitate the consideration of MagIC-Cryo-EM by other electron microscopists, especially for analysis of native nucleosome complexes.

In pursuit of biologically important new structures, the immune-selected H1.8-containing nucleosomes were solved at about 4A resolution; their structure appears to be very similar to the previously determined structure of H1.8-reconstituted nucleosomes. There were no apparent differences between the metaphase and interphase complexes suggesting that the on-dyad and off-dyad positioning does not explain the differences in H1.8 - nucleosome binding. However, they were able to identify and solve complexes of H1.8-GFP with histone chaperone NPM2 in a closed and open conformation providing mechanistic insights for H1-NPM2 binding and the reduced affinity of H1.8 to interphase chromatin as compared to metaphase chromosomes.

MagIC technique still has certain limitations resulting from formaldehyde fixation, use of bacterial-expressed recombinant H1.8-GFP, and potential effects of magnetic beads and/or spacer on protein structure, which are explicitly discussed in the text. Notwithstanding these limitations, MagIC-Cryo-EM is expected to become a great asset to other electron microscopists and biochemists studying native macromolecular complexes.

Comments on revisions:

In the revision, Arimura et al. have constructively addressed the reviewer's concerns, by discussing possible limitations and including additional information on proteomic analysis and H1.8-NPM2 structures.

The revised manuscript and rebuttal letter strengthen my initial opinion that this paper describes an innovative method for immune-selective concentrating of native molecules and macromolecular complexes thus enabling Cryo-EM imaging and structural analysis of native nucleosome complexes at low concentration. This manuscript is meticulously and clearly written and may become a great asset to other electron microscopists and chromatin researchers

---

## [Referee Report · Reviewer #2 (Public review)]

Summary:

The authors present a straightforward and convincing demonstration of a reagent and workflow that they collectively term "MagIC-cryo-EM", in which magnetic nanobeads combined with affinity linkers are used to specifically immobilize and locally concentrate complexes that contain a protein-of-interest. As a proof of concept, they localize, image, and reconstruct H1.8-bound nucleosomes reconstructed from frog egg extracts. The authors additionally devised an image-processing workflow termed "DuSTER", which increases the true positive detections of the partially ordered NPM2 complex. The analysis of the NPM2 complex {plus minus} H1.8 was challenging because only ~60 kDa of protein mass was ordered. Overall, single-particle cryo-EM practitioners should find this study useful.

Strengths:

The rationale is very logical and the data are convincing.

Weaknesses:

I have seen an earlier version of this study at a conference. The conference presentation was much easier to follow than the current manuscript. It is as if this manuscript had undergone review at another journal and includes additional experiments to satisfy previous reviewers. Specifically, the NPM2 results don't seem to add much to the main story (MagIC-cryo-EM) and read more like an addendum. The authors could probably publish the NPM2 results separately, which would make the core MagIC results (sans DusTER) easier to read.

Comments on revisions:

The authors have addressed my concerns. Congratulations!

---

## [Author Response]

The following is the authors’ response to the original reviews.

**Reviewer #1 (Public review):**
Summary:In this manuscript, Arimura et al describe MagIC-Cryo-EM, an innovative method for immune-selective concentrating of native molecules and macromolecular complexes for Cryo-EM imaging and single-particle analysis. Typically, Cryo-EM imaging requires much larger concentrations of biomolecules than that are feasible to achieve by conventional biochemical fractionation. Overall, this manuscript is meticulously and clearly written and may become a great asset to other electron microscopists and chromatin researchers.Strengths:Previously, Arimura et al. (Mol. Cell 2021) isolated from *Xenopus* extract and resolved by Cryo-EM a sub-class of native nucleosomes conjugated containing histone H1.8 at the on-dyad position, similar to that previously observed by other researchers with reconstituted nucleosomes. Here they sought to analyze immuno-selected nucleosomes aiming to observe specific modes of H1.8 positioning (e.g. on-dyad and off-dyad) and potentially reveal structural motifs responsible for the decreased affinity of H1.8 for the interphase chromatin compared to metaphase chromosomes. The main strength of this work is a clever and novel methodological design, in particular the engineered protein spacers to separate captured nucleosomes from streptavidin beads for a clear imaging. The authors provide a detailed step-by-step description of MagIC-Cryo-EM procedure including nucleosome isolation, preparation of GFP nanobody attached magnetic beads, optimization of the spacer length, concentration of the nucleosomes on graphene grids, data collection and analysis, including their new DUSTER method to filter-out low signal particles. This tour de force methodology should facilitate considering of MagIC-CryoEM by other electron microscopists especially for analysis of native nucleosome complexes.In pursue of biologically important new structures, the immune-selected H1.8-containing nucleosomes were solved at about 4A resolution; their structure appears to be very similar to the previously determined structure of H1.8-reconstituted nucleosomes. There were no apparent differences between the metaphase and interphase complexes suggesting that the on-dyad and off-dyad positioning does not explain the differences in H1.8 - nucleosome binding. However, they were able to identify and solve complexes of H1.8-GFP with histone chaperone NPM2 in a closed and open conformation providing mechanistic insights for H1-NPM2 binding and the reduced affinity of H1.8 to interphase chromatin as compared to metaphase chromosomes.Weaknesses:Still, I feel that there are certain limitations and potential artifacts resulting from formaldehyde fixation, use of bacterial-expressed recombinant H1.8-GFP, and potential effects of magnetic beads and/or spacer on protein structure, that should be more explicitly discussed.

We thank the reviewer for recognizing the significance of our methods and for constructive comments. To respond to the reviewer's criticism, we revised the “Limitation of the study” section (page 12, line 420) as indicated by the underlines below.

“While MagIC-cryo-EM is envisioned as a versatile approach suitable for various biomolecules from diverse sources, including cultured cells and tissues, it has thus far been tested only with H1.8-bound nucleosome and H1.8-bound NPM2, both using antiGFP nanobodies to isolate GFP-tagged H1.8 from chromosomes assembled in *Xenopus* egg extracts after pre-fractionation of chromatin. To apply MagIC-cryo-EM for the other targets, the following factors must be considered: (1) Pre-fractionation. This step (e.g., density gradient or gel filtration) may be necessary to enrich the target protein in a specific complex from other diverse forms (such as monomeric forms, subcomplexes, and protein aggregates). (2) Avoiding bead aggregation. Beads may be clustered by targets (if the target complex contains multiple affinity tags or is aggregated), nonspecific binders, and the target capture modules. To directly apply antibodies that recognize the native targets and specific modifications, optimization to avoid bead aggregation will be important. (3) Stabilizing complexes. The target complexes must be stable during the sample preparation. Crosslink was necessary for the H1.8-GFP-bound nucleosome. (4) Loading the optimum number of targets on the bead. The optimal number of particles per bead differs depending on target sizes, as larger targets are more likely to overlap. For H1.8-GFP-bound nucleosomes, 500 to 2,000 particles per bead were optimal. We expect that fewer particles should be coated for larger targets.”

We would like to note that while the use of bacterially expressed GFP-tagged H1.8 and MagIC-cryo-EM may potentially influence the structure of the H1.8-bound nucleosome, the structures of GFP-tagged H1.8-bound nucleosomes isolated from chromosomes assembled in *Xenopus* egg extract are essentially identical to the endogenous H1.8bound nucleosome structure we previously determined. In addition, we have shown that GFP-H1.8 was able to replace the function of endogenous H1.8 to support the proper mitotic chromosome length (Fig. S3), which is based on the capacity of H1.8 to compete with condensin as we have previously demonstrated (PMID 34406118). Therefore, we believe that the effects of GFP-tagging to be minimal. This point incorporated into the main result section (page 6, line 215) to read as “The structures of GFP-tagged H1.8bound nucleosomes isolated from *Xenopus* egg extract chromosomes are essentially identical to the endogenous H1.8-bound nucleosome structure we previously determined. Therefore, although the usage of GFP-tagged H1.8 and MagIC-cryo-EM potentially influence the structure of the H1.8-bound nucleosome, we consider these influences to be minimal.”

Also, the GFP-pulled down H1.8 nucleosomes should be better characterized biochemically to determine the actual linker DNA lengths (which are known to have a strong effect of linker histone affinity) and presence or absence of other factors such as HMG proteins that may compete with linker histones and cause the multiplicity of nucleosome structural classes (such as shown on Fig. 3F) for which the association with H1.8 is uncertain.

We addressed the concerns brought by the reviewer as following:

(1) DNA length

As the reviewer correctly pointed out, linker DNA length is critical for linker histone binding, and conventional ChIP protocols often result in DNA over-digestion to lengths of 140–150 bp. To minimize DNA over-digestion and structural damage, we have optimized a gentle chromosomal nucleosome purification protocol that enabled the cryoEM analysis of chromosomal nucleosomes (PMID: 34478647). This protocol involves DNA digestion with a minimal amount of MNase at 4ºC, producing nucleosomal DNA fragments of 180–200 bp. Additionally, before each chromatin extraction, we performed small-scale MNase assays to ensure that the DNA lengths consistently fell within the 180–200 bp range (Fig. S4B). These DNA lengths are sufficient for linker histone H1 binding, in agreement with previous findings indicating that >170 bp is adequate for linker histone association (PMID: 26212454).

This information has been incorporated into the main text and Methods section;

On page 5, line 178, the sentence was added to read, “To prevent dissociation of H1.8 from nucleosomes during DNA fragmentation, the MNase concentration and the reaction time were optimized to generate DNA fragment lengths with 180–200 bp (Fig. S4B), which is adequate for linker histone association (PMID 26212454).”

On page 32, line 1192, the sentence was added to read, “To digest chromatin, MNase concentration and reaction time were tested on a small scale and optimized to the condition that produces 180-200 bp DNA fragments.”

(2) Co-associated proteins with H1-GFP nucleosome.

We now include mass spectrometry (MS) data for the proteins in the sucrose density gradient fraction 5 used for MagIC-cryo-EM analysis of GFP-H1.8-bound chromatin proteins as well as MS of proteins isolated with the corresponding MagIC-cryo-EM beads (Table S2 and updated Table S5). As the reviewer expected, HMG proteins (hmga2.L and hmga2.S in Table S2) were present in interphase sucrose gradient fraction 5, but their levels were less than 2% of H1.8. Accordingly, none of the known chromatin proteins besides histones and the nucleoplasmin were detected by MS in the GFP-nanobody MagIC-cryo-EM beads, including the FACT complex and PCNA, whose levels in the sucrose fraction were comparable to H1.8 (Table S2), suggesting that our MagIC-cryo-EM analysis was not meaningfully affected by HMG proteins and other chromatin proteins. Consistent with our interpretation, the structural features of H1.8bound nucleosomes isolated from interphase and metaphase chromosomes were essentially identical.

**Reviewer #2 (Public review):**
Summary:The authors present a straightforward and convincing demonstration of a reagent and workflow that they collectively term "MagIC-cryo-EM", in which magnetic nanobeads combined with affinity linkers are used to specifically immobilize and locally concentrate complexes that contain a protein-of-interest. As a proof of concept, they localize, image, and reconstruct H1.8-bound nucleosomes reconstructed from frog egg extracts. The authors additionally devised an image-processing workflow termed "DuSTER", which increases the true positive detections of the partially ordered NPM2 complex. The analysis of the NPM2 complex {plus minus} H1.8 was challenging because only ~60 kDa of protein mass was ordered. Overall, single-particle cryo-EM practitioners should find this study useful.Strengths:The rationale is very logical and the data are convincing.Weaknesses:I have seen an earlier version of this study at a conference. The conference presentation was much easier to follow than the current manuscript. It is as if this manuscript had undergone review at another journal and includes additional experiments to satisfy previous reviewers. Specifically, the NPM2 results don't seem to add much to the main story (MagIC-cryo-EM), and read more like an addendum. The authors could probably publish the NPM2 results separately, which would make the core MagIC results (sans DusTER) easier to read.

We thank the reviewer for constructive comments. We regret to realize that the last portion of the result section, where we have described a detailed analysis of NPM2 structures, was erroneously omitted from the submission due to MS Word's formatting error. We hope that the inclusion of this section will justify the inclusion of the NPM2 analysis. Specifically, we decided to include NPM2 structures to demonstrate that our method successfully determined the structure that had never been reported. Conformational changes in the NPM family have been proposed in previous studies using techniques such as NMR, negative stain EM, and simulations, and these changes are thought to play a critical role in regulating NPM function (PMID: 25772360, 36220893, 38571760), but there has been a confusion in the literature, for example, on the substrate binding site and on whether NPM2 recognizes the substrate as a pentamer or decamer. Despite their low resolution, our new cryo-EM structures of NPM2 suggest that NPM2 recognizes the substrate as a pentamer, identifies potential substrate-binding sites, and indicates the mechanisms underlying NPM2 conformational changes. We believe that publishing these results will provide valuable insights into the NPM research field and help guide and inspire further investigations.

**Reviewer #3 (Public review):**
Summary:In this paper, Arimura et al report a new method, termed MagIC-Cryo-EM, which refers to the method of using magnetic beads to capture specific proteins out of a lysate via, followed immunoprecipitation and deposition on EM grids. The so-enriched proteins can be analzyed structurally. Importantly, the nanoparticles are further functionalized with protein-based spacers, to avoid a distorted halo around the particles. This is a very elegant approach and allows the resolution of the stucture of small amounts of native proteins at atomistic resolution.Here, the authors apply this method to study the chromatosome formation from nucleosomes and the oocyte-specific linker histone H1.8. This allows them to resolve H1.8-containing chromatomosomes from oocyte extract in both interphase and metaphase conditions at 4.3 A resolution, which reveal a common structure with H1 placed right at the dyad and contacting both entry-and exit linker DNA.They then investigate the origin of H1.8 loss during interphase. They identify a nonnucleosomal H1.8-containing complex from interphase preparations. To resolve its structure, the authors develop a protocol (DuSTER) to exclude particles with ambiguous center, revealing particles with five-fold symmetry, that matches the chaperone NPM2. MS and WB confirms that the protein is present in interphase samples but not metaphase. The authors further separate two isoforms, an open and closed form that coexist. Additional densities in the open form suggest that this might be bound H1.8.Strengths:Together this is an important addition to the suite of cryoEM methods, with broad applications. The authors demonstrate the method using interesting applications, showing that the methods work and they can get high resolution structures from nucleosomes in complex with H1 from native environments.Weaknesses:The structures of the NPM2 chaperone is less well resolved, and some of the interpretation in this part seems only weakly justified.

We thank the reviewer for recognizing the significance of our methods and for constructive comments. We regret to realize that the last portion of the result section where we have described detailed analysis of NPM2 structures was erroneously omitted from the submission due to the MS word's formatting error. We hope that inclusion of this section will justify the inclusion of NPM2 analysis. Specifically, we agree that our NPM2 structures are low-resolution and that our interpretations may be revised as higher-resolution structures become available, although we believe that publishing these results will provide valuable insights into the NPM research field and also will illustrate the power of MagIC-cryo-EM and DuSTER. To respond to this criticism, the revised manuscript now clearly describes the limitations of our NPM2 structures while highlighting the key insights. In page 12 line 452, the sentence was added to read, “While DuSTER enables the structural analysis of NPM2 co-isolated with H1.8-GFP, the resulting map quality is modest, and the reported numerical resolution may be overestimated. Furthermore, only partial density for H1.8 is observed. Although structural analysis of small proteins is inherently challenging, it is possible that halo-like scattering further hinder high-resolution structural determination by reducing the S/N ratio. More detailed structural analyses of the NPM2-substrate complex will be addressed in future studies.

**Reviewer #1 (Recommendations for the authors):**
(1) To assess the advantage provided by the new technique for imaging of isolated pure or enriched fractions of native chromatin, the nucleosome structure analysis should be matched by a proper biochemical characterization of the isolated nucleosomes. Nucleosome DNA size is known to greatly affect linker histone affinity and additional proteins like HMG may compete with linker histone for binding. SDS-PAGE of the sucrose gradient fractions (Fig. 3E) shows many nonhistone proteins where H1-GFP appears to be a minor component. However, the gradient fractions contain both bound and unbound proteins. I would suggest that a larger-scale pull-down using the same GFP antibodies and streptavidin beads should be conducted and the captured nucleosome DNA and proteins characterized.

We addressed the concerns brought by the reviewer as following:

(1) DNA length

As the reviewer correctly pointed out, linker DNA length is critical for linker histone binding, and conventional ChIP protocols often result in DNA over-digestion to lengths of 140–150 bp. To minimize DNA over-digestion and structural damage, we have optimized a gentle chromosomal nucleosome purification protocol that enabled the cryoEM analysis of chromosomal nucleosomes (PMID: 34478647). This protocol involves DNA digestion with a minimal amount of MNase at 4ºC, producing nucleosomal DNA fragments of 180–200 bp. Additionally, before each chromatin extraction, we performed small-scale MNase assays to ensure that the DNA lengths consistently fell within the 180–200 bp range (Fig. S4B). These DNA lengths are sufficient for linker histone H1 binding, in agreement with previous findings indicating that >170 bp is adequate for linker histone association (PMID: 26212454).

This information has been incorporated into the main text and Methods section.

On page 5, line 178, the sentence was added to read, “To prevent dissociation of H1.8 from nucleosomes during DNA fragmentation, the MNase concentration and the reaction time were optimized to generate DNA fragment lengths with 180–200 bp (Fig. S4B), which is adequate for linker histone association (PMID 26212454).”

On page 32, line 1192, the sentence was added to read, “To digest chromatin, MNase concentration and reaction time were tested on a small scale and optimized to the condition that produces 180-200 bp DNA fragments.”

(2) Co-associated proteins with H1-GFP nucleosome.

We now include mass spectrometry (MS) data for the proteins in the sucrose density gradient fraction 5 used for MagIC-cryo-EM analysis of GFP-H1.8-bound chromatin proteins as well as MS of proteins isolated with the corresponding MagIC-cryo-EM beads (Table S2 and updated Table S5). As the reviewer expected, HMG proteins (hmga2.L and hmga2.S in Table S2) were present in interphase sucrose gradient fraction 5, but their levels were less than 2% of H1.8. Accordingly, none of known chromatin proteins besides histones and the nucleoplasmin were detected by MS in the GFP-nanobody MagIC-cryo-EM beads, including the FACT complex and PCNA, whose levels in the sucrose fraction were comparable to H1.8 (Table S2), suggesting that our MagIC-cryo-EM analysis was not meaningfully affected by HMG proteins and other chromatin proteins. Consistent with our interpretation, the structural features of H1.8bound nucleosomes isolated from interphase and metaphase chromosomes were essentially identical.

(2) A similar pull-down analysis with quantitation of NPM2 and GFP (in addition to analysis of sucrose gradient fractions) should be conducted to show whether the immune-selected particles do indeed contains a stoichiometric complex of H1.8 with NPM2.

Proteins isolated using MagIC-cryo-EM beads were identified through mass spectrometry (Fig. 4D). The MS signal suggests that the molar ratio of NPM2 is higher than that of H1.8 or sfGFP. This observation is consistent with the idea that an NPM2 pentamer can bind between one and five H1.8-GFP molecules.

(3) The use of recombinant, bacterial produced H1.8- GFP and just one type of antibodies (GFP) are certain limitations of this work. These limitations as well as future steps needed to use antibodies specific for native antigens, such as histone variants and epigenetic modifications should be discussed.

We clarified these points in the “Limitation of the study” section (page 12, line 420). The revised sections are indicated by the underlines below.

“While MagIC-cryo-EM is envisioned as a versatile approach suitable for various biomolecules from diverse sources, including cultured cells and tissues, it has thus far been tested only with H1.8-bound nucleosome and H1.8-bound NPM2, both using antiGFP nanobodies to isolate GFP-tagged H1.8 from chromosomes assembled in

*Xenopus* egg extracts after pre-fractionation of chromatin. To apply MagIC-cryo-EM for the other targets, the following factors must be considered: (1) Pre-fractionation. This step (e.g., density gradient or gel filtration) may be necessary to enrich the target protein in a specific complex from other diverse forms (such as monomeric forms, subcomplexes, and protein aggregates). (2) Avoiding bead aggregation. Beads may be clustered by targets (if the target complex contains multiple affinity tags or is aggregated), nonspecific binders, and the target capture modules. To directly apply antibodies that recognize the native targets and specific modifications, optimization to avoid bead aggregation will be important. (3) Stabilizing complexes. The target complexes must be stable during the sample preparation. Crosslink was necessary for the H1.8-GFP-bound nucleosome. (4) Loading the optimum number of targets on the bead. The optimal number of particles per bead differs depending on target sizes, as larger targets are more likely to overlap. For H1.8-GFP-bound nucleosomes, 500 to 2,000 particles per bead were optimal. We expect that fewer particles should be coated for larger targets.”

**Reviewer #2 (Recommendations for the authors):**
General:Figures: Most of the figures have tiny text and schematic items (like Fig. 2B). To save readers from having to enlarge the paper on their computer screen, consider enlarging the smallest text & figure panels.

We enlarged the text in the main figures.

Is it possible that the MagIC method also keeps more particles "submerged", i.e., away from the air:water interface? Does MagIC change the orientation distribution?

In theory, the preferred orientation bias should be reduced in MagIC-cryo-EM, as particles are submerged, and the bias is thought to arise from particle accumulation at the air-water interface. However, while the preferred orientation appears to be mitigated, the issue is not completely resolved, as demonstrated in Author response image 1. A possible explanation for the remaining preferred orientation bias in MagIC-cryo-EM data is that many particles are localized on graphene-water interfaces.

**Author response image 1. sa3fig1:** Orientation distributions of H1.8-nucleosome particles with or without MagIC-cryo-EM.

Consider adding a safety note to warn about possible pinching injuries when handling neodymium magnets.

This is a good idea. We added a sentence in the method section (page 24, line 878), “The two pieces of strong neodymium magnets have to be handled carefully as magnets can leap and slam together from several feet apart.”

In the methods section, the authors state that the grids were incubated on magnets, followed by blotting and plunge freezing in the Vitrobot. Presumably, the blotting was performed in the absence of magnets. The authors may want to clarify this in the text. If so, can the authors speculate how the magnet-treated beads are better retained on the grids during blotting? Is it due to the induced aggregation and/or deposition of the nanobeads on the grid surface?

In the limitation section (page 12 line 446), the sentence was added to read:

“The efficiency of magnetic bead capture can be further improved. In the current MagICcryo-EM workflow, the cryo-EM grid is incubated on a magnet before being transferred to the Vitrobot for vitrification. However, since the Vitrobot cannot accommodate a strong magnet, the vitrification step occurs without the magnetic force, potentially resulting in bead loss. This limitation could be addressed by developing a new plunge freezer capable of maintaining magnetic force during vitrification.”

In the method section (page 27 line 993), the sentence was modified. The revised sections are indicated by underlines.

“The grid was then incubated on the 40 x 20 mm N52 neodymium disc magnets for 5 min within an in-house high-humidity chamber to facilitate magnetic bead capture. Once the capture was complete, the tweezers anchoring the grid were transferred and attached to the Vitrobot Mark IV (FEI), and the grid was vitrified by employing a 2second blotting time at room temperature under conditions of 100% humidity.”

Do you see an extra density corresponding to the GFP in your averages?

Since GFP is connected to H1.8 via a flexible linker, the GFP structure was observed in complex with the anti-GFP nanobody, separate from the H1.8-nucleosome and H1.8NPM2 complexes, as shown in Fig. S10.

Fig. 5 & Fig. S11: The reported resolutions for NPM2 averages were ~5Å but the densities appear - to my eyes - to resemble a lower-resolution averages.

Although DuSTER enables the 3D structural determination of NPM2 co-isolated with H1-GFP, we recognize that the quality of the NPM2 map falls short of the standard expected for a typical 5 Å-resolution map. To appropriately convey the quality of the NPM2 maps, we have included the 3D FSC and local resolution map of the NPM2 structure (new Fig. S12). Furthermore, we have revised the manuscript to deemphasize the resolution of the NPM2 structure to avoid any potential misinterpretation.

Fig. 5D: The cartoon says: "less H1.8 on interphase nucleosome" and "more H1.8 on metaphase nucleosome". Please help the readers understand this conclusion with the gel in Fig. 3C and the population histograms in Fig. 3F.

As depicted in Fig. 3A, we previously identified the preferential binding of H1.8 to metaphase nucleosomes (PMID: 34478647). In this study, to obtain sufficient H1.8bound nucleosomes for MagIC-cryo-EM, we used 2.5 times more starting material for interphase samples compared to M-phase samples. This discrepancy complicates the comparison of H1-GFP binding ratios in western blots. However, in GelCode Blue staining (Fig. S4A), where both H1-GFP and histone bands are visible, the preferential binding of H1.8 to metaphase nucleosomes can be observed (See fractions 11 in interphase and metaphase).

Abstract - that removes low signal-to-noise ratio particles -> to exclude low signal-tonoise ratio particles; The term "exclude" is more accurate and is in the DuSTER acronym itself.

We edited it accordingly.

P1 - to reduce sample volume/concentration -> to lower the sample volume/concentration needed

We edited it accordingly.

P1 - Flow from 1st to 2nd paragraph could be improved. It's abrupt. Maybe say that some forms of nucleoprotein complexes are rare, with one example being H1.8-bound nucleosomes in interphase chromatin?

We have revised the text to address the challenges involved in the structural characterization of native chromatin-associated protein complexes. The revised text reads, “Structural characterization of native chromatin-associated protein complexes is particularly challenging due to their heterogeneity and scarcity: more than 300 proteins directly bind to the histone core surface, while each of these proteins is targeted to only a fraction of nucleosomes in chromatin.”

P2 - interacts both sides of the linker DNA -> interacts with both the entry and exit linker DNA

We have edited it accordingly.

P2 - "from the chromatin sample isolated from metaphase chromosomes but not from interphase chromosomes" - meaning that the interphase nucleosomes don't have H1.8 densities at all, or that they do, but the H1.8 only interacts with one of the two linker DNAs?

In our original attempt to analyze nucleosome structures assembled in *Xenopus* egg extracts without MagIC-cryo-EM, we were not able to detect the density confidently assigned to H1.8 in interphase chromatin samples. To avoid potential confusion, the revised text reads, “We were able to resolve the 3D structure of the H1.8-bound nucleosome isolated from metaphase chromosomes but not from interphase chromosomes(3). The resolved structure indicated that H1.8 in metaphase is most stably bound to the nucleosome at the on-dyad position, in which H1 interacts with both the entry and exit linker DNAs(21–24). This stable H1 association to the nucleosome in metaphase likely reflects its role in controlling the size and the shape of mitotic chromosomes through limiting chromatin accessibility of condensins(25), but it remains unclear why H1.8 binding to the nucleosome in interphase is less stable. Since the low abundance of H1.8-bound nucleosomes in interphase chromatin might have prevented us from determining their structure, we sought to solve this issue by enriching H1.8bound nucleoprotein complexes through adapting ChIP-based methods.”

P1, P2 - The logical leap from "by adapting ChIP-based methods" to MagIC is not clear.

We addressed this point by revising the text as shown above.

P2 - "Intense halo-like noise" - This is an awkward term. These are probably the Fresnel fringes that arise from underfocus. I wouldn't call this phenomenon "noise". https://www.jeol.com/words/emterms/20121023.093457.php

We re-phrased it as “halo-like scattering”.

P3 -It may help readers to explain how cryo-EM structures of the H1.8-associated interphase nucleosomes would differentiate from the two models in Fig. 3A.

We have revised the introduction section (lines 43~75), including the corresponding paragraph to address the comments above, highlighting the motivation behind determining the structures of interphase and metaphase H1.8-associated nucleosomes. We hope the revisions are now clear.

P6 - "they were masked by background noise from the ice, graphene". I thought that graphene would be contribute minimal noise because it is only one-carbon-layer thick?

That is a valid point. We have removed the term ‘graphene’ from the sentence.

P6 - What was the rationale to focus on particles with 60 - 80Å dimensions?

We observed that 60–80 Å particles were captured by MagIC-cryo-EM beads, as numerous particles of this size were clearly visible in the motion-corrected micrographs surrounding the beads. To clarify this, we revised the sentence to read: 'Topaz successfully picked most of the 60–80 Å particles visible in the motion-corrected micrographs and enriched around the MagIC-cryo-EM beads (Figure S6A).

P7 - Please explain a technical detail about DuSTER: do independent runs of Topaz picks give particle centers than differ by up to ~40Å or is it that 2D classification gives particle centers that differ by up to ~40Å? Is it possible to distinguish these two possibilities by initializing CryoSPARC on two independent 2D classification jobs on the same set of Topaz picks?

Due to the small particle size of NPM2, the former type is predominantly generated when Topaz fails to pick the particles reproducibly. The first cycle of DuSTER removes both former-type particles (irreproducibly picked particles) and latter-type particles (irreproducibly centered particles), while subsequent cycles specifically target and remove the latter type. We have added the following sentence to clarify this (page 7, line 249). The revised sections are indicated by underlines below: “To assess the reproducibility of the particle recentering during 2D classification, two independent particle pickings were conducted by Topaz so that each particle on the grid has up to two picked points (Figure 4A, second left panel). Some particles that only have one picked point will be removed in a later step. These picked points were independently subjected to 2D classification. After recentering the picked points by 2D classification, distances (*D*) between recentered points from the first picking process and other recentered points from the second picking process were measured. DuSTER keeps recentered points whose *D* are shorter than a threshold distance (*DTH*). By setting *DTH* = 20 Å, 2D classification results were dramatically improved in this sample; a five-petal flower-shaped 2D class was reconstructed (Figure 4B). This step also removes the particles that only have one picked point.“

P8 - NPM2 was introduced rather abruptly (it was used as an initial model for 3D refinement). I see NPM2 appear in the supplemental figures cited before the text in P8, but the significance of NPM2 was not discussed there. The authors seem to have made a logical leap that is not explained.

We have removed the term NPM2 in P8.

P9 - "extra cryo-EM densities, which likely represent H1." This statement would be better supported if the resolution of the reconstruction was high enough to resolve H1specific amino acids in the "extra densities" protruding from the petals.

We concurred and softened the statement to read “extra cryo-EM densities, which may represent H1.8,”

P9 - "Notably, extra cryo-EM densities, which likely represent H1.8, are clearly observed in the open form but much less in the closed form near the acidic surface regions proximal to the N terminus of beta-1 and the C terminus of beta-8 (Fig. 5A and 5B)." It would be helpful to point out where the "extra densities" are in the figure for the open and closed form. Some readers may not be able to extrapolate from the single red arrow to the other extra densities.

Thank you for your comment. We have pointed out the density in the Fig 5A as well.

P9 - "Supporting this idea, the acidic tract A1 (aa 36-40) and A2 (aa 120-140) are both implicated in the recognition of basic substrates such as core histones..." Did this sentence get cut off in the next column?

We apologize for our oversight on this error. Due to an MS Word formatting error, the sentences (lines 316–343) were hidden beneath a figure. We have retrieved the missing sentences:

“Supporting this idea, the acidic tract A1 (aa 36-40) and A2 (aa 120-140), which are both implicated in recognition of basic substrates such as core histones(43,50), respectively interact with and are adjacent to the putative H1.8 density (Figure 5B). In addition, the NPM2 surface that is in direct contact with the putative H1.8 density is accessible in the open form while it is internalized in the closed form (Figure 5C). This structural change of NPM2 may support more rigid binding of H1.8 to the open NPM2, whereas H1.8 binding to the closed form is less stable and likely occurs through interactions with the C-terminal A2 and A3 tracts, which are not visible in our cryo-EM structures.

In the aforementioned NPM2-H1.8 structures, for which we applied C5 symmetry during the 3D structure reconstruction, only a partial H1.8 density could be seen (Figure 5B). One possibility is that H1.8 structure in NPM2-H1.8 does not follow C5 symmetry. As the size of the NPM2-H1.8 complex estimated from sucrose gradient elution volume is consistent with pentameric NPM2 binding to a single H1.8 (Figure 3C and Table S3), applying C5 symmetry during structural reconstruction likely blurred the density of the monomeric H1.8 that binds to the NPM2 pentamer. The structural determination of NPM2-H1.8 without applying C5 symmetry lowered the overall resolution but visualized multiple structural variants of the NPM2 protomer with different degrees of openness coexisting within a NPM2-H1.8 complex (Figure S14), raising a possibility that opening of a portion of the NPM2 pentamer may affect modes of H1.8 binding. Although more detailed structural analyses of the NPM2-substrate complex are subject of future studies, MagIC-cryo-EM and DuSTER revealed structural changes of NPM2 that was co-isolated H1.8 on interphase chromosomes.

Discussion

MagIC-cryo-EM offers sub-nanometer resolution structural determination using a heterogeneous sample that contains the target molecule at 1~2 nM, which is approximately 100 to 1000 times lower than the concentration required for conventional cryo-EM methods, including affinity grid approach(9–11).”

**Reviewer #3 (Recommendations for the authors):**
All with regards to the NPM2 part:It would be great if the authors could provide micrographs where the particles are visible, in addition to the classes.

The particles on the motion-corrected micrographs are available in Fig S9.

Also, the angular distribution in the SI looks very uniform.I also wonder, if the authors could indicate the local resolution for all structures.Could the authors provide the 3D FSC for NPM2?

Although DuSTER enables the 3D structural determination of NPM2 co-isolated with H1-GFP, we recognize that the quality of the NPM2 map falls short of the standard expected for a typical 5 Å resolution map. To appropriately convey the quality of the NPM2 maps, we have included the 3D FSC and local resolution map of the NPM2 structure (new Fig. S12).

I really cannot see a difference between the open and closed forms. Looking at the models, I am skeptical that the authors can differentiate the two forms with the available resolution. Could they provide statistics that support their assignments?

To better highlight the structural differences between the two forms, we added a new figure to compare the maps between open and closed forms (Fig S12J-K).

Also, the 'additional density' representing H1.8 in the NPM2 structures - I cannot see it.

We pointed out the density with the red arrow in the revised Fig 5A.

Minor comments:Something is missing at the end of Results, just before the beginning of the Discussion. The figure legend for Fig. S12 is truncated, so it is unclear what is going on

We apologize for our oversight on this error. Due to an MS Word formatting error, the sentences (lines 316–343) were hidden beneath a figure. We have retrieved the missing sentences:

“Supporting this idea, the acidic tract A1 (aa 36-40) and A2 (aa 120-140), which are both implicated in recognition of basic substrates such as core histones(43,50), respectively interact with and are adjacent to the putative H1.8 density (Figure 5B). In addition, the NPM2 surface that is in direct contact with the putative H1.8 density is accessible in the open form while it is internalized in the closed form (Figure 5C). This structural change of NPM2 may support more rigid binding of H1.8 to the open NPM2, whereas H1.8 binding to the closed form is less stable and likely occurs through interactions with the C-terminal A2 and A3 tracts, which are not visible in our cryo-EM structures.

In the aforementioned NPM2-H1.8 structures, for which we applied C5 symmetry during the 3D structure reconstruction, only a partial H1.8 density could be seen (Figure 5B). One possibility is that H1.8 structure in NPM2-H1.8 does not follow C5 symmetry. As the size of the NPM2-H1.8 complex estimated from sucrose gradient elution volume is consistent with pentameric NPM2 binding to a single H1.8 (Figure 3C and Table S2), applying C5 symmetry during structural reconstruction likely blurred the density of the monomeric H1.8 that binds to the NPM2 pentamer. The structural determination of NPM2-H1.8 without applying C5 symmetry lowered the overall resolution but visualized multiple structural variants of the NPM2 protomer with different degrees of openness coexisting within a NPM2-H1.8 complex (Figure S14), raising a possibility that opening of a portion of the NPM2 pentamer may affect modes of H1.8 binding. Although more detailed structural analyses of the NPM2-substrate complex are subject of future studies, MagIC-cryo-EM and DuSTER revealed structural changes of NPM2 that was co-isolated H1.8 on interphase chromosomes.

Discussion

MagIC-cryo-EM offers sub-nanometer resolution structural determination using a heterogeneous sample that contains the target molecule at 1~2 nM, which is approximately 100 to 1000 times lower than the concentration required for conventional cryo-EM methods, including affinity grid approach(9–11).”

Figure S13: I am not sure how robust these assignments are at this low resolution. Are these real structures or classification artifacts? It feels very optimistic to interpret these structures

We agree that our NPM2 structures are low-resolution and that our interpretations may be revised as higher-resolution structures become available, although we believe that publishing these results will provide valuable insights into the NPM research field and also will illustrate the power of MagIC-cryo-EM and DuSTER. Conformational changes in the NPM family have been proposed in previous studies using techniques such as NMR, negative stain EM, and simulations, and these changes are thought to play a critical role in regulating NPM function (PMID: 25772360, 36220893, 38571760), but there has been a confusion in the literature, for example, on the substrate binding site and on whether NPM2 recognizes the substrate as a pentamer or decamer. Despite their low resolution, our new cryo-EM structures of NPM2 suggest that NPM2 recognizes the substrate as a pentamer, identify potential substrate-binding sites, and indicate the mechanisms underlying NPM2 conformational changes. We believe that publishing these results will provide valuable insights into the NPM research field and help guide and inspire further investigations.

To respond to this criticism, we have revised the manuscript to clearly describe the limitations of our NPM2 structures while highlighting the key insights. On page 12, line 452, the sentence was added to read, “While DuSTER enables the structural analysis of NPM2 co-isolated with H1.8-GFP, the resulting map quality is modest, and the reported numerical resolution may be overestimated. Furthermore, only partial density for H1.8 is observed. Although structural analysis of small proteins is inherently challenging, it is possible that halo-like scattering further hinders high-resolution structural determination by reducing the S/N ratio. More detailed structural analyses of the NPM2-substrate complex will be addressed in future studies.”